# Exceptionally high biosphere productivity at the beginning of Marine Isotopic Stage 11

Margaux Brandon [1,2]✉, Amaelle Landais[1], Stéphanie Duchamp-Alphonse[2], Violaine Favre[1], Léa Schmitz[1], Héloïse Abrial[1], Frédéric Prié[1], Thomas Extier [1] & Thomas Blunier [3]

Significant changes in atmospheric $CO_2$ over glacial-interglacial cycles have mainly been attributed to the Southern Ocean through physical and biological processes. However, little is known about the contribution of global biosphere productivity, associated with important $CO_2$ fluxes. Here we present the first high resolution record of $\Delta^{17}O$ of $O_2$ in the Antarctic EPICA Dome C ice core over Termination V and Marine Isotopic Stage (MIS) 11 and reconstruct the global oxygen biosphere productivity over the last 445 ka. Our data show that compared to the younger terminations, biosphere productivity at the end of Termination V is 10 to 30 % higher. Comparisons with local palaeo observations suggest that strong terrestrial productivity in a context of low eccentricity might explain this pattern. We propose that higher biosphere productivity could have maintained low atmospheric $CO_2$ at the beginning of MIS 11, thus highlighting its control on the global climate during Termination V.

[1] Université Paris-Saclay, CNRS, CEA, UVSQ, Laboratoire des sciences du climat et de l'environnement, 91191 Gif-sur-Yvette, France. [2] Université Paris-Saclay, CNRS, GEOPS, 91405 Orsay, France. [3] Centre for Ice and Climate, Niels Bohr Institute, University of Copenhagen, Copenhagen, Denmark. ✉email: margaux.brandon@universite-paris-saclay.fr

The largest pre-anthropogenic changes in atmospheric $CO_2$ concentration of the last 800,000 years are observed during deglaciations, with increases of up to 100 ppm in a few thousand years[1]. Oceanic carbon reservoir is broadly suspected to play a central role in these atmospheric $CO_2$ increases. Leading hypotheses invoke $CO_2$ degassing from the ocean induced by more vigorous convection[2] and concomitant decrease of the net organic matter export in the Southern Ocean[3], enhanced exchanges between ocean surface and atmosphere due to sea-ice melting[4], and increased sea-surface temperature[5]. Modelling studies simulating changes in oceanic processes are quite controversial[6,7] but in all cases, they are not able to explain the full increase of atmospheric $CO_2$ rises during deglaciations. In parallel, terrestrial primary productivity and carbon stocks increase during deglaciations, thus acting as a significant $CO_2$ land sink[8,9] so that an important additional source of $CO_2$ is required to explain the entire deglacial $CO_2$ pattern.

Quantifying changes in the carbon cycle over deglaciations relies on data compilation and modelling studies to estimate both carbon stocks and carbon fluxes[8]. Some information on carbon stocks can be obtained from $\delta^{13}C$ of carbonates using a wealth of data obtained from marine sediments[10] and $\delta^{13}C$ of atmospheric $CO_2$[11,12]. In parallel, information on the evolution of past global productivity, a major component of carbon flux, is very sparse and often limited to the last deglaciation[8,9].

In the ocean, past changes in biological carbon pump are best represented by changes in buried organic biomarkers combined with marine Total Organic Carbon/Particulate Inorganic Carbon ratio (TOC/$CaCO_3$), suggested to reflect the C-rain ratio[13]. However, despite their accuracy to provide biological export production, only one site combine such records in the Southern Ocean, for the last 800 ka, so far[14].

On continents, pollen counting and sedimentary TOC are useful for biosphere productivity reconstruction[15,16] but they are unfortunately indirect and rely on the use of biosphere models[9]. Moreover, similarly to oceanic records, these observations provide regional records that are not easy to use for documenting the past global carbon cycle. Ciais et al.[8] proposed to use the isotopic composition of oxygen of atmosphere ($\delta^{18}O_{atm}$) as a tracer for terrestrial biosphere productivity. However, this proxy is a complex tracer being influenced by hydrological cycle at first order[17,18] and its use as a quantitative tool for productivity reconstruction depends on the exact determination of associated fractionation factors in the water and biosphere cycles[19].

A total estimate of the global biospheric fluxes and their temporal variations can be obtained more directly from measurements of $\Delta^{17}O$ of $O_2$ ($\ln(\delta^{17}O + 1) - 0.516*\ln(\delta^{18}O + 1)$) in ice cores[20–22]. This method provides $O_2$ fluxes and the conversion from $O_2$ to $CO_2$ fluxes can be done from the stoichiometry of the biological processes of photosynthesis and respiration[23]. $\Delta^{17}O$ of $O_2$ measures the variation of the triple isotopic composition of atmospheric $O_2$ with respect to modern oxygen so that by definition, $\Delta^{17}O$ of $O_2$ is nil today. Previous experimental studies showed that $\Delta^{17}O$ of $O_2$ increases in a closed biospheric system when the exchanges with the stratosphere are prevented: the biological productivity leads to $\Delta^{17}O$ of $O_2$ increase while photochemical reactions occurring in the stratosphere have the effect of decreasing $\Delta^{17}O$ of $O_2$[20]. For paleoproductivity reconstructions, measurements of the evolution of $\Delta^{17}O$ of $O_2$ in ancient air trapped in the Vostok and GISP2 ice cores already provided information on the evolution of global biosphere productivity, over the last 400 ka[21,22]. The results show a systematic larger productivity during interglacial than during glacial periods with interglacial levels remaining close to the current biosphere productivity.

Over the last nine deglaciations, Termination V (433–426 ka on ice core records on the latest AICC2012 chronology[24]) is probably the most intriguing. This Termination is framed by the particularly long and strong glacial Marine Isotopic Stage 12 (MIS 12), followed by the long and warm interglacial MIS 11 (426–398 ka on AICC2012). This is the first Termination after the Mid-Brunhes event marking a fundamental change in the climate system from mild to warm periods with associated lower to higher $CO_2$ concentrations. Termination V is also occurring in a particular orbital context of low eccentricity, which is known to have an influence on the carbon cycle as observed in $\delta^{13}C$ oceanic records[25]. On the continents, pollen data[16,26] suggest a strong and long increase in terrestrial productivity during MIS 11. In the Ocean, MIS 11 displays an unusual increase in carbonate storage in low[27] and high-latitude environments[28]. Yet, while it is clearly associated with a major phase in coral reef expansion[29] and a climax in calcareous phytoplankton productivity respectively[30], the impact of this large carbonate production on atmospheric $pCO_2$ is not understood. Therefore, the biosphere productivity fluxes during Termination V and MIS 11 need to be investigated.

Here we present the first measurements of the triple isotopic composition of atmospheric oxygen ($\Delta^{17}O$ of $O_2$) in the Antarctic EPICA Dome C ice core over Termination V. Using these measurements and new correction factors compared to the previous record of Blunier et al.[22], we reconstruct the oxygen fluxes associated with biosphere productivity over this particularly strong Termination. The biosphere productivity over Termination V and the beginning of MIS 11 is found to be 10–30% higher than productivity over the pre-industrial period, an exceptional value never encountered over the last four interglacial periods. The ice core $\delta^{18}O_{atm}$ (or $\delta^{18}O$ of $O_2$) record, and terrestrial and oceanic records related to biosphere productivity are used to discuss the relative contribution of changes in oceanic vs terrestrial biosphere fluxes to the atmospheric $CO_2$ rise during Termination V. We show that this productivity peak is most probably due to an increase of the terrestrial productivity during this period favoured by a particular context of low eccentricity. We propose that such strong productivity occurring concomitantly with an exceptional productivity carbonate peak in the marine realm, plays a role in maintaining the $CO_2$ level at a relatively low level at the beginning of MIS 11.

## Results

**Record of $\Delta^{17}O$ of $O_2$ during Termination V**. The Antarctic EPICA Dome C (EDC) ice core (75°06′S; 123°21′E, 3233 m above sea level) covers the last 800 ka. We measured $\Delta^{17}O$ of $O_2$ over the depth range 2735–2797 m, corresponding to an age between 405.7 and 444.1 ka with an average resolution of 780 years (50 samples) (Fig. 1). The raw data were corrected for several effects (Methods): values of the atmospheric air $\Delta^{17}O$ of $O_2$ were obtained by corrections for gravitational fractionation, air bubble trapping fractionation and gas loss fractionation. Compared to the previous records[20–22], several corrections were addressed for the first time here (gas loss and air bubble trapping effects, see Supplementary Table 1). To check the coherency between the previous records that did not take into account these corrections, and the new one obtained on EDC, we measured $\Delta^{17}O$ of $O_2$ over Termination II on the EDC ice core and compared our corrected $\Delta^{17}O$ of $O_2$ curve with the previous record obtained by Blunier et al.[22] on the Vostok ice core (Fig. 1). The two records are in agreement, displaying the same 51 ppm decrease over Termination II without any shift in the mean $\Delta^{17}O$ of $O_2$ value because of compensating effects in the different corrections (see Methods). We thus conclude that our new record over Termination V can directly be used to complete the previous $\Delta^{17}O$ of $O_2$ record from

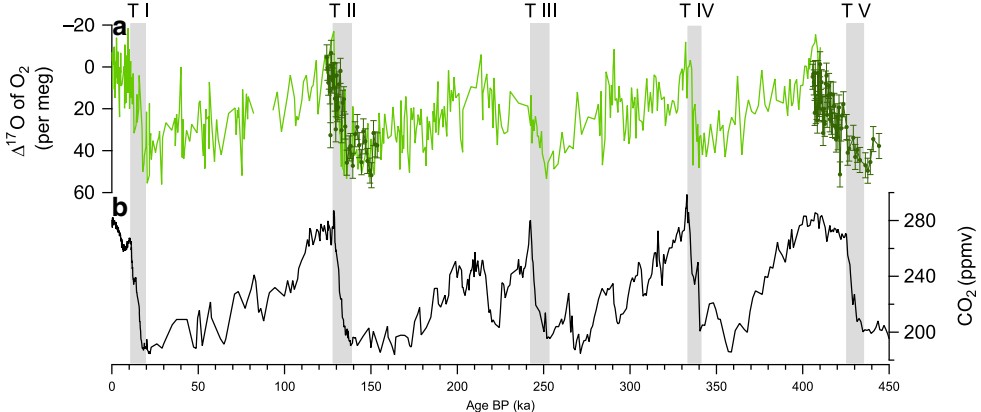

**Fig. 1 Record of $\Delta^{17}$O of $O_2$ over the last 450 ka compared to evolution of $CO_2$. a** $\Delta^{17}$O of $O_2$ (light green: record presented in Blunier et al.[21,22] covering the last 400 ka; dark green: new record with error bars showing the standard deviation of ± 6 per meg). **b** atmospheric $CO_2$ variations over the last 450 ka[69]. The ice core records are presented on the AICC2012 timescale[24,70]. The grey shadow bars represent the period of rapid increase in atmospheric $CO_2$.

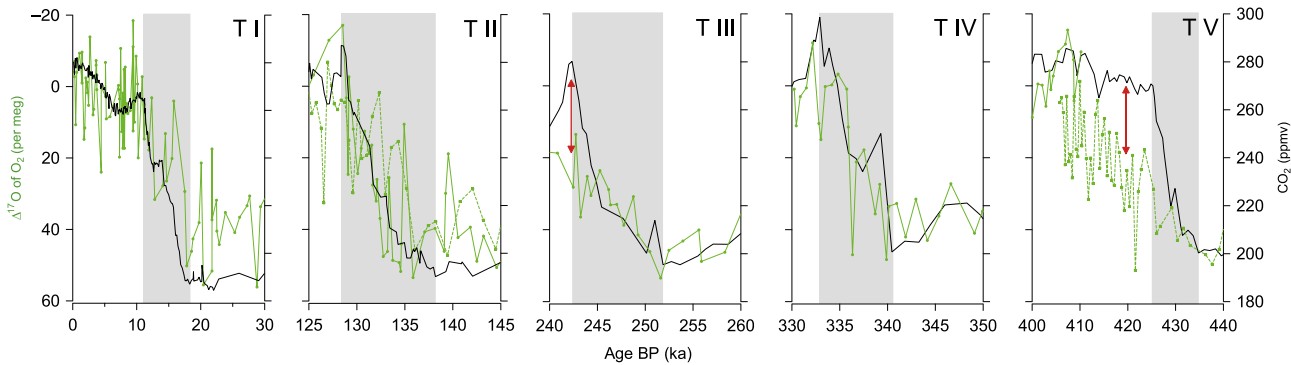

**Fig. 2 Comparison of $CO_2$ and $\Delta^{17}$O of $O_2$ evolutions over the last five terminations.** $CO_2$[69] (black); $\Delta^{17}$O of $O_2$ combining new data (Termination II and Termination V, dotted green line) and data from Blunier et al.[21,22] (solid green line). The red arrows indicate the strongest differences between the $CO_2$ and $\Delta^{17}$O of $O_2$ evolutions during interglacial periods. The grey shadow bars correspond to the main increase in atmospheric $CO_2$ over terminations.

Blunier et al.[22] and provide a full record of $\Delta^{17}$O of $O_2$ over the last five deglaciations.

Termination V displays a decrease in $\Delta^{17}$O of $O_2$ (56 ppm) of similar amplitude than the younger deglaciations (53, 35, 51 and 55 ppm for Terminations IV, III, II and I, respectively). The main difference with previous terminations is on the duration and timing of the $\Delta^{17}$O of $O_2$ decreases. For the four youngest terminations, the $\Delta^{17}$O of $O_2$ decreases are in parallel to the $CO_2$ increases[21,22]. Over Termination V, the main decrease of $\Delta^{17}$O of $O_2$ is more than twice as long as the $CO_2$ increase (from 434.8 to 410.2 ka and from 434.8 to 424.8 ka, i.e. 24.6 ka and 10 ka respectively, Fig. 2). This result contrasts with previous studies covering the last four deglaciations[21,22] where decrease of $\Delta^{17}$O of $O_2$ and increase of atmospheric $CO_2$ concentration are synchronous (Supplementary Fig. 1).

Focusing on the last five deglaciations, only two cases can be identified with a significant deviation between the $CO_2$ and the $\Delta^{17}$O of $O_2$ evolutions during the deglaciations (red arrows on Fig. 2): the first case at the very end of Termination III, between 240 and 245 ka, when the short peak in atmospheric $CO_2$ concentration does not correspond to a sharp decrease in $\Delta^{17}$O of $O_2$ and the second case over Termination V, between 430 and 415 ka, when the maximum in $CO_2$ concentration is reached at 425 ka, more than 10 ka before the minimum in $\Delta^{17}$O of $O_2$ recorded at 415 ka. Contrary to the case of Termination III where the deviation of the $\Delta^{17}$O of $O_2$ signal from the $CO_2$ optimum is

only based on three points, the mismatch between $CO_2$ and $\Delta^{17}$O of $O_2$ evolutions over Termination V relies on 40 data points of $\Delta^{17}$O of $O_2$. When looking at the anticorrelation between $\Delta^{17}$O of $O_2$ and $CO_2$ evolutions over the last five deglaciations, Termination V stands out with a significantly much lower slope for the relationship between $\Delta^{17}$O of $O_2$ and $CO_2$ than the last four Terminations (Supplementary Fig. 1). In addition, to highlight the specificity of Termination V, our new data hence show that the systematic use of $\Delta^{17}$O of $O_2$ to infer $CO_2$ values when $CO_2$ measurements are not available[31] is not always a valid approach (Supplementary Fig. 1).

The general anti-correlations between $CO_2$ and $\Delta^{17}$O of $O_2$ over terminations are explained at first order by stratospheric reactions. These reactions are leading to formation of ozone from the dioxygen reservoir and are associated with mass independent isotopic fractionations. These fractionations lead to an isotopic anomaly of ozone that can be transmitted to $CO_2$. There is eventually a correspondence between the $\Delta^{17}$O of $CO_2$ and the $\Delta^{17}$O of $O_2$, and it has been assumed that this anomaly transmission is rate-limited by the abundance of $CO_2$[20]. As a consequence, an increase in $CO_2$ in the absence of changes in biosphere productivity should be associated with a decrease in $\Delta^{17}$O of $O_2$. The increase of biosphere productivity comes as an opposite effect: it produces oxygen with mass dependent isotopic fractionation which has the effect of increasing $\Delta^{17}$O of $O_2$. The similar $CO_2$ and $\Delta^{17}$O of $O_2$ anticorrelation observed over the

latest terminations is hence an indication that biosphere productivity variations are parallel to the variations of atmospheric concentration in $CO_2$ and that a constant relationship exists between the amplitudes of biosphere productivity variations and the amplitudes of atmospheric $CO_2$ variations. As a consequence, the deviation of the classical $CO_2$ vs $\Delta^{17}O$ of $O_2$ anticorrelation toward unexpectedly high $\Delta^{17}O$ of $O_2$ values should be interpreted as an anomalous increase in biosphere productivity compared to the situation for other terminations. We thus propose that the anomalously strong difference in the $CO_2$ and $\Delta^{17}O$ of $O_2$ evolutions over the first part of MIS 11 at the end of Termination V is associated with a stronger biosphere productivity compared to the onset of the 4 last interglacial periods. We are aware that alternative possibilities can be suggested such as a decrease in the stratospheric ozone in agreement with relatively high $N_2O$ atmospheric concentration (sink of ozone) during MIS 11[32]. However, similarly high $N_2O$ atmospheric concentration is observed over the first part of MIS 9 without associated decoupling between atmospheric $CO_2$ increase and $\Delta^{17}O$ of $O_2$ decrease. In the following, we thus favour the interpretation in terms of an anomalous change of paleoproductivity during Termination V compared to younger terminations.

**Reconstruction of the global oxygen biosphere productivity**. For a more quantitative reconstruction of paleoproductivity based on the classical use of $\Delta^{17}O$ of $O_2$, we followed the approach detailed in Landais et al.[33] which relies on estimates of the $\Delta^{17}O$ of $O_2$ values produced by the terrestrial and oceanic biospheres. Such values are based on previously determined values of oxygen fractionation coefficients for the different biologic processes of oxygen uptake and production[34]. Because of different proportions of C3 vs C4 plants or terrestrial vs oceanic biosphere productivity ratios, the global $\Delta^{17}O$ of $O_2$ value produced by the biosphere can display strong variations between glacial and interglacial periods.

The reconstructed biosphere productivity using the Landais et al.[33] formulation and uncertainties is detailed in Supplementary Table 2 for different time periods of interest within this study, i.e. the pre-industrial, the LGM and the MIS 11. On average, we obtain a biosphere productivity at the beginning of MIS 11 which is 17 % higher than during MIS 1 and a biosphere productivity during the LGM which is 31% lower than during MIS 1, a result in agreement with output of the IPSL coupled model equipped with vegetation and ocean productivity modules (Supplementary Fig. 4).

As in Landais et al.[33], it is shown in Supplementary Table 2 that the uncertainty in fractionation coefficients as well as on the ratio of oceanic to terrestrial productivity is leading to uncertainties in the reconstructed past global productivity. The fractionation coefficients are based on physical properties and are hence not expected to vary with time so that the possible bias on these coefficients should apply to the different periods. On opposite, the ratio of oceanic to terrestrial productivity is expected to vary with time and is a large source of uncertainty. We thus present in Supplementary Table 2 calculations performed with the largest range of possible ratios of oceanic to terrestrial productivity estimated in Landais et al.[33].

Since the publication of Landais et al.[33], new estimates of fractionation coefficients within the oxygen cycle are available[19,35,36] and influence of the water cycle organisation has also been suggested[22]. As a consequence, we performed three additional types of sensitivity tests to better estimate the uncertainties related to fractionation processes within the photosynthesis[37], respiration[36] and water cycle[22] (see SOM for details).

First, we address the uncertainty in photosynthesis fractionation. Some marine species show fractionation during marine photosynthesis[37] while it was assumed in Landais et al.[33] that photosynthesis does not fractionate[38]. We performed sensitivity tests with the largest fractionation effects during marine photosynthesis observed in Eisenstadt et al.[37] and obtained a MIS 11 productivity level decreased by 3% with such fractionation effect.

The second uncertainty we tested is the uncertainty in respiration fractionation. Recent studies have highlighted large variations in the relationship between $\delta^{17}O$ and $\delta^{18}O$ during respiration linked to temperature variations[36]. When taking into account the maximum effect, i.e. a decrease of 0.005 for the slope of the relationship between $\ln(\delta^{17}O + 1)$ and $\ln(\delta^{18}O + 1)$ during respiration, we end up with a resulting biosphere productivity reconstruction 16% higher during MIS 11 than the average situation.

Then, we estimated the uncertainty in the fractionation within water cycle. The general assumption in Landais et al.[33] was that the relationship between $\delta^{17}O$ and $\delta^{18}O$ of water (meteoric water line) remains the same over glacial-interglacial cycles. However, measurements in the Vostok ice core have shown that the $^{17}O$-excess, defined as $^{17}O\text{-excess} = \ln(1 + \delta^{17}O) - 0.528 * \ln(1 + \delta^{18}O)$, is lower by up to 20 ppm during the last glacial maximum[39]. While it has been shown that this is a local effect[40], we still performed a sensitivity test with decreasing $^{17}O$-excess of all continental meteoric waters by 20 ppm during glacial periods and increasing $^{17}O$-excess by 10 ppm during MIS 11 with respect to our current interglacial period. Taking into account such variations, the reconstructed biosphere productivity does not differ by more than 6% from the case without change in meteoric water $^{17}O$-excess (Supplementary Table 2).

Finally, we also took into account the uncertainty in the model used by Landais et al.[33] by combining the above reconstruction with an alternative reconstruction of global oxygen productivity using an alternative model, the one of Blunier et al.[22], forced by our $\Delta^{17}O$ of $O_2$ data (Supplementary Fig. 2).

Combining our different sensitivity studies for a global reconstruction of biosphere productivity (Fig. 3), we find that the global productivity is reduced by 10–40% during glacial periods compared to interglacials. At the beginning of most interglacial periods, the global productivity remains close to the pre-industrial level. The only exception to this general behaviour is the strong oxygen biosphere productivity at the end of Termination V reaching values 10–30% higher than during the pre-industrial period.

## Discussion

Complementary information on the origin and specificity of the $\Delta^{17}O$ of $O_2$ signal over Termination V can be obtained from the ice core $\delta^{18}O_{atm}$ record over the last 800 ka[18] (Fig. 3). $\delta^{18}O_{atm}$ is a complex parameter resulting from both biosphere productivity and low-latitude water cycle[41]. In particular, it has been shown that Weak Monsoon Intervals observed during Heinrich events lead to increases in $\delta^{18}O_{atm}$ via changes in the low-latitude water cycle[42]. Another way to increase the $\delta^{18}O_{atm}$ is to increase the ratio between terrestrial and oceanic biosphere productivity[8,41].

As for $\Delta^{17}O$ of $O_2$, $\delta^{18}O_{atm}$ shows a particular behaviour over Termination V. First, while $\delta^{18}O_{atm}$ variations during other terminations of the last 800 ka do not exhibit variations larger than 1.5 ‰[18], the $\delta^{18}O_{atm}$ decrease over Termination V is of 1.8 ‰. The highest level of $\delta^{18}O_{atm}$ (1.43 ‰) is reached during Termination V at the beginning of the main increase of productivity inferred from $\Delta^{17}O$ of $O_2$. In contrast with the other interglacial periods where the maximum in biosphere productivity corresponds to a minimum of $\delta^{18}O_{atm}$, there is no minimum of $\delta^{18}O_{atm}$ observed at the beginning of MIS 11 (~420 ka) (Fig. 3).

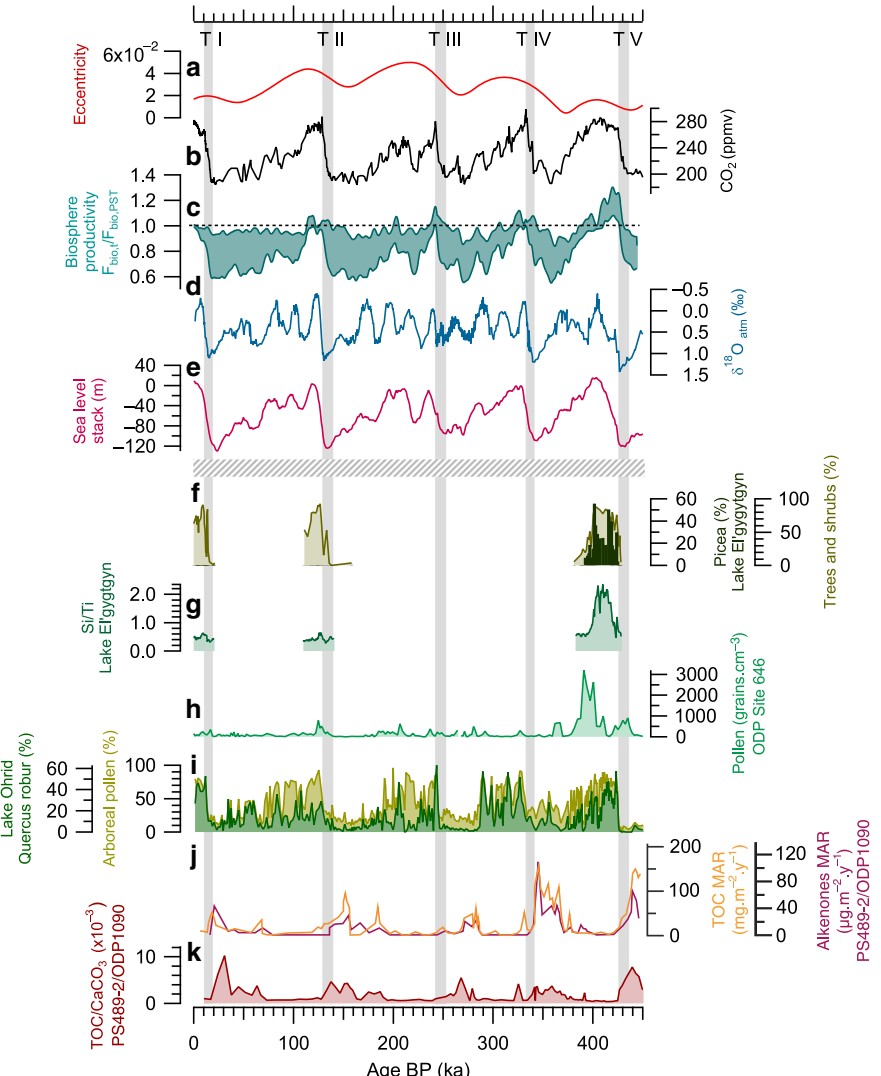

**Fig. 3 Global and regional productivity records since 450 ka. a** Eccentricity[71]. **b** $CO_2$ record (ppmv)[69]. **c** Reconstructed envelop for the ratio between global biosphere productivity and pre-industrial biosphere productivity as inferred from $\Delta^{17}O$ of $O_2$ at age t (interpolation to 200 years and 101 binomial smoothing with Igor software). **d** $\delta^{18}O_{atm}$ record from the EDC ice core[18,72-76]. **e** LR04 Sea level stack (m compared to present) on AICC2012 timescale[77]. **f** Trees, shrubs and Picea pollen (%)[16]; **g** Si/Ti ratio[16]; a proxy of biogenic silica normalised to detrital, reflecting the changes in diatom productivity in the lake. **h** Pollen abundance (grains.cm$^{-3}$) of ODP Site 646[26]. **i** Arboreal and Quercus robur pollen records (%) from Lake Ohrid, Balkan Peninsula[46,47]. **j** Alkenone mass accumulation rate (MAR) ($\mu g \cdot m^{-2} \cdot year^{-1}$) and TOC MAR ($mg \cdot m^{-2} \cdot year^{-1}$) records at Site PS2489-2/ODP1090[3]. **k** TOC/$CaCO_3$ ratio at Site PS2489-2/ODP1090, Atlantic Southern Ocean[14]. The horizontal dotted line separates the ice core records presented on the AICC2012 timescale[24,70] (above the line) and the terrestrial and oceanic records presented on the age model of each core (under the line). The grey shadow bars represent the period of rapid increase in atmospheric $CO_2$.

The minimum in $\delta^{18}O_{atm}$ actually occurs toward the end of MIS 11, i.e. around 405 ka. The anomalously high $\delta^{18}O_{atm}$ at the time of peak biosphere productivity inferred from $\Delta^{17}O$ of $O_2$ cannot be explained by a weak monsoon period in opposite to an explanation given for high $\delta^{18}O_{atm}$ levels observed over the last 800 ka[43]. Indeed, data from other archives rather show a relatively strong East Asia monsoon activity during MIS 11 similarly to what is observed over the last 4 interglacial periods[44,45]. Instead, this high $\delta^{18}O_{atm}$ values should be interpreted as an increase of the terrestrial vs oceanic productivity ratio expressed in flux of oxygen.

In most cases, terrestrial records are too short and not well dated enough to study the dynamic of vegetation and TOC over this termination. Moreover, hiatus of pollen data from oceanic records often prevent to study Termination V with high resolution. The best regional information on the terrestrial productivity

covering the last 445 ka can be recovered from palynological studies of sediments from El'gygytgyn[16], Ohrid[46,47], and Tenaghi Philippon[15] lakes, and from the oceanic ODP Site 646[26]. Pollen records can be used as indirect information on terrestrial biosphere productivity knowing the primary productivity associated with each ecosystem[48]. From this, the aforementioned records from the Northern Hemisphere highlight a major increase in terrestrial productivity during Termination V until mid-MIS 11 through the growth of plants associated with high productivity (Picea, Tree, Shrubs, Arboreal pollen, Quercus robur) in place of steppic species, tundra or frozen soils. This increase is the highest of the last 445 ka when focusing on high latitudes records (ODP site 646 at 58°N[26], Lake El'gygytgyn at 67°N)[16]. At lakes Ohrid[46,47] and Tenaghi Philippon[15], that are located around 40°N, while the increase in pollen abundance during Termination V is comparable to those observed during younger ones, the

pollen peak lasts longer (Supplementary Fig. 3). In parallel, the very high and long-lasting Si/Ti ratio and TOC percentage recorded in lake sediments[16,49] are the consequences of a strong and longer deglacial productivity[15,16,46,47] (Fig. 3). These regional indications of terrestrial productivity are in agreement with the increase in global productivity inferred from $\Delta^{17}O$ of $O_2$ and increase in the terrestrial vs oceanic productivity ratio inferred from $\delta^{18}O_{atm}$.

Several factors could explain the higher terrestrial productivity recorded during Termination V and MIS 11 compared to other younger terminations. First, the relatively slow sea level rise over Termination V[50] leaves some emerged vegetated terrestrial platforms (e.g. Sunda plate). Then, terrestrial biosphere productivity could be favoured in the context of low eccentricity leading to less pronounced seasonality and a longer duration of summer growing season[16]. Finally, MIS 11 biosphere productivity can be enhanced at high latitude of the northern Hemisphere because of the strong warming and melting of ice sheet volumes in Northern Hemisphere at mid-MIS 11 leading to more available lands[16,49], favouring plant development/productivity. Some also suggest a more humid and warm climate, with higher precipitation compared to MIS 1[16,49].

In the Ocean, little is known about deglacial oceanic productivity over the last 800 ka. Increased contents of organic matter (Total Organic Carbon (TOC) and biomarkers)[3,14] in sediments from the Subantarctic Zone of the Southern Ocean have been used to reflect increased downward flux of carbon associated with the photosynthetic biomass and thus, increased soft-tissue pump efficiency. However, they remain sparse and cannot directly be used to infer oceanic primary productivity. Besides, they do not consider the relative contribution of the downward flux of Particulate Inorganic Carbon (mainly $CaCO_3$) produced by calcifying plankton in the sunlit ocean (mainly coccolithophores and planktonic foraminifera), that creates a surface-to-deep alkalinity gradient, causing $CO_2$ to be released back to the atmosphere[51] and that represents the carbonate counter pump. While the biological carbon pump, including both the soft-tissue pump and the carbonate counter pump, is expected to impact the atmospheric $pCO_2$, the soft-tissue pump is the only one to affect $O_2$ fluxes and thus, the $\Delta^{17}O$ of $O_2$.

To the best of our knowledge, the best regional information on Southern Ocean productivity spanning the last 445 ka is obtained from site PS2489-2/ODP1090[3,14] located within the Atlantic sector of the Subantarctic Zone. This site provides TOC and alkenone mass accumulation rates (MARs) as well as marine $TOC/CaCO_3$ ratio that are helpful to discuss both soft-tissue pump and biological carbon pump efficiencies. Records show decreases in TOC and alkenone MARs, correlated with decreased $TOC/CaCO_3$ during Terminations V to I (Fig. 3). TOC and alkenone records during Termination V display no different trend compared to younger terminations, i.e. they show a less efficient soft-tissue pump probably associated with a decreased biological productivity in the Subantarctic Zone of the Southern Ocean, hence in agreement with the increase of terrestrial vs oceanic productivity over Termination V inferred from $\delta^{18}O_{atm}$.

The most striking pattern comes from the $TOC/CaCO_3$ ratio directly linked to biological pump efficiency and exhibiting a peak of very large amplitude over Termination V. This productivity signal goes along the exceptional production and accumulation of the coccolith *Gephyrocapsa caribbeanica* observed worldwide[28,52], and particularly within the Subantarctic Zone of the Southern Ocean[30,53] at the beginning of, and during Termination V. Such exceptional coccolithophore acme could be enhanced by the minimum of eccentricity during this period, causing a longer duration of summer growing season[52]. In parallel, in the low-latitude ocean, this period is marked by a strong increase in reef

carbonate accumulation, probably associated with the sea level rise documented at that time[29,54]. The singular carbonate production by coccolithophore and coral ecosystem is more difficult to interpret straightforwardly in term of increase in biological productivity but supports the exceptional biological context over Termination V. Indeed this worldwide increase in carbonate production and preservation is expected to considerably decrease ocean $[CO_3^{2-}]$[28,55], and thus, significantly increase the atmospheric $CO_2$ concentration[30,54] over Termination V. However, and opposite to the evolution of atmospheric $CO_2$ concentration during the other interglacial periods, the onset of MIS 11 is not associated with the highest $CO_2$ level of the whole interglacial period which means that a competing effect is at play. Some processes can play a role in the moderation of the $pCO_2$ such as physical pumping of $CO_2$ in the Southern Ocean through solubility effect or ventilation. A recent study provided indication on the change in the surface temperature of the Southern Ocean on the same timescale than atmospheric $CO_2$ variations over the last 720 ka with a link to ocean ventilation[56]. The changes in Southern Ocean temperature and ventilation are not different during Termination V than over the four youngest terminations, therefore showing no evidence of an enhanced physical pumping over this time period. We rather postulate that the competing effect for a moderate $CO_2$ level over the first part of MIS 11 is the exceptional productivity during Termination V leading to a fixation of excess atmospheric $CO_2$ in the biosphere. Moreover, the maximum in atmospheric $CO_2$ concentration is actually observed more than 10 ka after the onset of MIS 11, i.e. at the exact time when global biosphere productivity, as inferred from $\Delta^{17}O$ of $O_2$, is decreasing. We thus propose that the strong peak of terrestrial productivity over Termination V, mainly driven by terrestrial biosphere, might slow down the increase of atmospheric $CO_2$ concentration induced by the carbonate production.

Our new record of $\Delta^{17}O$ of $O_2$ over Termination V reveals a very different picture from the evolution observed over the last 4 terminations: $\Delta^{17}O$ of $O_2$ is not strongly anti-correlated with $CO_2$ which should be interpreted as an anomalous evolution of the biosphere productivity during Termination V and beginning of MIS 11. The global oxygen biosphere productivity is larger by about 10 to 30% at the beginning of MIS 11 compared to the productivity observed at the onset of the younger interglacial periods. Comparison with the $\delta^{18}O_{atm}$ record over the last 800 ka and the few available data on continent and ocean shows that this exceptional productivity increase is mainly due to an increase in terrestrial productivity. In parallel, the exceptional change in the carbonate cycle in oceanic environment also suggests that this peak in biosphere productivity may have contributed to maintain the atmospheric $CO_2$ concentration at relatively low level at the beginning of MIS 11 while strong carbonate production and export, would have the opposite tendency of increasing it. The exceptional strong biosphere productivity during this period can be linked to low eccentricity around 400 ka, probably influencing productivity in both the terrestrial and oceanic ecosystems, due to longer summer insolation duration. The slow sea level rise during Termination V followed by an exceptional warming at high latitude of the Northern hemisphere over mid-MIS 11 may also have favoured increasing terrestrial biosphere productivity through vegetation in emerged low-latitude platform during Termination V and vegetation on high latitude terrestrial regions usually characterised by frozen soils during MIS 11.

Finally, this important result on the global productivity calls for future studies, especially at the regional scale through measurements of productivity signals in oceanic and terrestrial reservoirs over Termination V. This study should also be extended to refine recent terminations (e.g. Termination III where $\Delta^{17}O$ of $O_2$ resolution is very low) and bring attention to older ones. The

comparisons of high-resolution records of $CO_2$ and $\Delta^{17}O$ of $O_2$ should permits to study the coupled dynamic of biosphere productivity and carbon cycle associated with large climate changes, under different orbital configurations.

## Method

**Material.** The ice samples used for this study come from bag samples allocated to gas consortium (A cut) of the EPICA Dome C 2 ice core. They were stored after drilling (2002-2003) in France (Le Fontanil) at $-20\,°C$ before being transported to LSCE, Gif sur Yvette in December 2017 for analysis in first semester of 2018. In total, 53 ice samples were analysed but 3 samples were lost due to an excess of pressure in the line.

**Extraction of air trapped in bubble from ice.** Extraction of air trapped in bubbles of ice was performed using a semi-automatic extraction and separation line. A 5-mm-thick section of ice was removed from each sample prior to measurement to prevent any contamination from ambient air or drilling fluid. The sample is then introduced in a frozen flask ($-20\,°C$) attached to the extraction line together with 5cc flasks of outside air (standard for isotopic composition of oxygen). For each daily sequence, two flasks of exterior air and three duplicated ice samples (i.e. 6 ice samples) were analysed. After pumping the line 40 min with samples kept at $-20\,°C$, the ice samples are melted to extract the air trapped in the bubble from the melted ice. The air then goes through $H_2O$ and $CO_2$ traps immersed in ethanol cooled at $-90\,°C$ and in liquid nitrogen ($-196\,°C$) respectively. The gas sample is transferred in a molecular sieve at $-196\,°C$, then in a chromatographic column (1 m*2 mm) to separate $O_2$ and Ar from $N_2$ and finally goes through another molecular sieve to purify $O_2$ and Ar from remaining He. The extracted air is then trapped in stainless-steel tubes in an 8 port collection manifold immerged in liquid helium (4.13 K). The method follow description from Barkan and Luz[57] except that the chromatographic column is longer (1 m instead of 20 cm) and kept at higher temperature ($0\,°C$ instead of $-90\,°C$).

**Mass spectrometer measurements.** Measurements of $^{18}O/^{16}O$, $^{17}O/^{16}O$ and $^{32}O_2/^{40}Ar$ ($\delta O_2/Ar$) were performed using a multi-collector mass spectrometer Thermo Scientific MAT253 run in dual inlet mode. For each extracted air sample, 3 runs of 24 dual inlet measurements were performed against a laboratory standard gas obtained by mixing commercial $O_2$ and commercial Ar in atmospheric concentration. The mean of the three runs were calculated for each sample and the standard mean deviation is calculated as the pooled standard deviation over all duplicate samples hence integrating the variability associated with extraction, separation and mass spectrometry analysis.

The resulting pooled standard deviations before the corrections are 0.05‰, 0.02‰, 6.4‰ and 6 per meg for $\delta^{18}O_{atm}$, $\delta^{17}O_{atm}$, $\delta O_2/Ar$ measurements and $\Delta^{17}O$ of $O_2$ calculation, respectively.

**Atmospheric air calibration.** Every day, $\delta^{18}O$, $\delta^{17}O$ and $\delta O_2/Ar$ of atmospheric air are measured and are then used to calibrate our measurements following:

$$\delta^{18}O_{ext\ air\ corr} = \left[ \frac{\left(\delta^{18}O_{sample}/1000\right)+1}{\left(\delta^{18}O_{ext\ air}/1000\right)+1} - 1 \right] * 1000, \quad (1)$$

$$\delta^{17}O_{ext\ air\ corr} = \left[ \frac{\left(\delta^{17}O_{sample}/1000\right)+1}{\left(\delta^{17}O_{ext\ air}/1000\right)+1} - 1 \right] * 1000. \quad (2)$$

The $\delta^{18}O_{ext\ air}$ and $\delta^{17}O_{ext\ air}$ were constant during the two measurement periods so that we used the average values over the two corresponding periods to correct the raw data. The daily correction was the same every day during each period.

**Correction due to fractionation in the firn column.** Several corrections were applied on the measured data due to fractionation in the firn. We describe here corrections due to gravitational and thermal fractionation as well as pore close-off effect.

Gravitational fractionation operates in firn due to Earth gravity field. $\delta^{18}O$ and $\delta^{17}O$ were obtained by corrections for this diffusive process using $\delta^{15}N$ in neighbouring samples[58]. The correction applied depends on the difference of mass between the two isotopes considered so that:

$$\delta^{18}O_{gravitational\ corr} = \delta^{18}O_{measured} - 2 * \delta^{15}N, \quad (3)$$

$$\delta^{17}O_{gravitational\ corr} = \delta^{17}O_{measured} - 1 * \delta^{15}N. \quad (4)$$

Diffusive processes operating in the firn column due to changes in temperature or because of the Earth gravity must also be taken into account. These processes lead to isotopic fractionation of $O_2$ that was taken into account for $\delta^{18}O_{atm}$ reconstruction in the NGRIP ice core over abrupt temperature changes of the last glacial period[59]. To check the effect of thermal fractionation on $\Delta^{17}O$ of $O_2$, we performed measurements of $\Delta^{17}O$ of $O_2$ over the top 17 m of the EastGRIP firn

were a strong seasonal gradient is present. However, the resulting $\Delta^{17}O$ of $O_2$ was not showing any significant deviation from the atmospheric value hence suggesting that thermal fractionation does not modify the $\Delta^{17}O$ of $O_2$ of the atmosphere in the firn column (Supplementary Table 3). Moreover, in Antarctica, surface temperature variations are much lower than in Greenland during deglaciations or climatic variability of the last glacial period so that thermal fractionation is not expected to have a significant effect on the isotopic composition of trapped oxygen.

Pore close-off at the bottom of the firn has been shown to affect $\delta O_2/N_2$, $\delta Ar/N_2$ with potential effects on $\delta^{15}N$ and $\delta^{40}Ar$ in certain cases[60]. We checked this possible effect on $\Delta^{17}O$ of $O_2$ by comparing $\Delta^{17}O$ of $O_2$ in bubbly ice at the top of the NEEM ice core. After correction of gravitational effect, we found a systematic enrichment of 13 per meg which could potentially bias the reconstruction of atmospheric $\Delta^{17}O$ of $O_2$ from $\Delta^{17}O$ of $O_2$ in trapped air.

**Gas loss correction.** During storage, $O_2$ in ice samples is subject to gas loss fractionation due to diffusion processes[61] and $O_2/N_2$ ratio is always lower by several % in ice samples stored several years at $-20\,°C$ than ice samples stored at $-50\,°C$[62]. Such gas loss effect is also associated with isotopic fractionation of oxygen, $\delta^{18}O_{atm}$ trapped in the ice being higher when $\delta O_2/N_2$ decreases with a slope for the relationship of $-0.01$ ($\delta^{18}O_{atm}$ vs $\delta O_2/N_2$)[17,18,63,64]. We thus expect that $\delta^{17}O$ can also be affected by this gas loss process and that it may create an anomaly of $\Delta^{17}O$ of $O_2$. To check this effect, measurements of $\delta^{17}O$, $\delta^{18}O_{atm}$, $\Delta^{17}O$ of $O_2$ and $\delta O_2/Ar$ have been performed on three samples of GRIP ice core (clathrate ice stored during more than 20 years at $-20\,°C$). Each of the three ice samples have been cut in order to analyse the interior and the exterior of the sample separately. $\delta O_2/N_2$ measurements could not be performed on exactly the same samples so that we used the $\delta O_2/Ar$ measurements to estimate the amount of oxygen loss. Argon is also known to be affected by gas loss[17,61] but in a smaller extend than oxygen[65] so that the decrease of $\delta O_2/Ar$ is still a good indication of larger gas loss.

Data show a systematic lower $\Delta^{17}O$ of $O_2$ value in the exterior sample compared to the interior sample, paralleling the lower $\delta O_2/Ar$ value in the exterior sample compared to the interior sample as expected by gas loss (Supplementary Table 1). This systematic relationship and the $\Delta^{17}O$ of $O_2$ and $\delta O_2/Ar$ values can be used to propose a correction for the $\Delta^{17}O$ of $O_2$ that takes into account the gas loss effect:

$$\Delta^{17}O_{gas\ loss\ corr} = \Delta^{17}O_{sample} - 0.3945 * \left[ (\delta O_2/Ar)_{sample} - (\delta O_2/Ar)_{std} \right]. \quad (5)$$

**Comparison of EDC $\Delta^{17}O$ of $O_2$ with previous $\Delta^{17}O$ of $O_2$ record.** Previous $\Delta^{17}O$ of $O_2$ measurements only took into account correction linked with gravitational fractionation. As a consequence, we checked the consistency of previous records performed on the GISP2 and Vostok ice core with our new EDC data by measuring again $\Delta^{17}O$ of $O_2$ over Termination II using EDC samples and integrating the aforementioned correction (Fig. 1).

**Flux of oxygen associated with biosphere productivity.** We follow here the calculation of the flux of oxygen associated with biosphere productivity described in Landais et al.[33].

Since the atmospheric $\Delta^{17}O$ of $O_2$ ($\Delta^{17}O_{atm}$) is influenced by the exchanges with biosphere and stratosphere, it is possible to write the following equation in a 3-box system at equilibrium (biosphere—bio-, atmosphere—atm-, stratosphere—strat-):

$$F_{bio} * \left( \Delta^{17}O_{bio} - \Delta^{17}O_{atm} \right) = F_{start} * \left( \Delta^{17}O_{start} - \Delta^{17}O_{atm} \right), \quad (6)$$

where $F_{bio}$ is the flux of oxygen exchanged by photosynthesis and respiration (both fluxes being considered at equilibrium) and $F_{strat}$ is the flux of oxygen exchanged between the lower atmosphere (or troposphere) and the stratosphere (fluxes in and out of the stratosphere are assumed equal).

To reconstruct past biospheric fluxes between terrestrial/oceanic biosphere and atmosphere based on $\Delta^{17}O$ of $O_2$, it is necessary to know the evolution of the stratospheric flux as well as of the $\Delta^{17}O_{strat}$. Luz et al.[20] and Blunier et al.[22] showed that a good assumption is to consider that the production rate of depleted $O_2$ in the stratosphere can be related to the atmospheric $CO_2$ concentration. It is then possible to express the evolution of biosphere oxygen flux in the past compared to pre-industrial from the following equation:

$$\frac{F_{bio,t}}{F_{bio,pre-industrial}} = \frac{(CO_2)_t}{(CO_2)_{pre-industrial}} * \frac{\Delta^{17}O_{bio,pre-industrial}}{\Delta^{17}O_{bio,t} - \Delta^{17}O_{atm,t}}, \quad (7)$$

where $F_{bio,t}$ is the biosphere oxygen flux at a given time, $F_{bio,pre-industrial}$ is the present time biosphere oxygen flux, $(CO_2)_t$ and $(CO_2)_{pre-industrial}$ correspond to the atmospheric $CO_2$ concentration at a given time and at the pre-industrial period respectively, $\Delta^{17}O_{bio,pre-industrial}$ and $\Delta^{17}O_{bio,t}$ are the values of $\Delta^{17}O$ of $O_2$ in an atmosphere that would only be influenced by exchanges with the biosphere at the present time and at a given time respectively. $\Delta^{17}O_{atm,\ t}$ is the $\Delta^{17}O$ of $O_2$ of the atmosphere at a given time measured in the air trapped in ice core. We refer to pre-industrial period because of the long residence time of oxygen in the atmosphere

(>1000 years), $\Delta^{17}O$ of $O_2$ is not exhibiting any significant variation over the last centuries.

$\Delta^{17}O_{bio}$ can be calculated from the fractionation coefficients associated with the different processes leading to oxygen fluxes in the biosphere (mainly photosynthesis, dark respiration and photorespiration). Detailed calculations of $\Delta^{17}O_{bio}$ at the LGM and pre-industrial periods were obtained in Landais et al.[33] taking into account uncertainties in the determination of the fractionation coefficients[34] as well as on the relative fluxes of oxygen (Supplementary Table 2).

From the calculation of $\Delta^{17}O_{bio, pre-industrial}$ and $\Delta^{17}O_{bio,LGM}$ (Supplementary Table 2), $\Delta^{17}O_{bio,t}$ is calculated through a scaling on the variations of $CO_2$ concentration between pre-industrial period (280 ppmv) and the LGM (190 ppmv)[66] such as

$$\Delta^{17}O_{bio,t} = \Delta^{17}O_{bio,pre-industrial} + \left(\Delta^{17}O_{bio,LGM} - \Delta^{17}O_{bio,pre-industrial}\right) * \left(\frac{280 - (CO_2)_t}{90}\right).$$

(8)

**Details on uncertainty in photosynthesis fractionation.** We performed sensitivity tests to compare biosphere productivity reconstructions without any fractionation and reconstruction with the different fractionation effects during marine photosynthesis as observed in Eisenstadt et al.[37]. The largest change on the reconstructed biosphere productivity is obtained using the observed photosynthesis fractionation effect associated with *Emiliania huxleyi* ($\delta^{18}O = 5.81‰$ and slope between $\ln(\delta^{17}O + 1)$ and $\ln(\delta^{18}O + 1)$ of 0.5253). The reconstructed global biosphere productivity is lower with such fractionation effect: the MIS 11 productivity level is decreased by 3% with such fractionation effect compared to the reconstruction with no fractionation at photosynthesis (i.e. the average situation, see Supplementary Table 2).

**Details on uncertainty in respiration fractionation.** When taking into account the maximum effect, i.e. a decrease of 0.005 for the slope of the relationship between $\ln(\delta^{17}O + 1)$ and $\ln(\delta^{18}O + 1)$ during respiration, we end up with a decrease of $\Delta^{17}O_{bio}$ by 65 ppm. The resulting biosphere productivity reconstruction is about 16% higher during MIS 11 than the average situation (Supplementary Table 2) leading to an anomalously high $O_2$ flux associated with gross primary productivity during Termination V.

**Validation of the biosphere productivity reconstruction.** The three sensitivity tests displayed (uncertainty on photosynthesis fractionation, on respiration fractionation and on fractionation within the water cycle) suggest that the biosphere productivity during MIS 11 is significantly higher than for our current interglacial, leading to the envelop displayed on Supplementary Fig. 2. Actually, explaining the $\Delta^{17}O$ of $O_2$ anomaly without invoking a significant increase of the global productivity at the beginning of MIS 11 requires huge change of the $\Delta^{17}O$ of $O_2$ produced by terrestrial biosphere (through changes in fractionation factor or changes in water cycle, hypothesis 1) or of the ratio of oceanic to terrestrial biosphere productivity (hypothesis 2). These two possibilities are rather unrealistic as detailed below.

In hypothesis 1, an increase of 30 ppm for $\Delta^{17}O$ of $O_2$ produced by terrestrial biosphere is needed. This can be obtained either by a huge change in the fractionation factors between our current interglacial and MIS 11 (which is not realistic since these are based on physical processes which do not vary with time) or by a change of water cycle during MIS 11 with respect to present-day value. To reach this 30 ppm increase through modification of the water cycle, one option is to increase the $^{17}O$-excess of meteoric water by 30 ppm during MIS 11 compared to our current interglacial. We do not have $^{17}O$-excess values for MIS 11 yet, but ice core $^{17}O$-excess values obtained for the last interglacial are very similar to values obtained for the present interglacial[39]. Moreover, increasing $^{17}O$-excess by 30 ppm would require unrealistic decrease of relative humidity at evaporation (by 30%, Barkan and Luz[67]) during MIS 11 compared to our present interglacial. Finally, an increase of the mean $\delta^{18}O$ and $\delta^{17}O$ of meteoric water used by the plant along the meteoric water line (i.e. without global change in the global $^{17}O$-excess) would as well be a solution (see Fig. 4 of Landais et al.[33]) but to reach the expected increase of 30 ppm in $\Delta^{17}O$ of $O_2$ produced by the terrestrial biosphere, it would require an increase of global mean $\delta^{18}O$ of meteoric water by ~3 ‰ with respect to present-day that will be transmitted to the global $\delta^{18}O_{atm}$. The $\delta^{18}O_{atm}$ during MIS 11 is between 0 and 0.5 ‰ higher than during MIS 1 so that it does not support this hypothesis.

Hypothesis 2 requires an increase of the oceanic vs terrestrial productivity by a factor of 2 during MIS 11 compared to present-day value. This does not go along available observations suggesting an increase in the ratio of terrestrial to oceanic productivity during MIS 11 (see main text) and is still well above the uncertainty for the variation of the ratio of oceanic to terrestrial productivity during our interglacial period (14%, Supplementary Table 2).

An additional validation of our reconstruction of exceptional productivity during the beginning of MIS 11 comes from the comparison of our reconstruction with the oxygen biosphere productivity variations obtained by Blunier et al.[22] over the last four terminations using a different vegetation model and different formulation of the link between atmospheric $CO_2$ and flux of $\Delta^{17}O$ of $O_2$ anomaly from the stratosphere. Indeed, in our approach, we directly dealt with $\Delta^{17}O$ of $O_2$ anomaly in the box model while it may be more appropriate to deal with the $\delta^{17}O$ and $\delta^{18}O$ of $O_2$ values as done in Blunier et al.[22] and suggested by Prokopenko et al.[68]. We thus compared our reconstruction with the one performed with the model of Blunier et al.[22] (Supplementary Fig. 2). The smaller difference in the oxygen flux associated with oxygen productivity between glacial and interglacial periods in Blunier et al.[22] mainly comes from the different formulations of the dependency of the stratospheric anomaly $\Delta^{17}O$ of $O_2$ to $CO_2$ atmospheric concentration. Another difference comes from the variations in the triple isotopic composition of oxygen in water between glacial and interglacial periods not taken into account in Landais et al.[33]. We took this possible variation into account in our approach through the third sensitivity test explained above. We ran here the model of Blunier et al.[22] over the new Termination V data. The comparison between the two methods of reconstruction (Landais et al.[33] vs Blunier et al.[22]) is significant. The biosphere productivity increase at the end of Termination V is less important using the Blunier et al.[22] approach than using the Landais et al.[33]: 10% larger than for other interglacial periods following Blunier et al.[22] instead of 20% on average following Landais et al.[33]. Still, both approaches show that the biosphere productivity increase over Termination V is exceptional compared to the younger terminations.

## Data availability

The corrected data of $\Delta^{17}O$ of $O_2$ are available in Pangaea database https://doi.pangaea.de/10.1594/PANGAEA.914609. The data are also available from the authors.

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

## Acknowledgements

The authors first acknowledge the three anonymous reviewers and the editor for their useful help in improving the manuscript. It is a pleasure to thank Jean Jouzel, Jochen Schmitt, Ji-Woong Yang and Michael Bender for useful discussions as well as Laurent Bopp, Masa Kageyama, Chiara Guarnieri and Jean-Yves Peterschmitt for their contribution on model reconstructions. We also thank Gregory Teste involved in cutting EDC samples. This work benefited from funding from the ANR program HUMI17 as well as the INSU-LEFE-IMAGO program BIOCOD. M.B. receives an IDEX-IDI PhD grant from Paris-Saclay. It is a contribution to the European Project for Ice Coring in Antarctica (EPICA). This work was supported by the French state aid managed by the ANR under the "Investissements d'avenir" programme with the reference ANR-11-IDEX-0004-17-EURE-0006.

## Author contributions

A.L., S.D.A. and M.B. designed this study. M.B, F.P., V.F, H.A. and L.S performed the $\Delta^{17}O$ of $O_2$ measurements. M.B and A.L. corrected the data and calculated the flux of oxygen biosphere productivity with the help of T.B. M.B., A.L. and S.D.A. wrote the manuscript with the contribution of T.B and T.E.

## Competing interests

The authors declare no competing interests.
