## [Peer Review File · Nature Communications]

Reviewers' comments:

Reviewer #1 (Remarks to the Author):

The paper entitled "Exceptionally high biosphere productivity at the beginning of Marine Isotope Stage 11," by Brandon, Landais, Duchamp-Alphonse, Favre, Schmitz, Abrial, Prie, Extier, and Blunier provided quite new evidence of isotopic composition of atmospheric oxygen trapped in the Antarctic ice sheet, dated around 400ka when any studies did not reach ever, then discussed temporal variation of the global ecological productivity as well as its controlling factors. Moreover, they proposed new data corrections with regard to isotopic fractionations for gas-ice interactions. As a result, their efforts did not make a significant difference from the previous study, however, it must be valuable in the sense that they reduce the ambiguity.

Their results are original and significant, they use appropriate methodology and statistics including the treatment of uncertainties.

In the discussion part, however, it is not clear very much. I felt some confusion in the context throughout the discussion.

If I were not misunderstanding, the essence of this study based on their new findings is that 1) exceptionally high biological productivity relative to later termination periods, and 2) this is basically due to terrestrial ecosystem, that is, higher productivity ratio of terrestrial to marine ecosystems. In addition to them, the peak height of CO₂ was equivalent to other terminations but its broader shape was different from D170 of O₂, which were consistently sharp for other periods.

First, the authors show results of palynological studies from lake sediments, and sedimentation rates of organic matters and CaCO₃ from sediments of the Southern Ocean, for demonstrating high productivities in land and ocean, respectively. As shown in Fig. 3 or supplementary Fig. 4, however, it is hard to read evidence for higher terrestrial productivity than other periods. Although each peak appears to last longer than others, its value may be approximately equal to them. Similarly, I could not find any higher marine productivity from the variation of TOC and alkenone MARS. As for alkenone, termination IV indicates rather higher peak value. Supporting evidence is thus higher carbonate productivity in the ocean only. This is consistent with the result in the Indian Ocean by Rickaby et al. (2007). Additionally, as the authors refer, the rapid development of coral reefs in this period may imply the high marine productivity in terms of not only CaCO₃ but also O₂. Both coccolithophore and corals generate photosynthetic oxygen (by symbiotic algae for corals) with co-occurring CaCO₃ formation. As a result, it seems only readable from evidence demonstrated by the authors that there was relatively high marine productivity in terms both of CaCO₃ and O₂.

If the atmospheric d18O during the termination V were evident, it is easily assumable that the terrestrial ecosystem sustained relatively higher productivity than the marine ecosystem. Therefore, I do not deny their conclusion. However, the authors must show other appropriate evidence for terrestrial productivity, OR revise their discussion in a major way.

Second, according to the study by Schilt et al. (2010), the long-lasting peak of CO₂ during MIS11 seems to co-occur with similar peak shapes of methane and nitrous oxide. Therefore, their hypothesis that the enhanced terrestrial productivity during the termination V might reduce or disappear the CO₂ peak which should be there by the carbonate formation may be required re-consideration. Moreover, Uemura et al. (2018) indicated that there was a strong positive relationship between atmospheric CO₂ concentration and the temperature over the Southern Ocean, as well as the positive relationship between CO₂ and the strength of deepwater circulation. For justification, authors should refer their studies properly.

The last, the periodicities of precession and obliquity are illustrated in Figs 1 and 3 as well as supplementary Figs 3 and 4, although I could not find any description for them in the text. I found a description for eccentricity (L270) but not illustrated. Why and why not?

Specific comments:

Terms "marine" and "oceanic" are mixed. Please choose one for clarity.
Terms "continental" and "terrestrial" are mixed. Please choose one for clarity.
L42: "than" instead of "over"
L60: "BCP" did not appear afterwards. Not necessary.
L71: "as same as" instead of "as for"
L78: "their temporal variation" instead of "their variation"
L81-84: References must be needed.
L85-86: One or more reference must be needed.
L93-95: References must be needed.
L94: What is "milder to hotter warm"?
L97-98: higher sea level than the present
L96-100: I suggest "Termination V than the present, whereas the CO2 level.....warm periods. The CO2 fluxes....."
L103-104: What is "their patterns and their specific relationships"? Please specify.
L109: What kinds of fluxes?
L111-112: Are "local" and "global" in conflicting structures? In the following text, they seem to complement each other.
L120: "obtained by corrections for" instead of "corrected from"
L153-155: Hard to understand. Does the age mean the right shoulder of broad peak (plateau) of CO2?
L155-157: Please apply statistical tests if you mention the data robustness.
L159: Delete "or possible"
L160: "always" instead of "systematically"
L174: "rate-limited" instead of "limited" for clarity
L188: I suggest "possibility" instead of "propositions"
L221-223: This sentence may let readers confused because authors give freedom to the ratio soon after.
L236: "during other terminations" instead of "over the terminations"
L247: Values are neither strong or weak, but either high or low.
L269-270: As forementioned, "eccentricity" appears suddenly without illustration.
L285: I disagree with this. Termination IV looks larger.

Fig.1: The upper panel is not treated in the text.

Fig.3: I suggest authors to combine with supplementary Fig.4 and replace it.

Fig.3: The upper panel is not treated in the text.

Fig.3: Shaded zones slightly mismatch the text. L149-L155, for example.

Fig.3: I suggest to add up and down arrows for qualitative controls on the variation of $\delta^{18}O_{atm}$.

Supplementary Fig.2: Identical to the middle panel of Fig. 1. It seems not necessary.

Supplementary Fig.3: The upper panel is not treated in the text.

Supplementary Fig.4: The upper panel is not treated in the text.

Supplementary Fig.4: Suggesting to be combined with Fig.3.

Suggested references

Rickaby R. E. M. et al. (2007) *Earth Planet. Sci. Lett.* 253, 83–95 (2007)

Uemura R. et al. (2018) *Nature Communications*, DOI: 10.1038/s41467-018-03328-3

Reviewer #2 (Remarks to the Author):

Brandon and coworkers present new oxygen triple-isotope data in O_2 across Termination V that show a similar, but perhaps slightly larger peak-to-peak range in $\delta^{17}O-O_2$ as Terminations I, II, and IV (TIII is smaller). Importantly, however, this $\delta^{17}O$ change shows a different relationship with coeval atmospheric CO_2 concentrations than in the other glacial terminations: CO_2 seems to rise more rapidly across Termination V, relative to $\delta^{17}O-O_2$. Again, there is a bit of variability in this relationship, but it appears demonstrably different over Termination V. The authors use this asynchrony and the model of Landais et al. 2007 to infer an excess in biosphere productivity during the initial parts of the subsequent interglacial.

The data here are likely good, but I think the model (which serves as the basis for the productivity

claim) is flawed, and its potential shortcomings are not mentioned in the text or supplement.

Methodologically, I think the model's main flaw is its treatment of $\Delta 17\text{O}$ as a conserved quantity upon mixing; it is not, as many have shown in the literature (Miller et al. 2002, Prokopenko et al., 2011, and Kaiser et al., 2011 to name a few). The net result is an inaccurate calculation of the mixing mass balance between biological and stratospheric endmembers. It therefore predicts larger productivity swings between glacial and interglacial climates than a model that uses a more accurate mixing formulation (Blunier et al. 2012). One can see this in Supplementary Figure 3. Blunier et al.'s model would probably yield muted changes in global productivity, much more similar to the PST scenario than the authors claim in the abstract.

The Blunier glacial-interglacial productivity swings are much more in line with other estimates such as from $\delta 18\text{O}_{\text{atm}}$ and $\delta 13\text{C-CO}_2$ by Ciais et al. 2012. While one could conceivably take issue with how those authors' interpretations, these important caveats are not mentioned in the manuscript at all. It seems appropriate to at least interpret the data using both models and acknowledge potential model biases. If the interpretation is robust with respect to at least these two models (there is at least one more, from Young et al. 2014), then the argument would be more convincing.

On this basis, unfortunately, I am not sure the titular claim made by the authors is supported by the model. Productivity might still end up higher than preindustrial when using others' models, but I suspect it will not be "exceptionally high."

Reviewer #3 (Remarks to the Author):

This paper deals with the triple O isotope composition of O_2 ($^{16}\text{O}_2/^{17}\text{O}^{16}\text{O}/^{18}\text{O}^{16}\text{O}$) in O_2 trapped in ice cores. The basic idea is that the relation between $\delta 17\text{O}$ and $\delta 18\text{O}$ is dependent on photochemical reactions involving CO_2 in the stratosphere, and the turnover of O_2 by respiration and photosynthesis of the biosphere. The focus of the paper is on Glacial Termination 5, which occurred about 420 thousand years ago.

The paper reports new data for the oxygen isotope property of merit, $\Delta 17\text{O}$, for terminations 2 and 5. The Termination 5 result is quite interesting because the deglacial response between CO_2 and $\Delta 17\text{O}$ occur at quite different times (separated by 10 kyr or more). In contrast, one generally finds that these properties are tightly linked in time. The separation might be due to changes in the hydrologic cycle, a decrease in ocean productivity, and increase in land productivity, and changes in the distribution of productivity in the land and oceans. The authors argue that the unusual relationship between CO_2 and $\Delta 17\text{O}$ reflects higher land productivity during the termination. They support their conclusion by regional studies showing higher productivity around the time of interest.

My problem with this paper is that there are multiple factors influencing $\Delta 17\text{O}$ of O_2 . Many of these are discussed in the paper. The problem is that the authors do not have a good way of quantifying the magnitude of the changes induced by the various influences. In particular, fractionations in the hydrologic cycle can be large and are not well known. The isotope effects associated with photosynthesis and respiration remain, to my mind, poorly characterized. For example, work of Luz raises the possibility of an inverse (positive) isotope effect associated with photosynthesis, and recent work of Stolper raises questions about respiratory isotope fractionation factors. Uncertainties in hydrologic fractionation and fractionation associated with respiration and photosynthesis are first order issues limiting the interpretation. Overall, one has a single constraint (perhaps 2 adding $\delta 18\text{O}$ of O_2 , but it is unclear how useful this property is), but there are many more than 2 factors governing $\Delta 17\text{O}$ of paleoatmospheric O_2 .

I think this paper could be modified by adding a rigorous discussion of challenges associated with reconstructing productivity from $\Delta 17\text{O}$, and published in a geochemical journal. I do not recommend it for Nature Communications, because I think the interpretation is incomplete, and because I think the paper lacks robust climate insights that might be of interest to a more general audience.

We found the comments of the three reviewers very useful to improve the manuscript toward two main directions. First, we added a new model for biosphere productivity reconstruction from $\Delta^{17}\text{O}$ of O_2 as well as new sensitivity tests. It results in enlarged error bars while the same conclusion is unchanged. Second, we have rewritten the introduction and discussion to better discuss the specificity of the time period of interest, add new data related to continental and oceanic productivity and propose some explanations for an exceptional productivity during Termination V and MIS 11. All comments raised by the 3 reviewers have been taken into account and the details are explained in the point by point discussion below.

At the end of the point by point answer, we also provide the new manuscript and SOM with major changes highlighted in yellow.

Reviewer #1 (Remarks to the Author):

The paper entitled "Exceptionally high biosphere productivity at the beginning of Marine Isotope Stage 11," by Brandon, Landais, Duchamp-Alphonse, Favre, Schmitz, Abrial, Prie, Extier, and Blunier provided quite new evidence of isotopic composition of atmospheric oxygen trapped in the Antarctic ice sheet, dated around 400ka when any studies did not reach ever, then discussed temporal variation of the global ecological productivity as well as its controlling factors. Moreover, they proposed new data corrections with regard to isotopic fractionations for gas-ice interactions. As a result, their efforts did not make a significant difference from the previous study, however, it must be valuable in the sense that they reduce the ambiguity. Their results are original and significant, they use appropriate methodology and statistics including the treatment of uncertainties.

In the discussion part, however, it is not clear very much. I felt some confusion in the context throughout the discussion.

>>We thank Reviewer #1 for his/her constructive comments here, that enabled us to make the discussion clearer in the revised version of the paper.

If I were not misunderstanding, the essence of this study based on their new findings is that 1) exceptionally high biological productivity relative to later termination periods, and 2) this is basically due to terrestrial ecosystem, that is, higher productivity ratio of terrestrial to marine ecosystems. In addition to them, the peak height of CO_2 was equivalent to other terminations but its broader shape was different from $\Delta^{17}\text{O}$ of O_2 , which were consistently sharp for other periods.

>> Indeed, the main result from our study is a reconstruction of biosphere productivity during Termination V and MIS 11: it is $20\pm 10\%$ higher compared to the biosphere productivity for the four more recent terminations and interglacial periods. This reconstruction is directly deduced from the significantly slower $\Delta^{17}\text{O}$ of O_2 decrease over Termination V compared to the younger terminations (as illustrated by supplementary figure 1). This higher biosphere productivity can be related to the high terrestrial productivity observed during this period based on pollen, TOC and now Si/Ti records (see Figure 3) in the Northern high latitudes. The increase of atmospheric $\delta^{18}\text{O}$ of O_2 ($\delta^{18}\text{O}_{\text{atm}}$) during Termination V also supports this interpretation.

The text has been changed to take into account this comment in the introduction (lines 126-135) and discussion (lines 292 to 304) sections:

“The biosphere productivity over Termination V and the beginning of MIS 11 is found to be 10-30% higher than productivity over the pre-industrial period, an exceptional value never encountered over the last 4 interglacial periods. This productivity peak is most probably due to an increase of the terrestrial productivity during this period favored by a particular context of low eccentricity leading to long growing summer season, warm temperature at high latitudes of the northern hemisphere enabling biosphere productivity from continental soils usually frozen and possibly a relatively slow sea level rise during Termination V enabling productivity from low latitude emerged continental platforms. We propose that such strong productivity occurring concomitantly with an exceptional productivity carbonate peak in the marine realm plays a role in maintaining the CO₂ level at a relatively low level at the beginning of MIS 11.

“From this, the aforementioned records from the Northern Hemisphere highlight a major increase in terrestrial productivity during Termination V until mid-MIS 11 through the growth of plants associated with high productivity (Picea, Tree, Shrubs, Arboreal pollen, Quercus robur) in place of steppic species, tundra or frozen soils. This increase is the highest of the last 445 ka when focusing on high latitudes records (ODP site 646 at 58°N²⁹, Lake El'gygytgyn at 67°N)¹⁹. At lakes Ohrid^{53,54} and Tenaghi Philippon¹⁸, that are located around 40°N, while the increase in pollen abundance during Termination V is comparable to those observed during younger ones, the pollen peak lasts longer (Supplementary Figure 3). In parallel, the very high and long-lasting Si/Ti ratio and TOC accumulation recorded in lake sediments^{19,59} are the consequences of a strong and longer deglacial productivity^{18,19,53,54} (Figure 3). These regional indications of terrestrial productivity are in agreement with the increase in global productivity inferred from $\Delta^{17}\text{O}$ of O₂ and increase in the terrestrial vs oceanic productivity ratio inferred from $\delta^{18}\text{O}_{\text{atm}}$.”

First, the authors show results of palynological studies from lake sediments, and sedimentation rates of organic matters and CaCO₃ from sediments of the Southern Ocean, for demonstrating high productivities in land and ocean, respectively. As shown in Fig. 3 or supplementary Fig. 3, however, it is hard to read evidence for higher terrestrial productivity than other periods. Although each peak appears to last longer than others, its value may be approximately equal to them.

>> We agree that it is difficult to find evidence for higher biosphere productivity from regional records. This is first due to the fact that there is no direct regional tracer of biological productivity comparable to what is measured with the $\Delta^{17}\text{O}$ of O₂. $\Delta^{17}\text{O}$ of O₂ gives access to a gross primary productivity flux of oxygen while regional records give merely access to carbon stock and biome coverage through (1) total organic carbon (TOC) measurements and pollen counting on continent, and (2) export of phytoplankton biomass (i.e. the soft-tissue pump) using a combination of sedimentary TOC, and alkenones contents in the ocean. They may be large difference between these different key parameters of the carbon cycle as already noted by Ciais et al. (2012) for the last deglaciation.

In the main text and especially in the new introduction, we have strongly rewritten the text to better introduce the different terms as well as the difficulty to have alternative estimates of the biosphere productivity (lines 45 to 86).

“In parallel, terrestrial primary productivity and carbon stock increase during deglaciations, thus acting as a significant CO₂ land sink^{8,9} so that an important additional source

of CO₂ is required to explain the entire deglacial CO₂ pattern. The missing component has been suggested to be the important release of inert carbon from a thawing permafrost^{8,10}.

Quantifying changes in the carbon cycle over deglaciations relies on data compilation and modelling studies to estimate both carbon stocks and carbon fluxes⁸. Some information on carbon stocks can be obtained from $\delta^{13}\text{C}$ of carbonates using a wealth of data obtained from marine sediments¹¹ and $\delta^{13}\text{C}$ of atmospheric CO₂^{12,13}. In parallel, information on the evolution of past global productivity, a major component of carbon flux, is very sparse and often limited to the last deglaciation^{8,9}.

In the ocean, little is known about deglacial oceanic productivity over the last 800 ka. Increased contents of organic matter (Total Organic Carbon (TOC) and biomarkers)^{3,14} in sediments, have been used to reflect increased downward flux of carbon associated with the photosynthetic biomass. However, while such records provide crucial information about the efficiency of the soft-tissue pump, they remain sparse and cannot reliably be used to infer oceanic primary productivity. Besides, they do not consider the relative contribution of the downward flux of Particulate Inorganic Carbon (mainly CaCO₃) produced by calcifying plankton in the sunlit ocean (mainly coccolithophores and planktonic foraminifera), that creates a surface-to-deep alkalinity gradient, causing CO₂ to be released back to the atmosphere¹⁵ and that represents the carbonate counter pump. Past changes in biological carbon pump are best represented by changes in the buried TOC/CaCO₃ ratio, suggested to reflect C-rain ratio¹⁶. However, despite its accuracy to provide biological export production, only one record of sedimentary TOC/CaCO₃ exists in the Southern Ocean, for the last 800 ka, so far¹⁴. This record is however very precious since Southern Ocean biosphere productivity has a strong potential for increases in the past while on lower latitudes, the biosphere productivity is already maximum today¹⁷.

On continents, variations in vegetation cover and type can be related, although indirectly, to variations of the biosphere terrestrial productivity through the use of dynamic vegetation models modelling both biome and associated productivity⁹. Pollen counting and sedimentary TOC are thus useful for biosphere productivity reconstruction^{18,19} but they are unfortunately indirect and rely on the use of biosphere models. Moreover, similarly to oceanic records, these observations provide regional records that are not easy to use for documenting the past global carbon cycle. Ciais et al. (2012)⁸ proposed to use the isotopic composition of oxygen of atmosphere ($\delta^{18}\text{O}_{\text{atm}}$) as a tracer for terrestrial biosphere productivity. However, this proxy is a complex tracer being influenced by hydrological cycle at first order^{20,21} and its use as a quantitative tool for productivity reconstruction depends on the exact determination of associated fractionation factors in the water and biosphere cycle²².

A total estimate of the global biospheric fluxes and their temporal variations can be obtained more directly from measurements of $\Delta^{17}\text{O}$ of O₂ ($\ln(\delta^{17}\text{O}+1)-0.516*\ln(\delta^{18}\text{O}+1)$) in ice cores^{23,24}. This method provides O₂ fluxes and the conversion from O₂ to CO₂ fluxes can be done from the stoichiometry of the biological processes of photosynthesis and respiration²⁵.”

Still, we expect some correlation between vegetation coverage and gross productivity, and this is the reason why, in the revised text and figures, we added new data from Termination V compared to earlier terminations: the tree shrubs and picea records as well as the Si/Ti signal of lake El'Gygytgyn (67°N) from Melles et al., (2012) were added (Fig. 3). Pollen records show

the presence of *Picea* only during MIS 11 and a high and long peak of vegetation along MIS 11, revealing the retreat of the ice sheet at high latitudes and the strongest primary productivity recorded over the last 450 ka. Highest Si/Ti values reflect higher diatom productivity in this lake during MIS 11 compared to younger interglacials, thus confirming the overall productivity pattern in the area.

Furthermore, we added the pollen reconstruction of ODP site 646 (58°N) from de Vernal and Hillaire-Marcel (2008), showing the highest peak of pollen during MIS 11 compared to latest interglacial periods (Figure 3). The uncertainties in the age model of this site limit our interpretation of the timing between this record and the record of biosphere productivity but it still shows a very high terrestrial productivity in Northern Atlantic during MIS 11 which is in agreement with evidence for Greenland ice sheet melting during this period (Reyes et al., 2014). These records (Melles et al., (2012) and de Vernal et Hillaire-Marcel, (2008)) support the interpretation of a higher terrestrial productivity during MIS 11 compared to younger interglacial periods.

The discussion section has been modified to highlight the strong terrestrial productivity at the high latitudes of the Northern Hemisphere during MIS 11 (lines 287 to 304):

“The best regional information on the terrestrial productivity covering the last 445 ka can be recovered from palynological studies of sediments from El’gygytgyn¹⁹, Ohrid^{53,54}, and Tenaghi Philippon¹⁸ lakes, and from the oceanic ODP Site 646²⁹. Pollen records can be used as indirect information on terrestrial biosphere productivity knowing the primary productivity associated with each ecosystem⁵⁸. From this, the aforementioned records from the Northern Hemisphere highlight a major increase in terrestrial productivity during Termination V until mid-MIS 11 through the growth of plants associated with high productivity (*Picea*, Tree, Shrubs, Arboreal pollen, *Quercus robur*) in place of steppic species, tundra or frozen soils. This increase is the highest of the last 445 ka when focusing on high latitudes records (ODP site 646 at 58°N²⁹, Lake El’gygytgyn at 67°N)¹⁹. At lakes Ohrid^{53,54} and Tenaghi Philippon¹⁸, that are located around 40°N, while the increase in pollen abundance during Termination V is comparable to those observed during younger ones, the pollen peak lasts longer (Supplementary Figure 3). In parallel, the very high and long-lasting Si/Ti ratio and TOC accumulation recorded in lake sediments^{19,59} are the consequences of a strong and longer deglacial productivity^{18,19,53,54} (Figure 3). These regional indications of terrestrial productivity are in agreement with the increase in global productivity inferred from $\Delta^{17}\text{O}$ of O_2 and increase in the terrestrial vs oceanic productivity ratio inferred from $\delta^{18}\text{O}_{\text{atm}}$.”

Similarly, I could not find any higher marine productivity from the variation of TOC and alkenone MARs. As for alkenone, termination IV indicates rather higher peak value.

Supporting evidence is thus higher carbonate productivity in the ocean only. This is consistent with the result in the Indian Ocean by Rickaby et al. (2007).

>> We thank Reviewer 1 for highlighting this important aspect. We certainly agree that the specific characteristic of oceanic productivity during Termination V, is to be found in the carbonate production.

In this paper, we discuss the variations of oceanic primary productivity based on TOC and alkenone records from ODP core 1090. We agree with the reviewer that alkenone are higher around 350 ka, before Termination IV. What we want to show here, is that organic proxies don't

show any particular increase in oceanic primary productivity during Termination V and MIS 11 compared to other terminations and interglacial periods. Instead, we explain that the decrease of these two records during Termination V and MIS 11 can reflect a decrease in oceanic productivity, thus supporting the interpretation that this is a higher terrestrial productivity that drives the long-lasting decreasing trend of $\Delta^{17}\text{O}$ of O_2 signal during that time interval. There is a lack in publications on oceanic primary productivity (and particularly in the Southern Ocean that is expected to be the key region of the ocean modulating marine primary productivity during this time period), so we can't further discuss its impact on biosphere productivity.

Furthermore, the combination of TOC and CaCO_3 records permits to depict the specific pattern of the productivity at the ODP 1090 site during Termination V. We paid particularly attention to clearly outline this aspect in the revised manuscript and now discuss TOC/ CaCO_3 ratio (Figure 3), that is best to represent past changes in Biological Carbon Pump (Hain et al., 2013; Duchamp-Alphonse et al., 2018). Indeed, this ratio considers the two biological processes linked to the carbon cycle: the net export of organic carbon to the deep ocean (i.e. the Soft Tissue Pump), that contributes to decrease atmospheric CO_2 , and the production and export of CaCO_3 (i.e. the Carbonate Counter Pump) that has the opposite effect. While this ratio shows its highest values around 30 ka, it clearly highlights this striking decrease during Termination V compared to the younger ones. As highlighted by Reviewer 1, this is indeed consistent with Rickaby et al., (2007) but also Barker et al., (2006), or more recently Saavedra-Pellitero et al., (2017) who bring to light an overall increase in coccolith carbonate export between 600 and 200 ka with a peak during MIS 11. This higher carbonate export in pelagic settings, implies a higher efficiency of the Carbonate Counter Pump and should therefore induce an increase in pCO_2 in the atmosphere compared to other interglacial periods, which is not seen in the CO_2 record.

We have included the following text in the new manuscript in the discussion section (lines 315 to 341) to discuss the changes in oceanic productivity:

“In the Ocean, while the biological carbon pump, including both the soft-tissue pump and the carbonate counter pump, is expected to impact the atmospheric pCO_2 , the soft-tissue pump is the only one to affect O_2 fluxes and thus, the $\Delta^{17}\text{O}$ of O_2 . To our knowledge, only one robust productivity reconstruction exists so far spanning the last 445 ka: we discuss here information from soft-tissue pump and biological carbon pump efficiencies, based on TOC and alkenone mass accumulation rates (MARs) as well as TOC/ CaCO_3 ratio obtained at site PS2489-2/ODP1090^{3,14} located within the Atlantic sector of the Subantarctic Southern Ocean. At this site, records show decreases in TOC and alkenone MARs, correlated with decreased TOC/ CaCO_3 during Terminations V to I (Figure 3). TOC and alkenone records during Termination V display no different trend compared to younger terminations, i.e. they show a less efficient soft-tissue pump probably associated with a decreased biological productivity in the Subantarctic Zone of the Southern Ocean, hence in agreement with the increase of terrestrial vs oceanic productivity over Termination V inferred from $\delta^{18}\text{O}_{\text{atm}}$.

The most striking pattern comes from the TOC/ CaCO_3 ratio directly linked to biological pump efficiency and exhibiting a peak of very large amplitude over Termination V. This productivity signal goes along the exceptional production and accumulation of the coccolith *Gephyrocapsa caribbeanica* observed worldwide^{33,61}, and particularly within the Subantarctic Zone of the Southern Ocean^{35,62} at the beginning of, and during Termination V. Such exceptional

coccolithophore acme could be enhanced by the minimum of eccentricity during this period, causing a longer duration of summer growing season⁶¹. In parallel, in the low-latitude ocean, this period is marked by a strong increase in reef carbonate accumulation, probably associated with the sea level rise documented at that time, that immersed platforms^{34,63}. The singular carbonate production by coccolithophore and coral ecosystem is more difficult to interpret straightforwardly in term of increase in biological productivity but supports the exceptional biological context over Termination V. Indeed this worldwide increase in carbonate production and preservation is expected to considerably decrease ocean $[\text{CO}_3^{2-}]$ ^{33,64}, and thus, significantly increase the atmospheric CO_2 concentration^{35,63} over Termination V.”

Additionally, as the authors refer, the rapid development of coral reefs in this period may imply the high marine productivity in terms of not only CaCO_3 but also O_2 . Both coccolithophore and corals generate photosynthetic oxygen (by symbiotic algae for corals) with co-occurring CaCO_3 formation. As a result, it seems only readable from evidence demonstrated by the authors that there was relatively high marine productivity in terms both of CaCO_3 and O_2 .

>> Indeed, Termination V is also associated with an increase in reef production at low latitudes (Husson et al., 2018). Both, coccolithophore and coral carbonate productions decrease the carbonate ion concentration CO_3^{2-} in the ocean, particularly within the deep reservoir (Qin et al., 2018), and favour deep-ocean carbonate dissolution during that interval (Barker et al., 2006), as pointed out lines 339 to 344 of the revised manuscript. Reviewer #1 links the higher coccolithophore and coral productions during this period with an increase in O_2 productivity because of the oxygen coccolithophore and symbiotic algae produce during photosynthesis. We agree that these organisms produce oxygen, but we did not find references quantifying the O_2 flux they produced, so we only can speculate their specific impact on the biosphere oxygen productivity flux during Termination V.

We made this part clearer in the revised version of the paper (lines 336 to 341).

“The singular carbonate production by coccolithophore and coral ecosystem is more difficult to interpret straightforwardly in term of increase in biological productivity but supports the exceptional biological context over Termination V. Indeed this worldwide increase in carbonate production and preservation is expected to considerably decrease ocean $[\text{CO}_3^{2-}]$ ^{33,64}, and thus, significantly increase the atmospheric CO_2 concentration^{35,63} over Termination V.”

If the atmospheric $\delta^{18}\text{O}$ during the termination V were evident, it is easily assumable that the terrestrial ecosystem sustained relatively higher productivity than the marine ecosystem. Therefore, I do not deny their conclusion. However, the authors must show other appropriate evidence for terrestrial productivity, OR revise their discussion in a major way.

>> First, as mentioned above, we modified Fig. 3 to add data recording the highest terrestrial productivity during MIS 11 (DeVernal et Hillaire-Marcel, 2008; Melles et al., 2012).

Second, we now explain carefully in the introduction that the comparison between O_2 gross productivity and local/regional information obtained on continental and oceanic sediment cores is not straightforward since these later pieces of information are not providing biosphere productivity fluxes.

Third, we now discuss the different reasons which would have favoured a higher biosphere productivity during Termination V and MIS 11: low eccentricity enabling longer summer

growing seasons, exceptional high temperature especially in the high latitude of the northern hemisphere and a relatively slow increase of sea level during Termination V enabling persistence of vegetation on low latitude emerged continental shelves over the first part of Termination V (i.e. Sunda shelf).

Finally, we also enhanced in the new manuscript the unique biosphere productivity information that we can get from the Southern Ocean. Indeed, it is suspected that the Southern Ocean is the key region of the ocean modulating global carbon variations (Sigman and Boyle, 2000; Sigman et al., 2010), and particularly when dealing with the ocean primary productivity. It has been demonstrated that the efficiency of the soft-tissue pump has changed in the past, thanks to higher export production within the Southern Ocean, and more particularly within the Subantarctic Zone (Martínez-García et al., 2009). We report here biological pump patterns and by extension productivity during Termination V compared to younger ones from the TOC, alkenone (and CaCO_3) fluxes obtained at site PS2489-2/ODP1090 (Subantarctic Zone), as the only high-resolution and well dated existing records related to productivity for the last 800 ka, so far (Diekmann and Kuhn, 2002, Martínez García et al., 2009, Duchamp-Alphonse et al., 2018). The information on ocean productivity with an emphasis on Southern Ocean is discussed in the revised manuscript (within the introduction: lines 59 to 74 and the discussion: lines 315 to 341 and 344 to 350).

" In the ocean, little is known about deglacial oceanic productivity over the last 800 ka. Increased contents of organic matter (Total Organic Carbon (TOC) and biomarkers)^{3,14} in sediments, have been used to reflect increased downward flux of carbon associated with the photosynthetic biomass. However, while such records provide crucial information about the efficiency of the soft-tissue pump, they remain sparse and cannot reliably be used to infer oceanic primary productivity. Besides, they do not consider the relative contribution of the downward flux of Particulate Inorganic Carbon (mainly CaCO_3) produced by calcifying plankton in the sunlit ocean (mainly coccolithophores and planktonic foraminifera), that creates a surface-to-deep alkalinity gradient, causing CO_2 to be released back to the atmosphere¹⁵ and that represents the carbonate counter pump. Past changes in biological carbon pump are best represented by changes in the buried TOC/ CaCO_3 ratio, suggested to reflect C-rain ratio¹⁶. However, despite its accuracy to provide biological export production, only one record of sedimentary TOC/ CaCO_3 exists in the Southern Ocean, for the last 800 ka, so far¹⁴. This record is however very precious since Southern Ocean biosphere productivity has a strong potential for increases in the past while on lower latitudes, the biosphere productivity is already maximum today¹⁷."

"In the Ocean, while the biological carbon pump, including both the soft-tissue pump and the carbonate counter pump, is expected to impact the atmospheric pCO_2 , the soft-tissue pump is the only one to affect O_2 fluxes and thus, the $\Delta^{17}\text{O}$ of O_2 . To our knowledge, only one robust productivity reconstruction exists so far spanning the last 445 ka: we discuss here information from soft-tissue pump and biological carbon pump efficiencies, based on TOC and alkenone mass accumulation rates (MARs) as well as TOC/ CaCO_3 ratio obtained at site PS2489-2/ODP1090^{3,14} located within the Atlantic sector of the Subantarctic Southern Ocean. At this site, records show decreases in TOC and alkenone MARs, correlated with decreased TOC/ CaCO_3 during Terminations V to I (Figure 3). TOC and alkenone records during Termination V display no different trend compared to younger terminations, i.e. they show a less efficient soft-tissue pump probably associated with a decreased biological productivity in the Subantarctic Zone of

the Southern Ocean, hence in agreement with the increase of terrestrial vs oceanic productivity over Termination V inferred from $\delta^{18}\text{O}_{\text{atm}}$.

The most striking pattern comes from the TOC/CaCO₃ ratio directly linked to biological pump efficiency and exhibiting a peak of very large amplitude over Termination V. This productivity signal goes along the exceptional production and accumulation of the coccolith *Gephyrocapsa caribbeanica* observed worldwide^{33,61}, and particularly within the Subantarctic Zone of the Southern Ocean^{35,62} at the beginning of, and during Termination V. Such exceptional coccolithophore acme could be enhanced by the minimum of eccentricity during this period, causing a longer duration of summer growing season⁶¹. In parallel, in the low-latitude ocean, this period is marked by a strong increase in reef carbonate accumulation, probably associated with the sea level rise documented at that time, that immersed platforms^{34,63}. The singular carbonate production by coccolithophore and coral ecosystem is more difficult to interpret straightforwardly in term of increase in biological productivity but supports the exceptional biological context over Termination V. Indeed this worldwide increase in carbonate production and preservation is expected to considerably decrease ocean [CO₃²⁻]^{33,64}, and thus, significantly increase the atmospheric CO₂ concentration^{35,63} over Termination V.”

“Some processes can play a role in the moderation of the pCO₂ such as physical pumping of CO₂ in the Southern Ocean through solubility effect or ventilation. A recent study provided indication on the change in the surface temperature of the Southern Ocean on the same timescale than atmospheric CO₂ variations over the last 700 ka with a link to ocean ventilation⁶⁵. The changes in Southern Ocean δ temperature and ventilation are not different during Termination V than over the 4 youngest terminations, therefore showing no evidence of an enhanced physical pumping over this time period.”

Second, according to the study by Schilt et al. (2010), the long-lasting peak of CO₂ during MIS11 seems to co-occur with similar peak shapes of methane and nitrous oxide. Therefore, their hypothesis that the enhanced terrestrial productivity during the termination V might reduce or disappear the CO₂ peak which should be there by the carbonate formation may be required re-consideration.

>> We agree that the long peak for MIS 11 is seen in many parameters (CO₂, temperature, CH₄, etc...). This is a general characteristic of MIS 11 probably due to the low eccentricity context as now mentioned in the new manuscript (l. 107). Still, the natural sources for CO₂, CH₄ and N₂O are different, and their past variations are not similar. The CH₄ variations are in general much more abrupt than the CO₂ variations during termination and glacial periods. This is also seen during Termination V and beginning of MIS 11 with the CH₄ abrupt increase being delayed by several ka with respect to the beginning of the slower increase of CO₂ (Figure R1). Similarly, while the CH₄ record exhibits two clear peaks over MIS 11, the CO₂ record over MIS 11 better shows a slow increase and a delayed maximum over the second CH₄ peak (Figure R1). The CH₄ variations are closely linked to hydrological cycle (monsoon intensity) in the low latitude continental areas (Rhodes et al., 2017). Monsoon intensity is not particularly strong during MIS11 because of low precession variations so that no large peak in CH₄ is expected during this period. On the other hand, carbonate formation is not expected to lead to CH₄ or NO₂ atmospheric variations while it should strongly affect atmospheric CO₂ concentration.

What we question here is the fact that the maximum of CO₂ during MIS 11 is not occurring at the beginning of MIS 11 as observed for the younger Terminations but better about 10 ka

(depending on the timescale...) after the beginning of MIS 11. Such particular shape of CO₂ could be linked to a strong biological productivity during Termination V which would lower the CO₂ atmospheric concentration. Still, we agree with the reviewer that other processes may be at play such as the effect of a relatively slow increase of the austral ocean temperature (maximum of ocean temperature reconstructed by Uemura et al. (2018) is not reached at the beginning of MIS 11). This could explain a slower pCO₂ increase by solubility effect but it could also be linked to the relatively slow sea level increase favouring emergence of vegetated low latitude continental shelves contributing to terrestrial productivity. The discussion of such mechanisms is now included in the new manuscript (lines 305 to 314 and 344 to 350).

“Several factors could explain the higher terrestrial productivity recorded during Termination V and MIS 11 compared to other younger terminations. First, the relatively slow sea level rise over Termination V⁶⁰ leaves some emerged vegetated terrestrial platforms (e.g. Sunda plate). Then, terrestrial biosphere productivity could be favoured in the context of low eccentricity leading to less pronounced seasonality and a longer duration of summer growing season¹⁹. Finally, MIS 11 biosphere productivity can be enhanced at high latitude of the northern Hemisphere because of the strong warming and melting of ice sheet volumes in Northern Hemisphere at mid-MIS 11 leading to more available lands^{19,59}, favouring plant development/productivity. Some also suggest a more humid and warm climate, with higher precipitation compared to MIS 1^{19,59}.”

“Some processes can play a role in the moderation of the pCO₂ such as physical pumping of CO₂ in the Southern Ocean through solubility effect or ventilation. A recent study provided indication on the change in the surface temperature of the Southern Ocean on the same timescale than atmospheric CO₂ variations over the last 700 ka with a link to ocean ventilation⁶⁵. The changes in Southern Ocean temperature and ventilation are not different during Termination V than over the 4 youngest terminations, therefore showing no evidence of an enhanced physical pumping over this time period.”

Figure R1: Records of CO₂, CH₄ and δD from the same EPICA Dome C ice core (i.e. without any possible uncertainty between CH₄ and CO₂ temporal evolutions). The vertical bars indicate the onsets for main increases of CO₂ and CH₄ over Termination V.

Moreover, Uemura et al. (2018) indicated that there was a strong positive relationship between atmospheric CO₂ concentration and the temperature over the Southern Ocean, as well as the positive relationship between CO₂ and the strength of deepwater circulation. For justification, authors should refer their studies properly.

We thank reviewer 1 for highlighting the paper of Uemura et al., 2018 on reconstruction of Antarctic and Southern Ocean surface Temperature (ΔT_{source}). In this paper, the authors show that the relationship in the timing of the variations of ΔT_{source} , CO₂ and insolation is influenced by eccentricity. In the context of our study, the ΔT_{source} reconstruction and its correlation with atmospheric CO₂ confirms the key role of the Southern Ocean on the variations of CO₂.

Uemura et al. (2018) show that the asynchrony between these three parameters is reduced during periods of low eccentricity. In order to explain this 400 ka cycle on asynchrony between ΔT_{source} and CO₂, they show in Fig.5c, a record of the variations of the deep ocean ventilation in the Southern Ocean. This record indicates an increase in deep ocean ventilation during Termination V and MIS 11 but it doesn't show any specific pattern during MIS 11 compared to the younger interglacial periods. This confirms that an exceptional biospheric productivity is a better candidate to prevent a strong CO₂ peak at the beginning of MIS 11. Similarly, the ΔT_{source} reconstruction does not display a stronger increase over Termination V than over younger terminations hence suggesting that the solubility effect is not playing a different role during Termination V than over younger terminations.

Therefore, while the paper of Uemura et al. (2018) is interesting to better understand the relation between orbital parameters and the timing in the variations of Southern Ocean Temperature and CO₂, it doesn't show any specific impact of ΔT_{source} or oceanic ventilation on CO₂ changes over Termination V and MIS 11.

We added a paragraph on the role of ventilation and surface temperature of the Southern Ocean in the discussion of the revised manuscript (lines 344 to 350):

“Some processes can play a role in the moderation of the pCO₂ such as physical pumping of CO₂ in the Southern Ocean through solubility effect or ventilation. A recent study provided indication on the change in the surface temperature of the Southern Ocean on the same timescale than atmospheric CO₂ variations over the last 720 ka with a link to ocean ventilation⁵². The changes in Southern Ocean temperature and ventilation are not different during Termination V than over the 4 youngest terminations, therefore showing no evidence of an enhanced physical pumping over this time period.”

The last, the periodicities of precession and obliquity are illustrated in Figs 1 and 3 as well as supplementary Figs 3 and 4, although I could not find any description for them in the text. I found a description for eccentricity (L270) but not illustrated. Why and why not?

>> We thank the reviewer for highlighting this issue in our paper.

In the revised version of the manuscript, we added the eccentricity curve in Fig. 3 and we introduced the importance of the eccentricity context in the introduction (lines 106 to 108). We have removed precession and obliquity records because it does not provide any information to the discussion.

“Termination V is also occurring in a particular orbital context, i.e. low eccentricity around 400 ka, which is known to have an influence on the carbon cycle as observed in $\delta^{13}\text{C}$ oceanic records³¹.”

Specific

comments:

Terms "marine" and "oceanic" are mixed. Please choose one for clarity.

We chose the term oceanic to talk about productivity or records and we chose marine to talk about cores and sediments.

Terms "continental" and "terrestrial" are mixed. Please choose one for clarity.

We chose the term terrestrial to refer to the terrestrial productivity.

L42: "than" instead of "over"

corrected

L60: "BCP" did not appear afterwards. Not necessary.

We removed the abbreviation.

L71: "as same as" instead of "as for"

amended

L78: "their temporal variation" instead of "their variation"

Revised (l.83)

L81-84: References must be needed.

We added the following reference: Luz et al., 1999 (l. 88-92)

L85-86: One or more reference must be needed.

We added the following reference: Luz et al., 1999 (l. 92-94)

L93-95: References must be needed.

We added the following references: Jouzel et al., 2007, Siegenthaler et al., 2005, DeVernal et Hillaire-Marcel, 2008, Raymo et Mitrovica, 2012 (l.103-105)

L94: What is "milder to hotter warm"?

It was replaced by "mild to warm" (l.104)

L97-98: higher sea level than the present
amended

L96-100: I suggest "Termination V than the present, whereas the CO₂ level.....warm periods. The CO₂ fluxes....."

corrected

L103-104: What is "their patterns and their specific relationships"? Please specify.

To be clearer, we replaced this sentence by "the impact of this large carbonate production" (l.115)

L109: What kinds of fluxes?

We added "oxygen fluxes" (l.120)

L111-112: Are "local" and "global" in conflicting structures? In the following text, they seem to complement each other.

The sentence has been removed in the revised version

L120: "obtained by corrections for" instead of "corrected from"

Revised (l.143)

L153-155: Hard to understand. Does the age mean the right shoulder of broad peak (plateau) of CO₂?

We added ages to the sentence to make it clearer (l. 176-177): "over Termination V, between 430 and 415 ka, when the maximum in CO₂ concentration is reached at 425 ka, more than 10 ka before the minimum in $\Delta^{17}\text{O}$ of O₂ recorded at 415 ka."

L155-157: Please apply statistical tests if you mention the data robustness.

We added in Supplementary figure 1 the calculated uncertainties for the slopes of the relationships between $\Delta^{17}\text{O}$ of O₂ and CO₂ to show that the absolute value of the slope of Termination V is statistically significantly lower than for the other terminations.

L159: Delete "or possible"

Deleted

L160: "always" instead of "systematically"
Modified (l. 185)

L174: "rate-limited" instead of "limited" for clarity
Corrected (l.199)

L188: I suggest "possibility" instead of "propositions"
The term has been removed in the revised version

L221-223: This sentence may let readers confused because authors give freedom to the ratio soon after.
We removed the sentence to be clearer

L236: "during other terminations" instead of "over the terminations"
It doesn't appear anymore in the revised version

L247: Values are neither strong or weak, but either high or low.
The term has been changed (l. 283)

L269-270: As forementioned, "eccentricity" appears suddenly without illustration.
We added the eccentricity record in Fig.3 to illustrate the discussion.

L285: I disagree with this. Termination IV looks larger.
We revised the text to better interpret the variations of COT and CaCO₃

Fig.1: The upper panel is not treated in the text.
We removed the upper panel from this figure

Fig.3: I suggest authors to combine with supplementary Fig.4 and replace it.
We combined the two figures by adding the terrestrial productivity records of Melles et al., (2012) and de Vernal et Hillaire-Marcel, (2008) in the Figure 3 and we put the pollen records of Tenaghi Philippon lake (Tzedakis et al., 2006) in the supplementary figure 3 to better illustrate the increasing terrestrial productivity during MIS 11 in the main text.

Fig.3: The upper panel is not treated in the text.
We removed the upper panel from this figure and added the eccentricity record to illustrate the discussion.

Fig.3: Shaded zones slightly mismatch the text. L149-L155, for example.
Shaded zones were better defined as the period of rapid increase in CO₂ during deglaciations.

Fig.3: I suggest to add up and down arrows for qualitative controls on the variation of d18O_{atm}.
We chose not to add arrows because the figure contains already a lot of information.

Supplementary Fig.2: Identical to the middle panel of Fig. 1. It seems not necessary.
We removed this figure.

Supplementary Fig.3: The upper panel is not treated in the text.
We removed the upper panel from Fig. S3

Supplementary Fig.4: The upper panel is not treated in the text.
We removed the upper panel from Fig S4.

Supplementary Fig.4: Suggesting to be combined with Fig.3.
We added pollen records and Si/Ti ratio of Lake El'gygytgyn (Melles et al., 2012) as well as pollen record of ODP site 646 (de Vernal et Hillaire Marcel, 2008) to the main figure and let pollen record of Tenaghi philippon (Tzedakis et al., 2006) in the supplementary figure 3.

Suggested references

Rickaby R. E. M. et al. (2007) *Earth Planet. Sci. Lett.* 253, 83–95 (2007)
Uemura R. et al. (2018) *Nature Communications*, DOI: 10.1038/s41467-018-03328-3

References:

Barker, S. *et al.* Globally increased pelagic carbonate production during the Mid-Brunhes dissolution interval and the CO₂ paradox of MIS 11. *Quaternary Science Reviews* **25**, 3278–3293 (2006).

Ciais, P. *et al.* Large inert carbon pool in the terrestrial biosphere during the Last Glacial Maximum. *Nature Geoscience* **5**, 74–79 (2012).

de Vernal, A. & Hillaire-Marcel, C. Natural Variability of Greenland Climate, Vegetation, and Ice Volume During the Past Million Years. *Science* **320**, 1622–1625 (2008).

Diekmann, B. & Kuhn, G. Sedimentary record of the mid-Pleistocene climate transition in the southeastern South Atlantic (ODP Site 1090). *Palaeogeography, Palaeoclimatology, Palaeoecology* **182**, 241–258 (2002).

Duchamp-Alphonse, S. *et al.* Enhanced ocean-atmosphere carbon partitioning via the carbonate counter pump during the last deglacial. *Nature Communications* **9**, (2018).

Hain, M. P., Sigmal, D. & Haug, G. H. 8.18–The biological Pump in the Past. in *Treatise on Geochemistry* vol. 8 485–517 (2014).

Husson, L. *et al.* Reef Carbonate Productivity During Quaternary Sea Level Oscillations. *Geochemistry, Geophysics, Geosystems* **19**, 1148–1164 (2018).

Jouzel, J. *et al.* Orbital and Millennial Antarctic Climate Variability over the Past 800,000 Years. *Science* **317**, 793–796 (2007).

Martínez-García, A. *et al.* Links between iron supply, marine productivity, sea surface temperature, and CO₂ over the last 1.1 Ma. *Paleoceanography* **24**, PA1207 (2009).

Melles, M. *et al.* 2.8 Million Years of Arctic Climate Change from Lake El'gygytyn, NE Russia. *Science* **337**, 315–320 (2012).

Qin, B., Li, T., Xiong, Z., Algeo, T. J. & Jia, Q. Deep-Water Carbonate Ion Concentrations in the Western Tropical Pacific Since the Mid-Pleistocene: A Major Perturbation During the Mid-Brunhes. *Journal of Geophysical Research: Oceans* **123**, 6876–6892 (2018).

Raymo, M. E. & Mitrovica, J. X. Collapse of polar ice sheets during the stage 11 interglacial. *Nature* **483**, 453–456 (2012).

Reyes, A. V. *et al.* South Greenland ice-sheet collapse during Marine Isotope Stage 11. *Nature* **510**, 525–528 (2014).

Rhodes, R. H. *et al.* Atmospheric methane variability: Centennial-scale signals in the Last Glacial Period: Centennial-Scale Methane Variability. *Global Biogeochem. Cycles* **31**, 575–590 (2017).

Rickaby, R. E. M. *et al.* Coccolith chemistry reveals secular variations in the global ocean carbon cycle? *Earth and Planetary Science Letters* **253**, 83–95 (2007).

Saavedra-Pellitero, M., Baumann, K.-H., Ullermann, J. & Lamy, F. Marine Isotope Stage 11 in the Pacific sector of the Southern Ocean; a coccolithophore perspective. *Quaternary Science Reviews* **158**, 1–14 (2017).

Siegenthaler, U. Stable Carbon Cycle-Climate Relationship During the Late Pleistocene. *Science* **310**, 1313–1317 (2005).

Sigman, D. M. & Boyle, E. A. Glacial/interglacial variations in atmospheric carbon dioxide. *Nature* **407**, 859–869 (2000).

Sigman, D. M., Hain, M. P. & Haug, G. H. The polar ocean and glacial cycles in atmospheric CO₂ concentration. *Nature* **466**, 47–55 (2010).

Tzedakis, P. C., Hooghiemstra, H. & Pälike, H. The last 1.35 million years at Tenaghi Philippon: revised chronostratigraphy and long-term vegetation trends. *Quaternary Science Reviews* **25**, 3416–3430 (2006).

Uemura, R. *et al.* Asynchrony between Antarctic temperature and CO₂ associated with obliquity over the past 720,000 years. *Nat Commun* **9**, 961 (2018).

Reviewer #2 (Remarks to the Author):

Brandon and coworkers present new oxygen triple-isotope data in O₂ across Termination V that show a similar, but perhaps slightly larger peak-to-peak range in D17O-O₂ as Terminations I, II, and IV (TIII is smaller). Importantly, however, this D17O change shows a different relationship with coeval atmospheric CO₂ concentrations than in the other glacial terminations: CO₂ seems to rise more rapidly across Termination V, relative to D17O-O₂. Again, there is a bit of variability in this relationship, but it appears demonstrably different over Termination V. The authors use this asynchrony and the model of Landais *et al.* 2007 to infer an excess in biosphere

productivity during the initial parts of the subsequent interglacial.

The data here are likely good, but I think the model (which serves as the basis for the productivity claim) is flawed, and its potential shortcomings are not mentioned in the text or supplement.

Methodologically, I think the model's main flaw is its treatment of $\Delta^{17}\text{O}$ as a conserved quantity upon mixing; it is not, as many have shown in the literature (Miller et al. 2002, Prokopenko et al., 2011, and Kaiser et al., 2011 to name a few). The net result is an inaccurate calculation of the mixing mass balance between biological and stratospheric endmembers. It therefore predicts larger productivity swings between glacial and interglacial climates than a model that uses a more accurate mixing formulation (Blunier et al. 2012). One can see this in Supplementary Figure 3. Blunier et al.'s model would probably yield muted changes in global productivity, much more similar to the PST scenario than the authors claim in the abstract.

>> We understand the concerns of the reviewer about the validity of our model and we have addressed it.

The difference between the approach in Luz et al. (1999), Landais et al. (2007) and Blunier et al. (2012) is on the way the stratospheric flux of anomaly $\Delta^{17}\text{O}$ of O_2 is scaled to CO_2 since CO_2 is the limiting species for production of the mass independent fractionation of O_2 . In the first approach, the scaling is performed on the $\Delta^{17}\text{O}$ of O_2 and in the second approach the scaling is performed in the $\delta^{17}\text{O}$ of O_2 .

As suggested by Reviewer 2, the best approach to deal with this difference of model conception is to run the two models. We thus ran the model of Blunier et al., (2012) on our data of $\Delta^{17}\text{O}$ of O_2 over Termination V and MIS 11. We show here two outputs of the model of Blunier et al. (2012): the green curve represents the biosphere productivity reconstruction with a constant H_2O anomaly of 40 per meg as expected from up to date knowledge of variations of H_2O anomaly over the last deglaciation (Winkler et al., CP, 2012; Schoenemann et al., JGR, 2014) and the red curve representing the biosphere productivity reconstruction with a H_2O anomaly derived from δD as performed in the original publication of Blunier et al. (2012). The second hypothesis was indeed favoured in Blunier et al. (2012) based on the record of ^{17}O -excess ($\ln(\delta^{17}\text{O}+1)-0.528*\ln(\delta^{18}\text{O}+1)$) over the last 150 ka from Landais et al. (2008) in the Vostok ice core but further analyses have shown that the 10 to 20 ppm ^{17}O -excess increase observed at Vostok, East Antarctica over the last deglaciation is a local effect and not a global effect (Winkler et al., 2012) so that we favour now the results issued from the first run. The model of Blunier et al. (2012) was still ran here with the two assumptions.

Whatever the chosen assumption, the model of Blunier et al. (2012) displays a lower biospheric productivity during glacial periods than during interglacial periods. However, the reconstructed glacial productivity is systematically higher using Blunier's model than the Landais' one.

In addition, using the two assumptions, the two runs performed with the Blunier's model show a higher biosphere productivity during Termination V and MIS 11 compared to younger interglacial periods, with a 10% larger productivity during MIS 11 than during the preindustrial period. There is therefore a consistent difference between the two models: the model of Landais et al. (2007) leads to larger reconstructed productivity differences with the pre-industrial period both in glacial periods and during MIS 11. However, the two models (Landais & Blunier) both show the highest increase in biosphere productivity during MIS 11 compared to other interglacial periods of the last 450 ka. Such consistent result supports our main

conclusion even if we agree with reviewer 2 that our error bars should be enlarged to take into account the uncertainty linked to the model.

The fact that the different models give the same exceptionally high productivity during MIS 11 is not surprising: it actually directly comes from the fact that the $\Delta^{17}\text{O}$ of O_2 is decreasing much more slowly compared to the CO_2 increase during Termination V and first part of MIS 11 than over the other deglaciations as illustrated in Figure S1.

We have included in the new manuscript and in the supplementary material the comparison between the two models for productivity reconstruction from $\Delta^{17}\text{O}$ of O_2 (Supplementary Figure 2).

The main text (part results, lines 228 to 245) has been modified as:

“We ran 3 sensitivity tests to better estimate the uncertainties related to fractionation processes within the water cycle²⁴, photosynthesis⁴² and respiration⁴³. The estimate for triple isotopic composition of oxygen produced by terrestrial and oceanic biospheres, $\delta^{17}\text{O}_{\text{bio}}$, was revised compared from those of Landais et al. (2007)⁴⁰ taking into account recent studies. The final estimates vary between 180.5 ± 60.5 ppm and 136 ± 44 ppm for the $\delta^{17}\text{O}_{\text{bio}}$ during the Last Glacial Maximum and the pre-industrial period respectively, the uncertainties being mainly associated with uncertainties in the fractionation coefficients for the different biospheric processes⁴⁰ (Supplementary Table 2). Combining these estimates with the covariation of CO_2 concentration and flux of negative $\Delta^{17}\text{O}$ of O_2 anomaly from the stratosphere, we converted the full $\Delta^{17}\text{O}$ of O_2 record over the last 445 ka into an estimate for the evolution of the global oxygen productivity (Figure 3). This reconstruction was also completed with an alternative reconstruction of global oxygen productivity using an alternative model, the one of Blunier et al. (2012)²⁴, forced by our $\Delta^{17}\text{O}$ of O_2 data (Supplementary Figure 2).

Combining our different sensitivity studies, we find that the global productivity is reduced by 60-90% during glacial periods compared to interglacials. At the beginning of most interglacial periods, the global productivity remains close to the preindustrial level. The only exception to this general behavior is the strong oxygen biosphere productivity at the end of Termination V reaching values 10 to 30% higher than during the pre-industrial period.”

We added more information on the 3 sensitivity tests and the alternative reconstruction using the model of Blunier et al. (2012) in the method section (lines 537 to 594) as well as in Supplementary Figure 2 shown below:

“The reconstructed biosphere productivity using the Landais et al. (2007) formulation is displayed on Figure 2: it leads to a biosphere productivity at the beginning of MIS 11 of 20% higher than during MIS 1. Since the publication of Landais et al. (2007), new estimates of fractionation coefficients within the oxygen cycle are available^{22,43,76} and influence of the water cycle organisation has also been suggested²⁴. As a consequence, we performed 3 types of sensitivity tests to address the influences of such new determination and associated uncertainty ranges on our results.

- **Uncertainty in photosynthesis fractionation.** Some marine species show fractionation during marine photosynthesis⁴² while it has long been assumed that photosynthesis does not fractionate⁷⁷. We performed sensitivity tests to compare biosphere

productivity reconstructions without any fractionation and reconstruction with the different fractionation effects during marine photosynthesis as observed in Eisenstadt et al. (2010)⁴². The largest change on the reconstructed biosphere productivity is obtained using the observed photosynthesis fractionation effect associated with *Emiliana huxleyi* ($\delta^{18}\text{O}=5.81\text{‰}$ and slope between $\ln(\delta^{17}\text{O}+1)$ and $\ln(\delta^{18}\text{O}+1)$ of 0.5253). The reconstructed global biosphere productivity is lower with such fractionation effect: The MIS 11 productivity level is decreased by 5% with such fractionation effect compared to the reconstruction with no fractionation at photosynthesis (Supplementary Figure 2).

- **Uncertainty in respiration fractionation.** Recent studies have highlighted large variations in the relationship between $\delta^{17}\text{O}$ and $\delta^{18}\text{O}$ during respiration linked to temperature variations⁴³. When taking into account the maximum effect, i.e. a decrease of 0.005 for the slope of the relationship between $\ln(\delta^{17}\text{O}+1)$ and $\ln(\delta^{18}\text{O}+1)$ during respiration, we end up with a decrease of $\Delta^{17}\text{O}_{\text{bio}}$ by 65 ppm. The resulting biospheric productivity reconstruction is about 10% higher during MIS 11 than the initial reconstruction leading to an anomalously high O_2 flux associated with gross primary productivity during Termination V.

- **Uncertainty in the fractionation within water cycle.** The general assumption is that the relationship between $\delta^{17}\text{O}$ and $\delta^{18}\text{O}$ (meteoric water line) remains the same over glacial – interglacial cycles. However, measurements Vostok ice core have shown that the ^{17}O -excess, defined as $^{17}\text{O}\text{-excess} = \ln(1+\delta^{17}\text{O}) - 0.528 \cdot \ln(1+\delta^{18}\text{O})$, is lower by up to 20 ppm during the last glacial maximum⁷⁸. While it has been shown that this is a local effect⁷⁹, we still performed a sensitivity test with varying ^{17}O -excess of all continental meteoric waters in a range of 20 ppm between glacial and interglacial periods (lower ^{17}O -excess values for glacial than interglacial periods). Taking into account such variations, the reconstructed biospheric productivity does not differ by more than 4% from the case without change in meteoric water ^{17}O -excess.

An additional validation of our calculation comes from the comparison of our reconstruction with the oxygen biosphere productivity variations obtained by Blunier et al. (2012)²⁴ over the last 4 terminations using a different vegetation model and different formulation of the link between atmospheric CO_2 and flux of $\Delta^{17}\text{O}$ of O_2 anomaly from the stratosphere. Indeed, in our approach, we directly dealt with $\Delta^{17}\text{O}$ of O_2 anomaly in the box model while it may be more appropriate to deal with the $\delta^{17}\text{O}$ and $\delta^{18}\text{O}$ of O_2 values as done in Blunier et al. (2012)²⁴ and suggested by Prokopenko et al. (2011)⁸⁰. We thus compared our reconstruction with the one performed with the model of Blunier et al., 2012²⁴ (Supplementary Figure 2). The smaller difference in the oxygen flux associated with oxygen productivity between glacial and interglacial periods in Blunier et al. (2012)²⁴ mainly comes from the different formulations of the dependency of the stratospheric anomaly $\Delta^{17}\text{O}$ of O_2 to CO_2 atmospheric concentration. Another difference comes from the variations in the triple isotopic composition of oxygen in water between glacial and interglacial periods not taken into account in Landais et al. (2007)⁴⁰. We took this possible variation into account in our approach through the third sensitivity test explained above. We ran here the model of Blunier et al. (2012)²⁴ over the new Termination V data. The comparison between the two methods of reconstruction (Landais et al., 2007⁴⁰ vs Blunier et al., 2012²⁴) is significant. The biosphere productivity increase at the end of Termination V is less important using the

Blunier et al. (2012)²⁴ approach than using the Landais et al. (2007)⁴⁰: 10% larger than for other interglacial periods following Blunier et al. (2012)²⁴ instead of 20% following Landais et al. (2007)⁴⁰. Still, both approaches show that the biospheric productivity increase over Termination V is exceptional compared to the younger terminations.”

Supplementary Figure 2. Biosphere productivity reconstructions over the last 450 ka. All the lines represent the ratio of global biospheric productivity between the time considered and pre-industrial. The dark grey area represents the ratio between global biosphere productivity and pre-industrial biosphere productivity with uncertainty bars deduced from uncertainties in the value of $\Delta^{17}\text{O}$ of O_2 produced by the earth biosphere (Landais et al., 2007)³. The solid and dotted green lines represent respectively the maximum and minimum biosphere productivity calculated with a lower ^{17}O associated with oceanic and terrestrial productivity, taking into account the uncertainty in respiration fractionation⁴ (See Method and Supplementary Table 2). The solid and dotted blue lines represent respectively the maximum and minimum biosphere productivity calculated with a higher ^{17}O associated with oceanic productivity, taking into account the uncertainty in photosynthesis fractionation⁵ (See Method and Supplementary Table 2), The solid black line is the ratio of biosphere productivity calculated for a constant H_2O anomaly of 40 per meg over the time period considered, reconstructed with the model of Blunier et al., 2012², the dotted black line is the ratio of biosphere productivity for an anomaly derived from δD , reconstructed with the model of Blunier et al., 2012². The light grey area represents the maximum uncertainty range of the biosphere productivity reconstruction, taking into account the reconstructions of biosphere productivity described above as well as the biosphere productivity of Blunier et al., 2012².

In addition, we give here an additional support to our reconstruction based on $\Delta^{17}\text{O}$ of O_2 . We ran the IPSL coupled model equipped with the terrestrial vegetation module ORCHIDEE (version used in Chen et al., 2019) and the ocean vegetation module PISCES (version used in Le Mézo et al., 2017) to provide an estimate of the change of gross productivity flux of oxygen over the last deglaciation and compare it with our estimate based on our $\Delta^{17}\text{O}$ of O_2 data. We find an excellent agreement between both reconstructions (figure R3) and take it as an additional element confirming our approach and hence the reconstruction of biosphere productivity over Termination V. If the editor and reviewers find it useful to support our reconstruction, we can provide figure R3 in supplementary material.

Figure R2 : Reconstructions of biospheric productivity fluxes of oxygen over the last deglaciation

- Top panel : ratio between global biosphere productivity at time “t” and global biosphere productivity for pre-industrial period (expressed in O₂ flux) obtained by the coupled IPSL model equipped with the PISCES and ORCHIDEE models (crosses) and by interpretation of $\Delta^{17}\text{O}$ of O₂ data following the approach described in Landais et al. (2007) (blue envelope)
- Bottom panel: evolutions of CO₂ (black) and $\Delta^{17}\text{O}$ of O₂ (green) as in the main manuscript.

The Blunier glacial-interglacial productivity swings are much more in line with other estimates such as from $\delta^{18}\text{O}_{\text{atm}}$ and $\delta^{13}\text{C}-\text{CO}_2$ by Ciais et al. (2012). While one could conceivably take issue with those authors' interpretations, these important caveats are not mentioned in the manuscript at all. It seems appropriate to at least interpret the data using both models and acknowledge potential model biases. If the interpretation is robust with respect to at least these two models (there is at least one more, from Young et al. 2014), then the argument would be more convincing.

>> We have addressed the use of an alternative model for interpretation of $\Delta^{17}\text{O}$ of O₂ signal above. We answer here the questions of comparison to alternative reconstructions of gross biosphere productivity, i.e. without using the constraint of $\Delta^{17}\text{O}$ of O₂.

Reviewer 2 mentions the paper of Ciais et al. (2012) who use model outputs to provide an oceanic GPP of 110 GtC / year for both LGM and PI and a calculation based on $\delta^{18}\text{O}_{\text{atm}}$ to estimate the terrestrial GPP of 80 GtC / year for PI and 40 GtC / year for LGM. We gathered other estimates as well in Table R1. Combining the different estimates, we see that there is a large range in the reconstruction of global GPP during LGM with respect to Pre-industrial. Note that the figures in table R1 are given in mass of carbon and not in oxygen fluxes as in our

reconstruction: a factor of two on average should be taken into account to convert carbon fluxes in GtC/year to oxygen fluxes in GtO₂/year. Our reconstruction for LGM biosphere productivity using $\Delta^{17}\text{O}$ of O₂ and the model of Landais et al. (2007) lies on the low part of the ensemble of estimates but well beneath the broad range of estimates.

References	Pre-industrial terrestrial GPP (GtC/y)	LGM terrestrial GPP (GtC/y)	Pre-industrial oceanic GPP (GtC/y)	LGM oceanic GPP (GtC/y)
Beer et al., 2010 (model + eddy covariance flux data)	123			
Prentice et al., 2011 (model + comparison with pollen and oxygen isotopes data)	120	86		
Ciais et al., 2012 (model + d18O _{atm} data for LGM terrestrial GPP)	80	40	110	110
Reutenauer et al., 2015 (model)	126.1	81.2		
PISCES model (Bopp, com. pers.)			72	60

Table R1: Comparison of GPP Carbon for the pre-industrial and LGM periods obtained by different methods

On this basis, unfortunately, I am not sure the titular claim made by the authors is supported by the model. Productivity might still end up higher than preindustrial when using others' models, but I suspect it will not be "exceptionally high."

References:

Beer, C. *et al.* Terrestrial Gross Carbon Dioxide Uptake: Global Distribution and Covariation with Climate. *Science* **329**, 834–838 (2010).

Blunier, T., Bender, M. L., Barnett, B. & von Fischer, J. C. Planetary fertility during the past 400 ka based on the triple isotope composition of O₂ in trapped gases from the Vostok ice core. *Climate of the Past* **8**, 1509–1526 (2012).

Chen, W. *et al.* Response of vegetation cover to CO₂ and climate changes between Last Glacial Maximum and pre-industrial period in a dynamic global vegetation model. *Quaternary Science Reviews* **218**, 293–305 (2019).

Ciais, P. *et al.* Large inert carbon pool in the terrestrial biosphere during the Last Glacial Maximum. *Nature Geoscience* **5**, 74–79 (2012).

Landais, A., Lathiere, J., Barkan, E. & Luz, B. Reconsidering the change in global biosphere productivity between the Last Glacial Maximum and present day from the triple oxygen isotopic composition of air trapped in ice cores. *Global Biogeochemical Cycles* **21**, GB1025 (2007).

Landais, A., Barkan, E. & Luz, B. Record of $\delta^{18}\text{O}$ and ^{17}O -excess in ice from Vostok Antarctica during the last 150,000 years. *Geophysical Research Letters* **35**, (2008).

Le Mézo, P., Beaufort, L., Bopp, L., Braconnot, P. & Kageyama, M. From monsoon to marine productivity in the Arabian Sea: insights from glacial and interglacial climates. *Climate of the Past* **13**, 759–778 (2017).

Luz, B., Barkan, E., Bender, M. L., Thiemens, M. H. & Boering, K. A. Triple-isotope composition of atmospheric oxygen as a tracer of biosphere productivity. *Nature* **400**, 547–550 (1999).

Prentice, I. C., Harrison, S. P. & Bartlein, P. J. Global vegetation and terrestrial carbon cycle changes after the last ice age. *New Phytologist* **189**, 988–998 (2011).

Reutenauer, C. *et al.* Quantifying molecular oxygen isotope variations during a Heinrich stadial. *Climate of the Past* **11**, 1527–1551 (2015).

Schoenemann, S. W., Steig, E. J., Ding, Q., Markle, B. R. & Schauer, A. J. Triple water-isotopologue record from WAIS Divide, Antarctica: Controls on glacial-interglacial changes in ^{17}O excess of precipitation: WAIS LGM-Holocene ^{17}O excess Record. *J. Geophys. Res. Atmos.* **119**, 8741–8763 (2014).

Winkler, R. *et al.* Deglaciation records of ^{17}O -excess in East Antarctica: reliable reconstruction of oceanic normalized relative humidity from coastal sites. *Climate of the Past* **8**, 1–16 (2012).

Reviewer #3 (Remarks to the Author):

This paper deals with the triple O isotope composition of O_2 ($^{16}\text{O}_2/^{17}\text{O}^{16}\text{O}/^{18}\text{O}^{16}\text{O}$) in O_2 trapped in ice cores. The basic idea is that the relation between $\delta^{17}\text{O}$ and $\delta^{18}\text{O}$ is dependent on photochemical reactions involving CO_2 in the stratosphere, and the turnover of O_2 by respiration and photosynthesis of the biosphere. The focus of the paper is on Glacial Termination 5, which occurred about 420 thousand years ago. The paper reports new data for the oxygen isotope property of merit, $\Delta^{17}\text{O}$, for terminations 2 and 5. The Termination 5 result is quite interesting because the deglacial response between CO_2 and $\Delta^{17}\text{O}$ occur at quite different times (separated by 10 kyr or more). In contrast, one generally finds that these properties are tightly linked in time. The separation might be due to changes in the hydrologic cycle, a decrease in ocean productivity, and increase in land productivity, and changes in the distribution of productivity in the land and oceans. The authors argue that the unusual relationship between CO_2 and $\Delta^{17}\text{O}$ reflects higher land productivity during the termination. They support their conclusion by regional studies showing higher productivity around the time of interest.

My problem with this paper is that there are multiple factors influencing $\Delta^{17}\text{O}$ of O_2 . Many of these are discussed in the paper. The problem is that the authors do not have a good way of quantifying the magnitude of the changes induced by the various influences. In particular,

fractionations in the hydrologic cycle can be large and are not well known. The isotope effects associated with photosynthesis and respiration remain, to my mind, poorly characterized. For example, work of Luz raises the possibility of an inverse (positive) isotope effect associated with photosynthesis, and recent work of Stolper raises questions about respiratory isotope fractionation factors. Uncertainties in hydrologic fractionation and fractionation associated with respiration and photosynthesis are first order issues limiting the interpretation. Overall, one has a single constraint (perhaps 2 adding $\delta^{18}\text{O}$ of O_2 , but it is unclear how useful this property is), but there are many more than 2 factors governing $\Delta^{17}\text{O}$ of paleoatmospheric O_2 .

>> We understand the concern of reviewer 3 which shares similarities with comments of Reviewer 2. The method for productivity reconstruction is indeed not forward.

Still, as mentioned in the manuscript and as explained in answer to previous Reviewer 2, we want to insist on the fact that the stronger biosphere productivity reconstructed over Termination V is already a feature clearly seen on the raw data since it arises from the much slower $\Delta^{17}\text{O}$ of O_2 decrease over Termination V compared to younger Terminations (see updated supplementary figure 1 with uncertainty on slope): the slope of anticorrelation between CO_2 and $\Delta^{17}\text{O}$ of O_2 is twice smaller over Termination V than over the other terminations. As a consequence, if we admit that the biosphere fractionation factors are not different over Termination V than for the other terminations (which should not be, since it is based on physical processes), the O_2 biosphere productivity should be higher over Termination V than for the younger terminations).

Still, we fully understand the concerns of Reviewer 3 and we actually share them. This was the reason why we presented some new sensitivity tests in this new version of the manuscript. This is also the sense of showing an alternative reconstruction (from Blunier et al., 2012) in the new manuscript as well as displaying a comparison between output of a coupled model equipped with biospheric modules and our reconstruction (cf answer to reviewer 2). All these reconstructions validate our approach and show a significantly larger biosphere productivity during Termination V in agreement with our raw data.

To specifically answer comments of Reviewer 3, we ran 3 additional sensitivity tests as described now in the method section (lines 544 to 572) as:

- **“Uncertainty in photosynthesis fractionation.** Some marine species show fractionation during marine photosynthesis⁴² while it has long been assumed that photosynthesis does not fractionate⁷⁷. We performed sensitivity tests to compare biosphere productivity reconstructions without any fractionation and reconstruction with the different fractionation effects during marine photosynthesis as observed in Eisenstadt et al. (2010)⁴². The largest change on the reconstructed biosphere productivity is obtained using the observed photosynthesis fractionation effect associated with *Emiliana huxleyi* ($\delta^{18}\text{O}=5.81\text{‰}$ and slope between $\ln(\delta^{17}\text{O}+1)$ and $\ln(\delta^{18}\text{O}+1)$ of 0.5253). The reconstructed global biosphere productivity is lower with such fractionation effect: The MIS 11 productivity level is decreased by 5% with such fractionation effect compared to the reconstruction with no fractionation at photosynthesis (Supplementary Figure 2).
- **Uncertainty in respiration fractionation.** Recent studies have highlighted large variations in the relationship between $\delta^{17}\text{O}$ and $\delta^{18}\text{O}$ during respiration linked to temperature variations⁴³. When taking into account the maximum effect, i.e. a decrease of 0.005 for

the slope of the relationship between $\ln(\delta^{17}\text{O}+1)$ and $\ln(\delta^{18}\text{O}+1)$ during respiration, we end up with a decrease of $\Delta^{17}\text{O}_{\text{bio}}$ by 65 ppm. The resulting biospheric productivity reconstruction is about 10% higher during MIS 11 than the initial reconstruction leading to an anomalously high O_2 flux associated with gross primary productivity during Termination V.

- **Uncertainty in the fractionation within water cycle.** The general assumption is that the relationship between $\delta^{17}\text{O}$ and $\delta^{18}\text{O}$ (meteoric water line) remains the same over glacial – interglacial cycles. However, measurements Vostok ice core have shown that the ^{17}O -excess, defined as $^{17}\text{O}\text{-excess} = \ln(1+\delta^{17}\text{O}) - 0.528 \cdot \ln(1+\delta^{18}\text{O})$, is lower by up to 20 ppm during the last glacial maximum⁷⁸. While it has been shown that this is a local effect⁷⁹, we still performed a sensitivity test with varying ^{17}O -excess of all continental meteoric waters in a range of 20 ppm between glacial and interglacial periods (lower ^{17}O -excess values for glacial than interglacial periods). Taking into account such variations, the reconstructed biospheric productivity does not differ by more than 4% from the case without change in meteoric water ^{17}O -excess.”

All these new tests are included in the new manuscript. The uncertainty range has been broadened accordingly on Figure 3 and the discussion has been modified.

In addition, we added other data from continent and oceanic biosphere over Termination V and MIS 11 to support our conclusion as explained in details in the answer to comments of reviewer 1. We have also fully rewritten the introduction and discussion to (1) better present the specificity of Termination V in the context of the last 5 deglaciations with a focus on biosphere signals and (2) give explanations for an enhanced biosphere productivity during this period.

I think this paper could be modified by adding a rigorous discussion of challenges associated with reconstructing productivity from $\Delta^{17}\text{O}$, and published in a geochemical journal. I do not recommend it for Nature Communications, because I think the interpretation is incomplete, and because I think the paper lacks robust climate insights that might be of interest to a more general audience.

**Title:** Exceptionally high biosphere productivity at the beginning of Marine Isotopic Stage 11

**Order of Authors:** Margaux Brandon^{*a,b}, Amaelle Landais^a, Stéphanie Duchamp-Alphonse^b,
Violaine Favre^a, Léa Schmitz^a, Héloïse Abrial^a, Frédéric Prie^a, Thomas Extier^a, Thomas
Blunier^c

**Author Affiliation :**

6 ^aLaboratoire des Sciences du Climat et de l'Environnement, LSCE/IPSL, CEA-CNRS-UVSQ,
Université Paris-Saclay, 91191, Gif-sur-Yvette, France

8 ^bGEOPS, Univ. Paris-Sud, CNRS, Université Paris-Saclay, 91405 Orsay, France

9 ^cCopenhagen Univ., Niels Bohr Institute, Centre for Ice and Climate, Juliane Maries Vej 30,
DK-2100 Copenhagen, Denmark

***Corresponding author:** Margaux Brandon

Address : GEOPS, Univ. Paris-Sud, CNRS, Université Paris-Saclay, 91405 Orsay, France

Phone number: 33 (1) 69 15 61 26

E-mail address: margaux.brandon@u-psud.fr

**Abstract**

Biosphere productivity is associated with important CO₂ fluxes through photosynthesis and
respiration. Quantifying the variations of global biosphere productivity over deglaciations is
thus key for a better comprehension of the variations of atmospheric CO₂ over glacial-
interglacial cycles. Using the first high resolution record of $\Delta^{17}\text{O}$ of O₂ over Termination V and
Marine Isotopic Stage (MIS) 11, we reconstruct the past global biosphere productivity over this
key period for glacial-interglacial cycles of the last 800,000 years, corresponding to the first
termination with a large CO₂ amplitude followed by the longest interglacial. We show that the
global oxygen biosphere productivity at the end of Termination V is 10 to 30 % higher
compared to the younger terminations. We suggest that higher biosphere productivity was due
to extended period of strong terrestrial productivity that probably contributed to reduce the
atmospheric CO₂ level at the beginning of MIS 11.

**INTRODUCTION**

The largest pre-anthropogenic changes in atmospheric CO₂ concentration of the last
800,000 years are observed during deglaciations with increases of up to 100 ppm in a few
thousand years¹. Oceanic carbon reservoir is broadly suspected to play a central role in these
atmospheric CO₂ increases. Leading hypotheses invoke CO₂ degassing from the ocean induced
by more vigorous convection in the Southern Ocean (physical pump)², concomitant decrease
of the net organic matter export to the deep ocean (soft-tissue pump)³, enhanced exchanges
between ocean surface and atmosphere due to sea-ice melting⁴, and increased sea-surface
temperature (solubility pump)⁵. Modelling studies simulating changes in oceanic processes are
quite controversial^{6,7} but in all cases, they are not able to explain the full increase of atmospheric
CO₂ rises during deglaciations. In parallel, terrestrial primary productivity and carbon stock

increase during deglaciations, thus acting as a significant CO₂ land sink^{8,9} so that an important
additional source of CO₂ is required to explain the entire deglacial CO₂ pattern. The missing
component has been suggested to be the important release of inert carbon from a thawing
permafrost^{8,10}.

Quantifying changes in the carbon cycle over deglaciations relies on data compilation
and modelling studies to estimate both carbon stocks and carbon fluxes⁸. Some information on
carbon stocks can be obtained from $\delta^{13}\text{C}$ of carbonates using a wealth of data obtained from
marine sediments¹¹ and $\delta^{13}\text{C}$ of atmospheric CO₂^{12,13}. In parallel, information on the evolution
of past global productivity, a major component of carbon flux, is very sparse and often limited
to the last deglaciation^{8,9}.

In the ocean, little is known about deglacial oceanic productivity over the last 800 ka.
Increased contents of organic matter (Total Organic Carbon (TOC) and biomarkers)^{3,14} in
sediments, have been used to reflect increased downward flux of carbon associated with the
photosynthetic biomass. However, while such records provide crucial information about the
efficiency of the soft-tissue pump, they remain sparse and cannot reliably be used to infer
oceanic primary productivity. Besides, they do not consider the relative contribution of the
downward flux of Particulate Inorganic Carbon (mainly CaCO₃) produced by calcifying
plankton in the sunlit ocean (mainly coccolithophores and planktonic foraminifera), that creates
a surface-to-deep alkalinity gradient, causing CO₂ to be released back to the atmosphere¹⁵ and
that represents the carbonate counter pump. Past changes in biological carbon pump are best
represented by changes in the buried TOC/CaCO₃ ratio, suggested to reflect C-rain ratio¹⁶.
However, despite its accuracy to provide biological export production, only one record of
sedimentary TOC/CaCO₃ exists in the Southern Ocean, for the last 800 ka, so far¹⁴. This record
is however very precious since Southern Ocean biosphere productivity has a strong potential

for increases in the past while on lower latitudes, the biosphere productivity is already
maximum today¹⁷.

On continents, variations in vegetation cover and type can be related, although
indirectly, to variations of the biosphere terrestrial productivity through the use of dynamic
vegetation models modelling both biome and associated productivity⁹. Pollen counting and
sedimentary TOC are thus useful for biosphere productivity reconstruction^{18,19} but they are
unfortunately indirect and rely on the use of biosphere models. Moreover, similarly to oceanic
records, these observations provide regional records that are not easy to use for documenting
the past global carbon cycle. Ciais et al. (2012)⁸ proposed to use the isotopic composition of
oxygen of atmosphere ($\delta^{18}\text{O}_{\text{atm}}$) as a tracer for terrestrial biosphere productivity. However, this
proxy is a complex tracer being influenced by hydrological cycle at first order^{20,21} and its use
as a quantitative tool for productivity reconstruction depends on the exact determination of
associated fractionation factors in the water and biosphere cycle²².

A total estimate of the global biospheric fluxes and their temporal variations can be
obtained more directly from measurements of $\Delta^{17}\text{O}$ of O_2 ($\ln(\delta^{17}\text{O}+1)-0.516*\ln(\delta^{18}\text{O}+1)$) in ice
cores^{23,24}. This method provides O_2 fluxes and the conversion from O_2 to CO_2 fluxes can be
done from the stoichiometry of the biological processes of photosynthesis and respiration²⁵.
$\Delta^{17}\text{O}$ of O_2 measures the variation of the triple isotopic composition of atmospheric O_2 with
respect to modern oxygen so that by definition, $\Delta^{17}\text{O}$ of O_2 is nil today. Previous experimental
studies showed that $\Delta^{17}\text{O}$ of O_2 increases in a closed biospheric system when the exchanges
with the stratosphere are prevented: the biological productivity leads to $\Delta^{17}\text{O}$ of O_2 increase
while photochemical reactions occurring in the stratosphere have the effect of decreasing $\Delta^{17}\text{O}$
of O_2 ²³. Such properties open a new way to infer the relative proportion of oxygen fluxes issued
from biosphere processes and from the exchanges between the stratosphere and the
troposphere²³. For paleoproductivity reconstructions, measurements of the evolution of $\Delta^{17}\text{O}$

of O₂ in ancient air trapped in the Vostok and GISP2 ice cores already provided information on
the evolution of global biosphere productivity, over the last 400 ka²⁴. The results show a
systematic larger productivity, during interglacial than during glacial periods with interglacial
levels remaining close to the current biosphere productivity.

Over the last 9 deglaciations where CO₂ concentration was measured, Termination V,
occurring between 433 and 426 ka on ice core records on the latest AICC2012 chronology²⁶, is
probably the most intriguing. This Termination is framed by the particularly long and strong
glacial Marine Isotopic Stage 12 (MIS 12), followed by the long and warm interglacial MIS 11
(426 to 398 ka on AICC2012). This is the first Termination after the Mid-Brunhes event
marking a fundamental change in the climate system from mild to warm periods with associated
lower to higher CO₂ concentrations, and from larger to smaller ice volumes^{27–30} associated with
higher sea-level than the present³⁰. Termination V is also occurring in a particular orbital
context, i.e. low eccentricity around 400 ka, which is known to have an influence on the carbon
cycle as observed in $\delta^{13}\text{C}$ oceanic records³¹. Surprisingly, the CO₂ level is not exceptionally
high compared to the following warm periods²⁸, and one key to this, is probably to be found in
the biosphere dynamic. On the continents, pollen data^{19,29} suggest a strong and long increase in
terrestrial productivity with probable impacts on terrestrial vs atmospheric C stocks, during
MIS 11. In the Ocean, MIS 11 displays an unusual increase in carbonate storage in low-latitude
neritic³² and high-latitude pelagic environments³³. While it is clearly associated with a major
phase in coral reef expansion³⁴ and a climax in calcareous phytoplankton productivity
respectively³⁵, the impact of this large carbonate production on atmospheric pCO₂ is not
understood. Therefore, the biosphere productivity fluxes during Termination V and MIS 11
need to be investigated.

Here we present the first measurements of the triple isotopic composition of atmospheric
oxygen ($\Delta^{17}\text{O}$ of O₂) over Termination V. Using these measurements and new correction factors

compared to the previous record of Blunier et al., (2012)²⁴, we reconstruct the oxygen fluxes
associated with biosphere productivity over this particularly strong Termination and compare
it with biosphere productivity reconstruction of the 4 youngest Terminations. Ice core $\delta^{18}\text{O}_{\text{atm}}$
(or $\delta^{18}\text{O}$ of O_2) record, and terrestrial and oceanic records related to biosphere productivity are
used to discuss the relative contribution of changes in oceanic vs terrestrial biosphere fluxes to
the atmospheric CO_2 rise during Termination V.

The biosphere productivity over Termination V and the beginning of MIS 11 is found
to be 10-30% higher than productivity over the pre-industrial period, an exceptional value never
encountered over the last 4 interglacial periods. This productivity peak is most probably due to
an increase of the terrestrial productivity during this period favored by a particular context of
low eccentricity leading to long growing summer season, warm temperature at high latitudes of
the northern hemisphere enabling biosphere productivity from terrestrial soils usually frozen
and possibly a relatively slow sea level rise during Termination V enabling productivity from
low latitude emerged continental platforms. We propose that such strong productivity occurring
concomitantly with an exceptional productivity carbonate peak in the marine realm plays a role
in maintaining the CO_2 level at a relatively low level at the beginning of MIS 11.

**RESULTS**

**A unique record of $\Delta^{17}\text{O}$ of O_2 during Termination V and the beginning of MIS 11**

[revised manuscript text omitted]
 water cycle²⁴, photosynthesis⁴² and respiration⁴³. The estimate for triple isotopic
composition of oxygen produced by terrestrial and oceanic biospheres, $\Delta^{17}\text{O}_{\text{bio}}$, was revised

compared from those of Landais et al. (2007)⁴⁰ taking into account recent studies. The final
estimates vary between 180.5 ± 60.5 ppm and 136 ± 44 ppm for the $\Delta^{17}\text{O}_{\text{bio}}$ during the Last
Glacial Maximum and the pre-industrial period respectively, the uncertainties being mainly
associated with uncertainties in the fractionation coefficients for the different biospheric
processes⁴⁰ (Supplementary Table 2). Combining these estimates with the covariation of CO_2
concentration and flux of negative $\Delta^{17}\text{O}$ of O_2 anomaly from the stratosphere, we converted the
full $\Delta^{17}\text{O}$ of O_2 record over the last 445 ka into an estimate for the evolution of the global
oxygen productivity (Figure 3). This reconstruction was also completed with an alternative
reconstruction of global oxygen productivity using an alternative model, the one of Blunier et
al. (2012)²⁴, forced by our $\Delta^{17}\text{O}$ of O_2 data (Supplementary Figure 2).

Combining our different sensitivity studies, we find that the global productivity is reduced by
60-90% during glacial periods compared to interglacials. At the beginning of most interglacial
periods, the global productivity remains close to the preindustrial level. The only exception to
this general behavior is the strong oxygen biosphere productivity at the end of Termination V
reaching values 10 to 30% higher than during the pre-industrial period.

DISCUSSION

Complementary information on the origin and specificity of the $\Delta^{17}\text{O}$ of O_2 signal over
Termination V can be obtained from the ice core $\delta^{18}\text{O}_{\text{atm}}$ (or $\delta^{18}\text{O}$ of O_2) record over the last
800 ka²¹ (Figure 3). $\delta^{18}\text{O}_{\text{atm}}$ is a complex parameter resulting from both biosphere productivity
and low latitude water cycle⁴⁴. In particular, it has been shown that Weak Monsoon Intervals
observed during Heinrich events lead to increases in $\delta^{18}\text{O}_{\text{atm}}$ via changes in the low latitude
water cycle⁴⁵. Another way to increase the $\delta^{18}\text{O}_{\text{atm}}$ is to increase the ratio between terrestrial
and oceanic biosphere productivity^{8,44}.

[revised manuscript text omitted]
^{18}\text{O}_{\text{atm}}$, $\delta^{17}\text{O}_{\text{atm}}$, $\delta\text{O}_2/\text{Ar}$ measurements and $\Delta^{17}\text{O}$ of O₂ calculation, respectively.

424

425 **Corrections on $\Delta^{17}\text{O}$ of O₂**

426 **Atmospheric air calibration.** Every day, $\delta^{18}\text{O}$, $\delta^{17}\text{O}$ and $\delta\text{O}_2/\text{Ar}$ of atmospheric air are
427 measured and are then used to calibrate our measurements following:

$$428 \quad \delta^{18}\text{O}_{\text{ext air corr}} = \left[\frac{(\delta^{18}\text{O}_{\text{sample}}/1000) + 1}{(\delta^{18}\text{O}_{\text{ext air}}/1000) + 1} - 1 \right] * 1000$$

$$429 \quad \delta^{17}\text{O}_{\text{ext air corr}} = \left[\frac{(\delta^{17}\text{O}_{\text{sample}}/1000) + 1}{(\delta^{17}\text{O}_{\text{ext air}}/1000) + 1} - 1 \right] * 1000$$

430 The $\delta^{18}\text{O}_{\text{ext air}}$ and $\delta^{17}\text{O}_{\text{ext air}}$ were constant during the two measurement periods so that we used
431 the average values over the two corresponding periods to correct the raw data. The daily
432 correction was the same every day during each period.

433

434 **Correction due to fractionation in the firn column**

435

436 - **Gravitational fractionation**

437 Gravitational fractionation operates in firn due to Earth gravity field. $\delta^{18}\text{O}$ and $\delta^{17}\text{O}$ were
438 obtained by corrections for this diffusive process using $\delta^{15}\text{N}$ in neighbouring samples⁶⁷. The
439 correction applied depends on the difference of mass between the two isotopes considered so
440 that:

441

$$442 \quad \delta^{18}\text{O}_{\text{gravitational corr}} = \delta^{18}\text{O}_{\text{measured}} - 2 * \delta^{15}\text{N}$$

443

$$444 \quad \delta^{17}\text{O}_{\text{gravitational corr}} = \delta^{17}\text{O}_{\text{measured}} - 1 * \delta^{15}\text{N}$$

445

446 - **Thermal fractionation**

In the firn column, diffusive processes operate due to changes in temperature or because of the
Earth gravity. These processes lead to isotopic fractionation of O_2 that was taken into account
for $\delta^{18}\text{O}_{\text{atm}}$ reconstruction in the NGRIP ice core over abrupt temperature changes of the last
glacial period⁶⁸.

To check the effect of thermal fractionation on $\Delta^{17}\text{O}$ of O_2 , we performed measurements of
$\Delta^{17}\text{O}$ of O_2 over the top 20 m of the EastGRIP firn where a strong seasonal gradient is present.
However, the resulting $\Delta^{17}\text{O}$ of O_2 was not showing any significant deviation from the
atmospheric value hence suggesting that thermal fractionation does not modify the $\Delta^{17}\text{O}$ of O_2
of the atmosphere in the firn column. Moreover, in Antarctica, surface temperature variations
are much lower than in Greenland during deglaciations or climatic variability of the last glacial
period so that thermal fractionation is not expected to have a significant effect on the isotopic
composition of trapped oxygen.

- **Pore close-off effect**

Pore close-off at the bottom of the firn has been shown to affect $\delta\text{O}_2/\text{N}_2$, $\delta\text{Ar}/\text{N}_2$ with potential
effects on $\delta^{15}\text{N}$ and $\delta^{40}\text{Ar}$ in certain cases⁶⁹. We checked this possible effect on $\Delta^{17}\text{O}$ of O_2 by
comparing $\Delta^{17}\text{O}$ of O_2 in bubbly ice at the top of the NEEM ice core. After correction of
gravitational effect, we found a systematic enrichment of 13 per meg which could potentially
bias the reconstruction of atmospheric $\Delta^{17}\text{O}$ of O_2 from $\Delta^{17}\text{O}$ of O_2 in trapped air.

**Gas loss correction.** During storage, O_2 in ice samples is subject to gas loss fractionation due
to diffusion processes⁷⁰ and O_2/N_2 ratio is always lower by several % in ice samples stored
several years at -20°C than ice samples stored at -50°C ⁷¹. Such gas loss effect is also associated
with isotopic fractionation of oxygen, $\delta^{18}\text{O}_{\text{atm}}$ trapped in the ice being higher when $\delta\text{O}_2/\text{N}_2$
decreases with a slope for the relationship of -0.01 ($\delta^{18}\text{O}_{\text{atm}}$ vs $\delta\text{O}_2/\text{N}_2$)^{20,21,72,73}. We thus expect
that $\delta^{17}\text{O}$ can also be affected by this gas loss process and that it may create an anomaly of $\Delta^{17}\text{O}$
of O_2 . To check this effect, measurements of $\delta^{17}\text{O}$, $\delta^{18}\text{O}_{\text{atm}}$, $\Delta^{17}\text{O}$ of O_2 and $\delta\text{O}_2/\text{Ar}$ have been
performed on 3 samples of GRIP ice core (clathrate ice stored during more than 20 years at -

20°C). Each of the 3 ice samples have been cut in order to analyse the interior and the exterior
of the sample separately. $\delta O_2/N_2$ measurements could not be performed on exactly the same
samples so that we used the $\delta O_2/Ar$ measurements to estimate the amount of oxygen loss. Argon
is also known to be affected by gas loss^{20,70} but in a smaller extend than oxygen⁷⁴ so that the
decrease of $\delta O_2/Ar$ is still a good indication of larger gas loss.

Data show a systematic lower $\Delta^{17}O$ of O_2 value in the “exterior” sample compared to the
“interior” sample, paralleling the lower $\delta O_2/Ar$ value in the “exterior” sample compared to the
“interior” sample as expected by gas loss (Table S1). This systematic relationship and the $\Delta^{17}O$
of O_2 and $\delta O_2/Ar$ values can be used to propose a correction for the $\Delta^{17}O$ of O_2 that takes into
account the gas loss effect:

$$485 \quad \Delta^{17}O_{gas\ loss\ corr} = \Delta^{17}O_{sample} - 0.3945 * [(\delta O_2/Ar)_{sample} - (\delta O_2/Ar)_{std}]$$

**Comparison of EDC $\Delta^{17}O$ of O_2 with previous $\Delta^{17}O$ of O_2 record.** Previous $\Delta^{17}O$ of O_2
measurements only took into account correction linked with gravitational fractionation. As a
consequence, we checked the consistency of previous records performed on the GISP2 and
Vostok ice core with our new EDC data by measuring again $\Delta^{17}O$ of O_2 over Termination 2
using EDC samples and integrating the aforementioned correction (Figure 1).

**Calculation of the flux of oxygen associated with biosphere productivity.** We follow here
the calculation of the flux of oxygen associated with biosphere productivity described in
Landais et al. (2007)⁴⁰.

Since the atmospheric $\Delta^{17}\text{O}$ of O_2 ($\Delta^{17}\text{O}_{\text{atm}}$) is influenced by the exchanges with biosphere and
 stratosphere, it is possible to write the following equation in a 3-box system at equilibrium
 (biosphere – bio -, atmosphere – atm-, stratosphere – strat-):

$$500 \quad F_{\text{bio}} * (\Delta^{17}\text{O}_{\text{bio}} - \Delta^{17}\text{O}_{\text{atm}}) = F_{\text{strat}} * (\Delta^{17}\text{O}_{\text{strat}} - \Delta^{17}\text{O}_{\text{atm}})$$

Where F_{bio} is the flux of oxygen exchanged by photosynthesis and respiration (both fluxes being
 considered at equilibrium) and F_{strat} is the flux of oxygen exchanged between the lower
 atmosphere (or troposphere) and the stratosphere (fluxes in and out of the stratosphere are
 assumed equal).

To reconstruct past biospheric fluxes between terrestrial/oceanic biosphere and atmosphere
 based on $\Delta^{17}\text{O}$ of O_2 , it is necessary to know the evolution of the stratospheric flux as well as
 of the $\Delta^{17}\text{O}_{\text{strat}}$. Luz et al (1999)²³ and Blunier et al. (2012)²⁴ showed that a good assumption is
 to consider that the production rate of depleted O_2 in the stratosphere can be related to the
 atmospheric CO_2 concentration. It is then possible to express the evolution of biosphere oxygen
 flux in the past compared to pre-industrial from the following equation:

$$514 \quad \frac{F_{\text{bio},t}}{F_{\text{bio},\text{pre-industrial}}} = \frac{(\text{CO}_2)_t}{(\text{CO}_2)_{\text{pre-industrial}}} * \frac{\Delta^{17}\text{O}_{\text{bio},\text{pre-industrial}}}{\Delta^{17}\text{O}_{\text{bio},t} - \Delta^{17}\text{O}_{\text{atm},t}} \quad (\text{eq S1})$$

Where $F_{\text{bio},t}$ is the biosphere oxygen flux at a given time, $F_{\text{bio},\text{pre-industrial}}$ is the present time
 biosphere oxygen flux, $(\text{CO}_2)_t$ and $(\text{CO}_2)_{\text{pre-industrial}}$ correspond to the atmospheric CO_2

concentration at a given time and at the pre-industrial period respectively, $\Delta^{17}\text{O}_{\text{bio,pre-industrial}}$ and
$\Delta^{17}\text{O}_{\text{bio,t}}$ are the values of $\Delta^{17}\text{O}$ of O_2 in an atmosphere that would only be influenced by
exchanges with the biosphere at the present time and at a given time respectively. $\Delta^{17}\text{O}_{\text{atm,t}}$ is
the $\Delta^{17}\text{O}$ of O_2 of the atmosphere at a given time measured in the air trapped in ice core. We
refer to pre-industrial period because of the long residence time of oxygen in the atmosphere (>
1000 years), $\Delta^{17}\text{O}$ of O_2 isn't exhibiting any significant variation over the last centuries.

$\Delta^{17}\text{O}_{\text{bio}}$ can be calculated from the fractionation coefficients associated with the different
processes leading to oxygen fluxes in the biosphere (mainly photosynthesis, dark respiration
and photorespiration). Detailed calculations of $\Delta^{17}\text{O}_{\text{bio}}$ at the LGM and pre-industrial periods
were obtained in Landais et al. (2007)⁴⁰ taking into account uncertainties in the determination
of the fractionation coefficients⁴¹ as well as on the relative fluxes of oxygen (Supplementary
Table 2).

From the calculation of $\Delta^{17}\text{O}_{\text{bio, pre-industrial}}$ and $\Delta^{17}\text{O}_{\text{bio,LGM}}$ (Supplementary Table 2), $\Delta^{17}\text{O}_{\text{bio,t}}$ is
calculated through a scaling on the variations of CO_2 concentration between pre-industrial
period (280 ppmv) and the LGM (190 ppmv)⁷⁵ such as:

$$535 \quad \Delta^{17}\text{O}_{\text{bio,t}} = \Delta^{17}\text{O}_{\text{bio,pre-industrial}} + (\Delta^{17}\text{O}_{\text{bio,LGM}} - \Delta^{17}\text{O}_{\text{bio,pre-industrial}}) * \left(\frac{280 - (\text{CO}_2)_t}{90} \right) \text{ (eq S2)}$$

The reconstructed biosphere productivity using the Landais et al. (2007) formulation is
displayed on Figure 2: it leads to a biosphere productivity at the beginning of MIS 11 of 20%
higher than during MIS 1. Since the publication of Landais et al. (2007), new estimates of
fractionation coefficients within the oxygen cycle are available^{22,43,76} and influence of the water

cycle organisation has also been suggested²⁴. As a consequence, we performed 3 types of
sensitivity tests to address the influences of such new determination and associated uncertainty
ranges on our results.

- **Uncertainty in photosynthesis fractionation.** Some marine species show fractionation
during marine photosynthesis⁴² while it has long been assumed that photosynthesis does
not fractionate⁷⁷. We performed sensitivity tests to compare biosphere productivity
reconstructions without any fractionation and reconstruction with the different
fractionation effects during marine photosynthesis as observed in Eisenstadt et al.
(2010)⁴². The largest change on the reconstructed biosphere productivity is obtained
using the observed photosynthesis fractionation effect associated with *Emiliana huxleyi*
($\delta^{18}\text{O}=5.81\text{‰}$ and slope between $\ln(\delta^{17}\text{O}+1)$ and $\ln(\delta^{18}\text{O}+1)$ of 0.5253). The
reconstructed global biosphere productivity is lower with such fractionation effect: The
MIS 11 productivity level is decreased by 5% with such fractionation effect compared
to the reconstruction with no fractionation at photosynthesis (Supplementary Figure 2).

- **Uncertainty in respiration fractionation.** Recent studies have highlighted large
variations in the relationship between $\delta^{17}\text{O}$ and $\delta^{18}\text{O}$ during respiration linked to
temperature variations⁴³. When taking into account the maximum effect, i.e. a decrease
of 0.005 for the slope of the relationship between $\ln(\delta^{17}\text{O}+1)$ and $\ln(\delta^{18}\text{O}+1)$ during
respiration, we end up with a decrease of $\Delta^{17}\text{O}_{\text{bio}}$ by 65 ppm. The resulting biospheric
productivity reconstruction is about 10% higher during MIS 11 than the initial
reconstruction leading to an anomalously high O_2 flux associated with gross primary
productivity during Termination V.

- **Uncertainty in the fractionation within water cycle.** The general assumption is that
the relationship between $\delta^{17}\text{O}$ and $\delta^{18}\text{O}$ (meteoric water line) remains the same over
glacial – interglacial cycles. However, measurements Vostok ice core have shown that

the ^{17}O -excess, defined as $^{17}\text{O}\text{-excess} = \ln(1+\delta^{17}\text{O}) - 0.528 \cdot \ln(1+\delta^{18}\text{O})$, is lower by up to
20 ppm during the last glacial maximum⁷⁸. While it has been shown that this is a local
effect⁷⁹, we still performed a sensitivity test with varying ^{17}O -excess of all continental
meteoric waters in a range of 20 ppm between glacial and interglacial periods (lower
570 ^{17}O -excess values for glacial than interglacial periods). Taking into account such
variations, the reconstructed biospheric productivity does not differ by more than 4%
from the case without change in meteoric water ^{17}O -excess.

An additional validation of our calculation comes from the comparison of our reconstruction
with the oxygen biosphere productivity variations obtained by Blunier et al. (2012)²⁴ over the
last 4 terminations using a different vegetation model and different formulation of the link
between atmospheric CO_2 and flux of $\Delta^{17}\text{O}$ of O_2 anomaly from the stratosphere. Indeed, in our
approach, we directly dealt with $\Delta^{17}\text{O}$ of O_2 anomaly in the box model while it may be more
appropriate to deal with the $\delta^{17}\text{O}$ and $\delta^{18}\text{O}$ of O_2 values as done in Blunier et al. (2012)²⁴ and
suggested by Prokopenko et al. (2011)⁸⁰. We thus compared our reconstruction with the one
performed with the model of Blunier et al., 2012²⁴ (Supplementary Figure 2). The smaller
difference in the oxygen flux associated with oxygen productivity between glacial and
interglacial periods in Blunier et al. (2012)²⁴ mainly comes from the different formulations of
the dependency of the stratospheric anomaly $\Delta^{17}\text{O}$ of O_2 to CO_2 atmospheric concentration.
Another difference comes from the variations in the triple isotopic composition of oxygen in
water between glacial and interglacial periods not taken into account in Landais et al. (2007)⁴⁰.
We took this possible variation into account in our approach through the third sensitivity test
explained above. We ran here the model of Blunier et al. (2012)²⁴ over the new Termination V
data. The comparison between the two methods of reconstruction (Landais et al., 2007⁴⁰ vs
Blunier et al., 2012²⁴) is significant. The biosphere productivity increase at the end of
Termination V is less important using the Blunier et al. (2012)²⁴ approach than using the

Landais et al. (2007)⁴⁰: 10% larger than for other interglacial periods following Blunier et al.
(2012)²⁴ instead of 20% following Landais et al. (2007)⁴⁰. Still, both approaches show that the
biospheric productivity increase over Termination V is exceptional compared to the younger
terminations.

References

- 1. Lüthi, D. *et al.* High-resolution carbon dioxide concentration record 650,000–800,000 years
before present. *Nature* **453**, 379–382 (2008).
- 2. Menviel, L. *et al.* Southern Hemisphere westerlies as a driver of the early deglacial atmospheric
CO₂ rise. *Nature Communications* **9**, (2018).
- 3. Martínez-García, A. *et al.* Links between iron supply, marine productivity, sea surface
temperature, and CO₂ over the last 1.1 Ma. *Paleoceanography* **24**, PA1207 (2009).
- 4. Stephens, B. B. & Keeling, R. F. The influence of Antarctic sea ice on glacial-interglacial CO₂
variations. *Nature* **404**, 171–175 (2000).
- 5. Volk, T. & Hoffert, M. I. Ocean Carbon Pumps: Analysis of Relative Strengths and Efficiencies in
Ocean-Driven Atmospheric CO₂ Changes. in *Geophysical Monograph Series* (eds. Sundquist, E. T.
& Broecker, W. S.) 99–110 (1985).
- 6. Tagliabue, A. *et al.* Quantifying the roles of ocean circulation and biogeochemistry in governing
ocean carbon-13 and atmospheric carbon dioxide at the last glacial maximum. *Climate of the*
*Past* **5**, 695–706 (2009).
- 7. Toggweiler, J. R. Variation of atmospheric CO₂ by ventilation of the ocean's deepest water.
*Paleoceanography* **14**, 571–588 (1999).
- 8. Ciais, P. *et al.* Large inert carbon pool in the terrestrial biosphere during the Last Glacial
Maximum. *Nature Geoscience* **5**, 74–79 (2012).

- 9. Prentice, I. C., Harrison, S. P. & Bartlein, P. J. Global vegetation and terrestrial carbon cycle
changes after the last ice age. *New Phytologist* **189**, 988–998 (2011).
- 10. Crichton, K. A., Bouttes, N., Roche, D. M., Chappellaz, J. & Krinner, G. Permafrost carbon as a
missing link to explain CO₂ changes during the last deglaciation. *Nature Geoscience* **9**, 683–686
(2016).
- 11. Peterson, C. D., Lisiecki, L. E. & Stern, J. V. Deglacial whole-ocean $\delta^{13}\text{C}$ change estimated from
480 benthic foraminiferal records. *Paleoceanography* **29**, 549–563 (2014).
- 12. Eggleston, S., Schmitt, J., Bereiter, B., Schneider, R. & Fischer, H. Evolution of the stable carbon
isotope composition of atmospheric CO₂ over the last glacial cycle. *Paleoceanography* **31**, 434–
452 (2016).
- 13. Schmitt, J. *et al.* Carbon Isotope Constraints on the Deglacial CO₂ Rise from Ice Cores. *Science*
**336**, 711–714 (2012).
- 14. Diekmann, B. & Kuhn, G. Sedimentary record of the mid-Pleistocene climate transition in the
southeastern South Atlantic (ODP Site 1090). *Palaeogeography, Palaeoclimatology,*
*Palaeoecology* **182**, 241–258 (2002).
- 15. Salter, I. *et al.* Carbonate counter pump stimulated by natural iron fertilization in the Polar
Frontal Zone. *Nature Geoscience* **7**, 885–889 (2014).
- 16. Duchamp-Alphonse, S. *et al.* Enhanced ocean-atmosphere carbon partitioning via the carbonate
counter pump during the last deglacial. *Nature Communications* **9**, (2018).
- 17. Hain, M. P., Sigmal, D. & Haug, G. H. 8.18–The biological Pump in the Past. in *Treatise on*
*Geochemistry* vol. 8 485–517 (2014).
- 18. Tzedakis, P. C., Hooghiemstra, H. & Pälike, H. The last 1.35 million years at Tenaghi Philippon:
revised chronostratigraphy and long-term vegetation trends. *Quaternary Science Reviews* **25**,
3416–3430 (2006).
- 19. Melles, M. *et al.* 2.8 Million Years of Arctic Climate Change from Lake El’gygytyn, NE Russia.
*Science* **337**, 315–320 (2012).

- 20. Severinghaus, J. P., Beaudette, R., Headly, M. A., Taylor, K. & Brook, E. J. Oxygen-18 of O₂ Records
the Impact of Abrupt Climate Change on the Terrestrial Biosphere. *Science* **324**, 1431–1434
(2009).
- 21. Extier, T. *et al.* On the use of δ 18 O atm for ice core dating. *Quaternary Science Reviews* **185**,
244–257 (2018).
- 22. Luz, B. & Barkan, E. The isotopic composition of atmospheric oxygen. *Global Biogeochemical*
*Cycles* **25**, GB3001 (2011).
- 23. Luz, B., Barkan, E., Bender, M. L., Thiemens, M. H. & Boering, K. A. Triple-isotope composition of
atmospheric oxygen as a tracer of biosphere productivity. *Nature* **400**, 547–550 (1999).
- 24. Blunier, T., Bender, M. L., Barnett, B. & von Fischer, J. C. Planetary fertility during the past 400 ka
based on the triple isotope composition of O₂ in trapped gases from the Vostok ice core. *Climate*
*of the Past* **8**, 1509–1526 (2012).
- 25. Hoffmann, G. *et al.* A model of the Earth's Dole effect. *Global Biogeochemical Cycles* **18**, GB1008
(2004).
- 26. Bazin, L. *et al.* An optimized multi-proxy, multi-site Antarctic ice and gas orbital chronology
(AICC2012): 120–800 ka. *Clim. Past* **9**, 1715–1731 (2013).
- 27. Jouzel, J. *et al.* Orbital and Millennial Antarctic Climate Variability over the Past 800,000 Years.
*Science* **317**, 793–796 (2007).
- 28. Siegenthaler, U. Stable Carbon Cycle-Climate Relationship During the Late Pleistocene. *Science*
**310**, 1313–1317 (2005).
- 29. de Vernal, A. & Hillaire-Marcel, C. Natural Variability of Greenland Climate, Vegetation, and Ice
Volume During the Past Million Years. *Science* **320**, 1622–1625 (2008).
- 30. Raymo, M. E. & Mitrovica, J. X. Collapse of polar ice sheets during the stage 11 interglacial.
*Nature* **483**, 453–456 (2012).

[revised manuscript text omitted]

SUPPLEMENTARY INFORMATION

**Supplementary Figure 1. Relationship between CO_2^1 and $\Delta^{17}\text{O}$ of O_2^2 and this study for Termination I to**

**V. The slope of the $\Delta^{17}\text{O}$ of O_2 vs CO_2 anti-correlation is much lower over Termination V than for**

**Terminations I-IV. The correlation coefficient is also lower for Termination V than for younger**

**Terminations.**

**Supplementary Figure 2. Biosphere productivity reconstructions over the last 450 ka. All the lines**

**represent the ratio of global biospheric productivity between the time considered and pre-industrial.**

**The dark grey area represents the ratio between global biosphere productivity and pre-industrial**

**biosphere productivity with uncertainty bars deduced from uncertainties in the value of $\Delta^{17}\text{O}$ of O_2**

**produced by the earth biosphere (Landais et al., 2007)³. The solid and dotted green lines represent**

**respectively the maximum and minimum biosphere productivity calculated with a lower ^{17}O associated**

**with oceanic and terrestrial productivity, taking into account the uncertainty in respiration**

**fractionation⁴ (See Method and Supplementary Table 2). The solid and dotted blue lines represent**

**respectively the maximum and minimum biosphere productivity calculated with a higher ^{17}O**

**associated with oceanic productivity, taking into account the uncertainty in photosynthesis**

**fractionation⁵ (See Method and Supplementary Table 2)), The solid black line is the ratio of biosphere**

**productivity calculated for a constant H_2O anomaly of 40 per meg over the time period considered,**

**reconstructed with the model of Blunier et al., 2012², the dotted black line is the ratio of biosphere**

**productivity for an anomaly derived from δD , reconstructed with the model of Blunier et al., 2012².**

**The light grey area represents the maximum uncertainty range of the biosphere productivity**

**reconstruction, taking into account the reconstructions of biosphere productivity described above as**

**well as the biosphere productivity of Blunier et al., 2012².**

**Supplementary Figure 3. Variation of the oxygen biosphere productivity compared with pollen and**
 **oceanic records. a** Eccentricity⁶; **b** atmospheric CO₂¹; **c** global oxygen biosphere productivity (this
 **study); d** Tree, shrubs and Picea pollen record (%) from El'Gygytyn Lake⁷; **e** Si/Ti ratio from El'Gygytyn
 **Lake⁷, a proxy of biogenic silica normalized to detrital, reflecting the changes in diatom productivity in**
 **the lake; f** Pollen record from oceanic core ODP 646⁸ ; **g** Arboreal and Quercus robur pollen records (%)
 **from Lake Ohrid, Balkan Peninsula^{9,10}; h** Arboreal pollen record (%) from Tenaghi Philippon Lake¹¹; **i**
 **alkenone mass accumulation rate (MAR) (µg.m⁻².y⁻¹) and TOC MAR (mg.m⁻².y⁻¹) records at Site PS2489-**
 **2/ODP1090¹²; j** CaCO₃ (%) record from Site PS2489-2/ODP1090, Atlantic sector of the Southern

Ocean¹³; k TOC/CaCO₃ ratio at Site PS2489-2/ODP1090, Atlantic Southern Ocean¹³. The grey shadow
bars represent the period of rapid increase in CO₂ during deglaciations.

Supplementary Figure 3 combines well-dated palynological and geochemical data covering the last five
deglaciations and related to terrestrial and oceanic biological productivities, respectively. El'Gygytgyn
core age model is based, in first order, on magnetostratigraphy and in second and third orders on the
correlation between sedimentary proxy data to the LR04 stack¹⁴ and insolation patterns⁶. The
chronology of marine core ODP 646 is based on the $\delta^{18}\text{O}$ in *N. pachyderma* and on the correlation with
the stack LR04 of Lisiecki and Raymo¹⁴. The age-model of Lake Ohrid record^{9,10} is based on
tephrochronology on 11 tephra layers¹⁵ and on a second order on the tuning of biogeochemical proxy
data to orbital parameters¹⁶. The Tenaghi Philippon core age-model is based on the correlation
between vegetation changes and March and June perihelion configuration. The age-models of PS2489-
2 and ODP Site 1090 were calculated using the correlation between the alkenone-based SST with the
EDC ice core temperature record using EDC3 chronology^{12,17,18}. Decreases in TOC and alkenone MARS
from PS2489-2/ODP 1090 are correlated, indicating that TOC decrease is not a consequence of the
increase in CaCO₃ in the sediment.

Sample	Interior sample		Exterior sample	
	$\delta O_2/Ar$	$\Delta^{17}O$ of O_2	$\delta O_2/Ar$	$\Delta^{17}O$ of O_2
GRIP Sample 1	-74	29	-146	13
GRIP Sample 2	-82	50	-134	13
GRIP Sample 3	-94	50	-156	26

852

853 **Supplementary Table 1. Comparison of $\delta O_2/Ar$ and $\Delta^{17}O$ of O_2 values between the interior and the**

854 **exterior part of the ice core**

855

$F_{\text{oce,pre-industrial}}/F_{\text{terr,pre-industrial}}$	$\Delta^{17}\text{O}_{\text{bio,pre-industrial}}$	$F_{\text{oce,LGM}}/F_{\text{terr,LGM}}$	$\Delta^{17}\text{O}_{\text{bio,LGM}}$
0.45	124 ^a	0.56	156
		1.08	178
	182 ^a	0.56	211
		1.08	227
	92 ^b	0.56	120
		1.08	134
169 ^c	0.56	202	
	1.08	230	
0.59	145 ^a	0.73	177
		1.41	205
	189 ^a	0.73	217
		1.41	234
	98 ^b	0.73	126
		1.41	140
180 ^c	0.73	213	
	1.41	241	

**Supplementary Table 2. Estimates of $\Delta^{17}\text{O}_{\text{bio}}$ for pre-industrial and Last Glacial Maximum (LGM) from**

**Landais et al., 2007¹⁴ and from sensitivity tests (see Method). The different $\Delta^{17}\text{O}_{\text{bio}}$ values arise from**

**uncertainties in the determination of the fractionation coefficients as well as uncertainties in the ratio**

**of oceanic vs terrestrial biospheric fluxes at present-day and at LGM as given in this table.**

**The ratio of oceanic flux over terrestrial flux for pre-industrial and LGM are taken from Landais et al.,**

**2007.**

a. $\Delta^{17}\text{O}_{\text{bio}}$ estimates from Landais et al., 2007

b. $\Delta^{17}\text{O}_{\text{bio}}$ estimates calculated based on the maximum fractionation effect during respiration, i.e.

a decrease of 0.005 for the slope of the relationship between $\delta^{17}\text{O}$ and $\delta^{18}\text{O}$ during respiration

(Stolper et al., 2018)

c. $\Delta^{17}\text{O}_{\text{bio}}$ estimates calculated considering the maximum photosynthesis fractionation effect

observed in Eisenstadt et al., 2010 (Emiliana huxleyi)

**Supplementary references**

- 1. Siegenthaler, U. Stable Carbon Cycle–Climate Relationship During the Late Pleistocene. *Science*
310, 1313–1317 (2005).
- 2. Blunier, T., Bender, M. L., Barnett, B. & von Fischer, J. C. Planetary fertility during the past 400
873 ka based on the triple isotope composition of O₂ in trapped gases from the Vostok ice core. *Climate of*
874 *the Past* 8, 1509–1526 (2012).
- 3. Landais, A., Lathiere, J., Barkan, E. & Luz, B. Reconsidering the change in global biosphere
productivity between the Last Glacial Maximum and present day from the triple oxygen isotopic
composition of air trapped in ice cores. *Global Biogeochemical Cycles* 21, GB1025 (2007).
- 4. Stolper, D. A., Fischer, W. W. & Bender, M. L. Effects of temperature and carbon source on
the isotopic fractionations associated with O₂ respiration for ¹⁷O/¹⁶O and ¹⁸O/¹⁶O ratios in *E. coli*.
*Geochimica et Cosmochimica Acta* 240, 152–172 (2018).
- 5. Eisenstadt, D., Barkan, E., Luz, B. & Kaplan, A. Enrichment of oxygen heavy isotopes during
photosynthesis in phytoplankton. *Photosynthesis Research* 103, 97–103 (2010).
- 6. Laskar, J. et al. A long-term numerical solution for the insolation quantities of the Earth.
*Astronomy & Astrophysics* 428, 261–285 (2004).
- 7. Melles, M. et al. 2.8 Million Years of Arctic Climate Change from Lake El’gygytgyn, NE Russia.
*Science* 337, 315–320 (2012).
- 8. de Vernal, A. & Hillaire-Marcel, C. Natural Variability of Greenland Climate, Vegetation, and
Ice Volume During the Past Million Years. *Science* 320, 1622–1625 (2008).
- 9. Sadori, L. et al. Pollen-based paleoenvironmental and paleoclimatic change at Lake Ohrid
(south-eastern Europe) during the past 500 ka. *Biogeosciences* 13, 1423–1437 (2016).
- 10. Kousis, I. et al. Centennial-scale vegetation dynamics and climate variability in SE Europe during
Marine Isotope Stage 11 based on a pollen record from Lake Ohrid. *Quaternary Science Reviews* 190,
20–38 (2018).
- 11. Tzedakis, P. C., Hooghiemstra, H. & Pälike, H. The last 1.35 million years at Tenaghi Philippon:
revised chronostratigraphy and long-term vegetation trends. *Quaternary Science Reviews* 25, 3416–
3430 (2006).
- 12. Martínez-García, A. et al. Links between iron supply, marine productivity, sea surface
temperature, and CO₂ over the last 1.1 Ma. *Paleoceanography* 24, PA1207 (2009).
- 13. Diekmann, B. & Kuhn, G. Sedimentary record of the mid-Pleistocene climate transition in the
southeastern South Atlantic (ODP Site 1090). *Palaeogeography, Palaeoclimatology, Palaeoecology* 182,
241–258 (2002).
- 14. Lisiecki, L. E. & Raymo, M. E. A Pliocene–Pleistocene stack of 57 globally distributed benthic δ
¹⁸O records. *Paleoceanography* 20, PA1003 (2005).

- 15. Leicher, N. et al. First tephrostratigraphic results of the DEEP site record from Lake Ohrid
(Macedonia and Albania). *Biogeosciences* 13, 2151–2178 (2016).
- 16. Francke, A. et al. Sedimentological processes and environmental variability at Lake Ohrid
(Macedonia, Albania) between 637 ka and the present. *Biogeosciences* 13, 1179–1196 (2016).
- 17. Jouzel, J. et al. Orbital and Millennial Antarctic Climate Variability over the Past 800,000
Years. *Science* **317**, 793–796 (2007).
- 18. Parrenin, F. et al. The EDC3 chronology for the EPICA Dome C ice core. *Clim. Past* 13 (2007).

Reviewers' comments:

Reviewer #1 (Remarks to the Author):

The authors replied to individual comments and revised their manuscript correctly. I do not have further comment on their revised manuscript.

Reviewer #2 (Remarks to the Author):

The revisions satisfactorily address my concerns from the first version of the manuscript.

I will point out, however, that the phrase (L241-242) "global productivity is reduced by 60-90% during glacial periods" is misleading. As written, the statement suggests that glacial productivity was 10-40% interglacial productivity; instead, what the study infers is that glacial productivity was 60-90% that of interglacials, i.e., that glacial productivity is reduced by 10-40%.

Figure R2 is indeed useful to have in the supplement. I recommend including it.

Reviewer #3 (Remarks to the Author):

I've gone through the reviewers' comments and authors' responses in detail, and attempted to understand the various estimates of GPP calculated in this paper. I regret that I have not been able to understand all the critical points but I hope my limitation will not cause me to make a wrong recommendation.

The new data in this paper are interesting. My basic concern has been the following. Any change in global productivity will almost certainly be linked to a change in the water cycle (with its large fractionations) and the ratio of terrestrial to marine production. Therefore, the question arises as to whether the decoupling between $\Delta^{17}O$ and CO_2 , observed at Termination 5, can be the consequence of a change in isotope fractionation associated with the hydrologic cycle or the ratio of terrestrial/marine production, rather than a change in global productivity.

In supplementary fig. 2, the authors vary properties associated with respiratory fractionation, photosynthetic fractionation, and other processes. None of these variations change the conclusion that global productivity was highest at Termination 5 or Stage 11. I believe that this plot shows that changing a property uniformly throughout the record leaves the productivity maximum at Stage 11 intact.

I am not sure if I understand supplementary table 2. If I understand correctly, this table indicates that you calculate very different values of gross production depending on your choice of the assumed ratio of terrestrial to marine O_2 production, and fractionation factors. If so, then how do you know that the decoupling at Stage 11 is due to exceptional productivity, or changes in terrestrial/marine production and various isotope effects?

I need to say that I do not understand supplementary table 2. I could not figure out where the $\Delta^{17}O_{bio}$ estimates come from in line 232 of the text, which refers to supplementary table 2. I cannot find matching numbers in the table, nor do averages match between the numbers in the text and the numbers in the supplemental table. In line 242, I am surprised by the statement that ocean productivity can be reduced by as much as 90%. This number is hard to understand, since total productivity is only suppressed by 40% or less during glacial times.

Before I could endorse this paper for publication, I would need to see a discussion of the possibility that the Stage 11 anomaly is due to differences, at Stage 11, of properties listed above, with respect to the other interglacials.

Reviewer #2 (Remarks to the Author):

The revisions satisfactorily address my concerns from the first version of the manuscript.

I will point out, however, that the phrase (L241-242) "global productivity is reduced by 60-90% during glacial periods" is misleading. As written, the statement suggests that glacial productivity was 10-40% interglacial productivity; instead, what the study infers is that glacial productivity was 60-90% that of interglacials, i.e., that glacial productivity is reduced by 10-40%.

>> We thank Reviewer #2 for pointing out this misunderstanding. The meaning of the sentence has been modified to correspond to the study (l. 243-244 of the new manuscript)

„Combining our different sensitivity studies, we find that the global productivity is reduced by 10 to 40% during glacial periods compared to interglacials.”

Figure R2 is indeed useful to have in the supplement. I recommend including it.

>> We take under consideration the recommendation of Reviewer #2 and we added Figure R2 in the Supplement (l. 901 to 910).

„Supplementary Figure 4. Agreement between reconstructions of biospheric productivity fluxes of oxygen over the last deglaciation from $\Delta^{17}\text{O}$ of O_2 and output of coupled model equipped with vegetation and marine productivity model. Top panel: ratio between global biosphere productivity at time “t” and global biosphere productivity for pre-industrial period (expressed in O_2 flux) obtained by the coupled IPSL model equipped with the PISCES¹⁹ and ORCHIDEE²⁰ models (crosses, Bopp and Kageyama, personal communication) and by interpretation of $\Delta^{17}\text{O}$ of O_2 data using the model of

Landais et al. (2007)³ and uncertainty of Supplementary Table 2 (grey envelope). Bottom panel: evolutions of CO₂ (black)¹ and Δ¹⁷O of O₂ (green)² as in the main manuscript. “

Reviewer #3 (Remarks to the Author):

I've gone through the reviewers' comments and authors' responses in detail, and attempted to understand the various estimates of GPP calculated in this paper. I regret that I have not been able to understand all the critical points but I hope my limitation will not cause me to make a wrong recommendation.

The new data in this paper are interesting. My basic concern has been the following. Any change in global productivity will almost certainly be linked to a change in the water cycle (with its large fractionations) and the ratio of terrestrial to marine production. Therefore, the question arises as to whether the decoupling between Δ¹⁷O and CO₂, observed at Termination 5, can be the consequence of a change in isotope fractionation associated with the hydrologic cycle or the ratio of terrestrial/marine production, rather than a change in global productivity.

>> We fully understand this concern, and this is indeed the reason why we addressed this issue in the uncertainty analyses described in the SOM. Still, thanks to these comments, we understand that our explanations were not clear enough, especially Supplementary Table 2, and we now propose a new table as well as a discussion on how to explain the Δ¹⁷O of O₂ anomaly by change in the ratio of terrestrial/marine production or change in the isotopic fractionation associated with the hydrological cycle as explained below.

In supplementary fig. 2, the authors vary properties associated with respiratory fractionation, photosynthetic fractionation, and other processes. None of these variations change the conclusion that global productivity was highest at Termination 5 or Stage 11. I believe that this plot shows that changing a property uniformly throughout the record leaves the productivity maximum at Stage 11 intact.

>> This is not totally true. Actually, we indeed make the assumption that the fractionation coefficients associated with photorespiration, dark respiration and photosynthesis do not change with time. We believe that this is a reasonable assumption since these are physically based processes and physical fractionation coefficients do not vary with time. However, we take into account the change of hydrological cycle, proportion of photorespiration and relative proportion of oceanic vs terrestrial productivity with time. In particular, we now provide in Supplementary Table 2, estimations of the influence of extreme changes in the relative proportion of terrestrial and oceanic productivities for LGM and MIS 11 and we added a specific test for the influence of the change in hydrological cycle (with a varying ¹⁷O-excess of water with time compared to a constant ¹⁷O-excess as assumed in our original model).

In the new version of the revised manuscript, we clearly explain our assumptions and the parameters that are allowed to vary and lead to the envelope displayed in Supplementary Figure 2 (l. 857 to 873). In order to help the reader, Supplementary Table 2 has been also fully redrawn to show the estimates of productivity associated with each assumption. We also explicitly give the estimates for productivity associated with each assumption for MIS 11 (as it was missing in the previous manuscript).

“Supplementary Figure 2. Biosphere productivity reconstructions over the last 450 ka. All the lines represent the ratio of global biospheric productivity between the considered time and pre-industrial. The dark grey area represents the ratio between biosphere productivity at the considered time and pre-industrial biosphere productivity as calculated from the Landais et al. (2007) model with the uncertainty bars deduced from uncertainties in the value of $\Delta^{17}\text{O}$ of O_2 produced by the earth biosphere calculated from uncertainties in the values of the fractionation coefficients, uncertainties in the ratio of oceanic to terrestrial biosphere as well as uncertainties on isotopic composition of meteoric water linked to temporal changes in the hydrological cycle (see Method and Supplementary Table 2). The solid and dotted black lines display the ratio between biosphere productivity at the considered time and pre-industrial biosphere productivity as calculated with the model of Blunier et al., 2012² with two different assumptions for the isotopic composition of meteoric water: solid line is associated with a constant H_2O anomaly with time while the dotted black line was obtained with a 20 ppm lower anomaly during the glacial periods. The light grey area represents the maximum uncertainty range of the biosphere productivity reconstruction, taking into account all uncertainties listed on Supplementary Table 2 as well as uncertainty in the model used for reconstruction and the biosphere productivity reconstruction of Blunier et al., 2012².”

I am not sure if I understand supplementary table 2. If I understand correctly, this table indicates that you calculate very different values of gross production depending on your choice of the assumed ratio of terrestrial to marine O_2 production, and fractionation factors. If so, then how do you know that the decoupling at Stage 11 is due to exceptional productivity, or changes in terrestrial/marine production and various isotope effects?

>> Indeed, the $\Delta^{17}\text{O}_{\text{bio}}$ is influenced by the ratio of terrestrial to marine O_2 production as shown by the new Supplementary Table 2 for both the LGM and MIS 11. This has a direct influence on the uncertainties associated with the reconstruction of the O_2 productivity flux as shown now explicitly on Supplementary Table 2 and included in Supplementary Figure 2. The same exercise has been done for the influence of the hydrological cycle and associated change in $\Delta^{17}\text{O}$ and $\delta^{18}\text{O}$ of water cycle as detailed below as answer to the other comments adding a new Supplementary Table 2 and the one-page discussion given at the end of the answer to reviewer 3. From such exercise, we are able to better discuss all the possible scenarios behind the $\Delta^{17}\text{O}$ of O_2 anomaly obtained during MIS 11 within the new version of the manuscript and demonstrate the undeniable contribution of global productivity in such a pattern (please, see our answers below).

I need to say that I do not understand supplementary table 2. I could not figure out where the $\Delta^{17}\text{O}_{\text{bio}}$ estimates come from in line 232 of the text, which refers to supplementary table 2. I cannot find matching numbers in the table, nor do averages match between the numbers in the text and the numbers in the supplemental table. In line 242, I am surprised by the statement that ocean productivity can be reduced by as much as 90%. This number is hard to understand, since total productivity is only suppressed by 40% or less during glacial times.

>> The $\Delta^{17}\text{O}_{\text{bio}}$ estimates of the pre-industrial period in the Supplementary Table 2 were wrong and corresponded to an older version of the manuscript, we apologize for this mistake. We recalculated the $\Delta^{17}\text{O}_{\text{bio}}$ estimates for the Last Glacial Maximum as well as for pre-industrial period and MIS 11 based on the new version of Supplementary Table 2 (l.231 to 237).

“The final estimates vary between 180 ± 50 ppm and 143 ± 50 ppm for the $\Delta^{17}\text{O}_{\text{bio}}$ during the Last Glacial Maximum and both the pre-industrial period and MIS 11 respectively. The maximum ranges of variations are not always independent from one period to another and results from uncertainties in the fractionation coefficients for the different biospheric processes⁴⁰, on the ratio between terrestrial and oceanic biosphere productivity⁴⁰ as well as on variations of the water cycle (Supplementary Table 2).”

As for l.242 in the revised manuscript, the meaning of the sentence has been modified to correspond to the study, that is a reduction of 10 to 40% of the global productivity during glacial periods or a global productivity of 60-90% compared to interglacials. Please, see l. 243-244 of the new manuscript: “Combining our different sensitivity studies, we find that the global productivity is reduced by 10 to 40% during glacial periods compared to interglacials.”

>> Finally, we understand that Supplementary Table 2 was complicated to use and we have it fully new in the new revised version (l. 918 to 931). In particular, it was not clear that some uncertainties are not independent and the MIS 11 results were missing. We propose this new table in the revised version that includes also explicitly the uncertainties associated with change in hydrological cycle and change in the ratio of terrestrial vs oceanic productivity at MIS 11. We directly give the associated productivity inferred from the different assumptions and we hope that it permits to better follow the interpretation taking into account the different sources of uncertainties.

Sensitivity test	$F_{\text{oce,PST}}/F_{\text{terr,PST}}$	$F_{\text{oce,LGM}}/F_{\text{terr,LGM}}$	$F_{\text{oce,MIS 11}}/F_{\text{terr,MIS 11}}$	$\Delta^{17}\text{O}_{\text{bio,PST}}$	$\Delta^{17}\text{O}_{\text{bio,LGM}}$	$\Delta^{17}\text{O}_{\text{bio,MIS 11}}$	$F_{\text{bio,LGM}}/F_{\text{bio,PST}}$	$F_{\text{bio,MIS 11}}/F_{\text{bio,PST}}$
Average situation	0.52	0.82	0.52	158	195	158	0.69	1.17
High $F_{\text{oce,PST}}/F_{\text{terr,PST}}$	0.59	0.82	0.52	162	195	162	0.71	1.17
Low $F_{\text{oce,PST}}/F_{\text{terr,PST}}$	0.45	0.82	0.52	153	195	153	0.67	1.18
High $F_{\text{oce,LGM}}/F_{\text{terr,LGM}}$	0.52	1.08	0.52	158	203	158	0.66	1.17
Low $F_{\text{oce,LGM}}/F_{\text{terr,LGM}}$	0.52	0.56	0.52	158	183	158	0.75	1.17
High $F_{\text{oce,MIS 11}}/F_{\text{terr,MIS 11}}$	0.52	0.82	0.59	158	195	162	0.69	1.14
Low $F_{\text{oce,MIS 11}}/F_{\text{terr,MIS 11}}$	0.52	0.82	0.45	158	195	153	0.69	1.22
Lowest estimate of the slopes associated with fractionation factor in the water cycle and biosphere (cumulative errors)	0.52	0.82	0.52	129	166	129	0.69	1.22
Highest estimate of the slopes associated with fractionation factor in the water cycle and biosphere (cumulative errors)	0.52	0.82	0.52	187	224	187	0.69	1.14
Maximum uncertainty on photosynthesis fractionation (Eisenstadt et al., 2010)	0.52	0.82	0.52	193	230	193	0.69	1.14
Maximum uncertainty on the slope of respiration (Stolper et al., 2018)	0.52	0.82	0.52	93	130	93	0.7	1.33
Change of hydrological cycle ($\Delta^{17}\text{O}$ of O_2 from terrestrial biosphere lower by 20 ppm during LGM and 10 ppm higher during MIS 11)	0.52	0.82	0.52	158	183	164	0.75	1.12

Supplementary Table 2. Estimates of $\Delta^{17}\text{O}_{\text{bio}}$ and associated reconstruction of the productivity for pre-industrial, Last Glacial Maximum (LGM) and MIS 11, using equations S1 and S2 with various sensitivity tests. The first 9 lines (in grey) were directly taken from sensitivity tests of Landais et al. (2007) taking into account uncertainties in the ratio between oceanic and terrestrial biosphere productivity or in the estimates of the fractionation factors used for calculations of $\Delta^{17}\text{O}_{\text{terr}}$ and $\Delta^{17}\text{O}_{\text{oce}}$. Note that these fractionation factors are based on physical processes which do not vary with time so that the associated uncertainties are not independent for pre-industrial, LGM and MIS 11. The last 3 lines correspond to 3 new sensitivity tests for the influence of fractionation during photosynthesis, possible low slope of $\delta^{17}\text{O}$ vs $\delta^{18}\text{O}$ during respiration, and possible changes in the water cycle leading to modification of the triple isotopic composition of oxygen in water (^{17}O -excess, see text) directly transmitted to the $\Delta^{17}\text{O}$ of O_2 produced by terrestrial productivity. For all sensitivity tests, we considered extreme values so that the reconstructed global productivities are also showing extreme values.

Before I could endorse this paper for publication, I would need to see a discussion of the possibility that the Stage 11 anomaly is due to differences, at Stage 11, of properties listed above, with respect to the other interglacials.

>> This is a very good suggestion that we propose to address in the new version by concentrating on possible variations of the ratio of terrestrial to oceanic productivity and change in hydrological cycle.

Actually, we can calculate that a decrease of the ratio between terrestrial and oceanic productivity by a factor of 2 over MIS 11 with respect to the present-day situation is needed to explain the anomalous $\Delta^{17}\text{O}_{\text{atm}}$ over MIS 11. This is quite extreme given the uncertainty on the ratio between terrestrial and oceanic productivities during interglacial period (Supplementary Table 2) and do not go along with what we actually highlight in the paper, based on $\Delta^{17}\text{O}$ values as well as regional marine vs terrestrial productivity records (Diekmann and Kuhn, 2002; Martínez-García et al., 2009; Melles et al., 2012; deVernal and Hillaire-Marcel, 2008), that document a probable increase of the ratio of terrestrial vs oceanic productivity during MIS 11.

As for the change in water cycle, there are two ways to obtain the observed anomaly: either a global increase of water ^{17}O -excess by more than 20 ppm during MIS 11 compared to the present interglacial or an increase of both $\delta^{18}\text{O}$ and $\delta^{17}\text{O}$ of global meteoric water and global ocean so that the $\Delta^{17}\text{O}$ of O_2 produced by biosphere is increased by 20 ppm. For the first solution, ^{17}O -excess of the ocean is expected to be closely related to ice volume so that ^{17}O -excess of the ocean water during MIS 11 should be very similar to the present-day value. In this case, a ^{17}O -excess increase of 30 ppm for continental meteoric water is needed during MIS 11 with respect to present-day value. This signal should be associated with a strong decrease of relative humidity at evaporation (30 % which is not realistic). The second solution implies an increase of global mean $\delta^{18}\text{O}$ of meteoric water by more than 3 permil with respect to present-day and would necessarily have an effect on the global $\delta^{18}\text{O}_{\text{atm}}$. The $\delta^{18}\text{O}_{\text{atm}}$ during MIS 11 is between 0 and 0.5 permil higher than during MIS 1 so that it does not support such a hypothesis.

We propose this fully new discussion in the supplementary material, to address this comment in detail, considering all possible scenarios (l. 536 to 555 and 576 to 616):

“The reconstructed biosphere productivity using the Landais et al. (2007)⁴⁰ formulation and uncertainties is detailed in Supplementary Table 2 for different time periods of interest within this study, i.e. the pre-industrial, the LGM and the MIS 11. On average, we obtain a biosphere productivity at the beginning of MIS 11 which is 17 % higher than during MIS 1 and a biosphere productivity during the LGM which is 31% lower than during MIS 1, a result in agreement with output of the IPSL coupled model equipped with vegetation and ocean productivity modules (Supplementary Figure 4).

As in Landais et al. (2007)⁴⁰, it is shown in Supplementary Table 2 that the uncertainty in fractionation coefficients as well as on the ratio of oceanic to terrestrial productivity is leading to uncertainties in the reconstructed past global productivity. The fractionation coefficients are based on physical properties and are hence not expected to vary with time so that the possible bias on these coefficients should apply to the different periods. On opposite, the ratio of oceanic to terrestrial productivity is expected to vary with time and is a large source of uncertainty. We thus present in Supplementary Table 2 calculations performed with the largest range of possible ratios of oceanic to terrestrial productivity estimated in Landais et al. (2007)⁴⁰.

Since the publication of Landais et al. (2007)⁴⁰, new estimates of fractionation coefficients within the oxygen cycle are available^{22,43,76} and influence of the water cycle organisation has also been suggested²⁴. As a consequence, we performed 3 **additional** types of sensitivity tests to address the influences of such new determination and associated uncertainty ranges on our results. “

- **“Uncertainty in the fractionation within water cycle.** The general assumption is that the relationship between $\delta^{17}\text{O}$ and $\delta^{18}\text{O}$ (meteoric water line) remains the same over glacial – interglacial cycles. However, measurements in the Vostok ice core have shown that the ^{17}O -excess, defined as $^{17}\text{O}\text{-excess} = \ln(1+\delta^{17}\text{O}) - 0.528 \cdot \ln(1+\delta^{18}\text{O})$, is lower by up to 20 ppm during the last glacial maximum⁷⁸. While it has been shown that this is a local effect⁷⁹, we still performed a sensitivity test with decreasing ^{17}O -excess of all continental meteoric waters by 20 ppm during glacial periods and increasing ^{17}O -excess by 10 ppm during MIS 11 with respect to our current interglacial period. Taking into account such variations, the reconstructed biospheric productivity does not differ by more than 6% from the case without change in meteoric water ^{17}O -excess (Supplementary Table 2).

All the sensitivity tests displayed above suggest that the biosphere productivity during MIS 11 is significantly higher than for our current interglacial, leading to the envelop displayed on Supplementary Figure 2. **Actually, explaining the $\Delta^{17}\text{O}$ of O_2 anomaly without invoking a significant**

increase of the global productivity at the beginning of MIS 11 requires huge change of the $\Delta^{17}\text{O}$ of O_2 produced by terrestrial biosphere (through changes in fractionation factor or changes in water cycle, hypothesis 1) or of the ratio of oceanic to terrestrial biosphere productivity (hypothesis 2). These two possibilities are rather unrealistic as detailed below.

In hypothesis 1, an increase of 30 ppm for $\Delta^{17}\text{O}$ of O_2 produced by terrestrial biosphere is needed. This can be obtained either by a huge change in the fractionation factors between our current interglacial and MIS 11 (which is not realistic since these are based on physical processes which do not vary with time) or by a change of water cycle during MIS 11 with respect to present-day value. To reach this 30 ppm increase through modification of the water cycle, one option is to increase the ^{17}O -excess of meteoric water by 30 ppm during MIS 11 compared to our current interglacial. We do not have ^{17}O -excess values for MIS 11 yet, but ice core ^{17}O -excess values obtained for the last interglacial are very similar to values obtained for the present interglacial (Landais et al., 2008)⁷⁸. Moreover, increasing ^{17}O -excess by 30 ppm would require unrealistic decrease of relative humidity at evaporation (by 30%, Barkan and Luz, 2007⁸⁰) during MIS 11 compared to our present interglacial. Finally, an increase of the mean $\delta^{18}\text{O}$ and $\delta^{17}\text{O}$ of meteoric water used by the plant along the meteoric water line (i.e. without global change in the global ^{17}O -excess) would as well be a solution (see Figure 4 of Landais et al., 2007⁴⁰) but to reach the expected increase of 30 ppm in $\Delta^{17}\text{O}$ of O_2 produced by the terrestrial biosphere, it would require an increase of global mean $\delta^{18}\text{O}$ of meteoric water by $\sim 3\text{‰}$ with respect to present-day that will be transmitted to the global $\delta^{18}\text{O}_{\text{atm}}$. The $\delta^{18}\text{O}_{\text{atm}}$ during MIS 11 is between 0 and 0.5 ‰ higher than during MIS 1 so that it does not support this hypothesis.

Hypothesis 2 requires an increase of the oceanic vs terrestrial productivity by a factor of 2 during MIS 11 compared to present day value. This does not go along available observations suggesting an increase in the ratio of terrestrial to oceanic productivity during MIS 11 (see main text) and is still well above the uncertainty for the variation of the ratio of oceanic to terrestrial productivity during our interglacial period (14%, Supplementary Table 2).”

**Title:** Exceptionally high biosphere productivity at the beginning of Marine Isotopic Stage 11

**Order of Authors:** Margaux Brandon^{*a,b}, Amaelle Landais^a, Stéphanie Duchamp-Alphonse^b,
Violaine Favre^a, Léa Schmitz^a, Héloïse Abrial^a, Frédéric Prie^a, Thomas Extier^a, Thomas
Blunier^c

**Author Affiliation :**

6 ^aLaboratoire des Sciences du Climat et de l'Environnement, LSCE/IPSL, CEA-CNRS-UVSQ,
Université Paris-Saclay, 91191, Gif-sur-Yvette, France

8 ^bGEOPS, Université Paris-Saclay, CNRS, 91405 Orsay, France

9 ^cCopenhagen Univ., Niels Bohr Institute, Centre for Ice and Climate, Juliane Maries Vej 30,
DK-2100 Copenhagen, Denmark

***Corresponding author:** Margaux Brandon

Address : GEOPS, Université Paris-Saclay, CNRS, 91405 Orsay, France

Phone number: 33 (1) 69 15 61 26

E-mail address: margaux.brandon@universite-paris-saclay.fr

**Abstract**

Biosphere productivity is associated with important CO₂ fluxes through photosynthesis and
respiration. Quantifying the variations of global biosphere productivity over deglaciations is
thus key for a better comprehension of the variations of atmospheric CO₂ over glacial-
interglacial cycles. Using the first high resolution record of $\Delta^{17}\text{O}$ of O₂ over Termination V and
Marine Isotopic Stage (MIS) 11, we reconstruct the past global biosphere productivity over this
key period for glacial-interglacial cycles of the last 800,000 years, corresponding to the first
termination with a large CO₂ amplitude followed by the longest interglacial. We show that the
global oxygen biosphere productivity at the end of Termination V is 10 to 30 % higher
compared to the younger terminations. We suggest that higher biosphere productivity was due
to extended period of strong terrestrial productivity that probably contributed to reduce the
atmospheric CO₂ level at the beginning of MIS 11.

**INTRODUCTION**

The largest pre-anthropogenic changes in atmospheric CO₂ concentration of the last
800,000 years are observed during deglaciations with increases of up to 100 ppm in a few
thousand years¹. Oceanic carbon reservoir is broadly suspected to play a central role in these
atmospheric CO₂ increases. Leading hypotheses invoke CO₂ degassing from the ocean induced
by more vigorous convection in the Southern Ocean (physical pump)², concomitant decrease
of the net organic matter export to the deep ocean (soft-tissue pump)³, enhanced exchanges
between ocean surface and atmosphere due to sea-ice melting⁴, and increased sea-surface
temperature (solubility pump)⁵. Modelling studies simulating changes in oceanic processes are
quite controversial^{6,7} but in all cases, they are not able to explain the full increase of atmospheric
CO₂ rises during deglaciations. In parallel, terrestrial primary productivity and carbon stock

increase during deglaciations, thus acting as a significant CO₂ land sink^{8,9} so that an important
additional source of CO₂ is required to explain the entire deglacial CO₂ pattern. The missing
component has been suggested to be the important release of inert carbon from a thawing
permafrost^{8,10}.

Quantifying changes in the carbon cycle over deglaciations relies on data compilation
and modelling studies to estimate both carbon stocks and carbon fluxes⁸. Some information on
carbon stocks can be obtained from $\delta^{13}\text{C}$ of carbonates using a wealth of data obtained from
marine sediments¹¹ and $\delta^{13}\text{C}$ of atmospheric CO₂^{12,13}. In parallel, information on the evolution
of past global productivity, a major component of carbon flux, is very sparse and often limited
to the last deglaciation^{8,9}.

In the ocean, little is known about deglacial oceanic productivity over the last 800 ka.
Increased contents of organic matter (Total Organic Carbon (TOC) and biomarkers)^{3,14} in
sediments, have been used to reflect increased downward flux of carbon associated with the
photosynthetic biomass. However, while such records provide crucial information about the
efficiency of the soft-tissue pump, they remain sparse and cannot reliably be used to infer
oceanic primary productivity. Besides, they do not consider the relative contribution of the
downward flux of Particulate Inorganic Carbon (mainly CaCO₃) produced by calcifying
plankton in the sunlit ocean (mainly coccolithophores and planktonic foraminifera), that creates
a surface-to-deep alkalinity gradient, causing CO₂ to be released back to the atmosphere¹⁵ and
that represents the carbonate counter pump. Past changes in biological carbon pump are best
represented by changes in the buried TOC/CaCO₃ ratio, suggested to reflect C-rain ratio¹⁶.
However, despite its accuracy to provide biological export production, only one record of
sedimentary TOC/CaCO₃ exists in the Southern Ocean, for the last 800 ka, so far¹⁴. This record
is however very precious since Southern Ocean biosphere productivity has a strong potential

for increases in the past while on lower latitudes, the biosphere productivity is already
maximum today¹⁷.

On continents, variations in vegetation cover and type can be related, although
indirectly, to variations of the biosphere terrestrial productivity through the use of dynamic
vegetation models modelling both biome and associated productivity⁹. Pollen counting and
sedimentary TOC are thus useful for biosphere productivity reconstruction^{18,19} but they are
unfortunately indirect and rely on the use of biosphere models. Moreover, similarly to oceanic
records, these observations provide regional records that are not easy to use for documenting
the past global carbon cycle. Ciais et al. (2012)⁸ proposed to use the isotopic composition of
oxygen of atmosphere ($\delta^{18}\text{O}_{\text{atm}}$) as a tracer for terrestrial biosphere productivity. However, this
proxy is a complex tracer being influenced by hydrological cycle at first order^{20,21} and its use
as a quantitative tool for productivity reconstruction depends on the exact determination of
associated fractionation factors in the water and biosphere cycle²².

A total estimate of the global biospheric fluxes and their temporal variations can be
obtained more directly from measurements of $\Delta^{17}\text{O}$ of O_2 ($\ln(\delta^{17}\text{O}+1)-0.516*\ln(\delta^{18}\text{O}+1)$) in ice
cores^{23,24}. This method provides O_2 fluxes and the conversion from O_2 to CO_2 fluxes can be
done from the stoichiometry of the biological processes of photosynthesis and respiration²⁵.
$\Delta^{17}\text{O}$ of O_2 measures the variation of the triple isotopic composition of atmospheric O_2 with
respect to modern oxygen so that by definition, $\Delta^{17}\text{O}$ of O_2 is nil today. Previous experimental
studies showed that $\Delta^{17}\text{O}$ of O_2 increases in a closed biospheric system when the exchanges
with the stratosphere are prevented: the biological productivity leads to $\Delta^{17}\text{O}$ of O_2 increase
while photochemical reactions occurring in the stratosphere have the effect of decreasing $\Delta^{17}\text{O}$
of O_2 ²³. Such properties open a new way to infer the relative proportion of oxygen fluxes issued
from biosphere processes and from the exchanges between the stratosphere and the
troposphere²³. For paleoproductivity reconstructions, measurements of the evolution of $\Delta^{17}\text{O}$

of O₂ in ancient air trapped in the Vostok and GISP2 ice cores already provided information on
the evolution of global biosphere productivity, over the last 400 ka²⁴. The results show a
systematic larger productivity, during interglacial than during glacial periods with interglacial
levels remaining close to the current biosphere productivity.

Over the last 9 deglaciations where CO₂ concentration was measured, Termination V,
occurring between 433 and 426 ka on ice core records on the latest AICC2012 chronology²⁶, is
probably the most intriguing. This Termination is framed by the particularly long and strong
glacial Marine Isotopic Stage 12 (MIS 12), followed by the long and warm interglacial MIS 11
(426 to 398 ka on AICC2012). This is the first Termination after the Mid-Brunhes event
marking a fundamental change in the climate system from mild to warm periods with associated
lower to higher CO₂ concentrations, and from larger to smaller ice volumes^{27–30} associated with
higher sea-level than the present³⁰. Termination V is also occurring in a particular orbital
context, i.e. low eccentricity around 400 ka, which is known to have an influence on the carbon
cycle as observed in $\delta^{13}\text{C}$ oceanic records³¹. Surprisingly, the CO₂ level is not exceptionally
high compared to the following warm periods²⁸, and one key to this, is probably to be found in
the biosphere dynamic. On the continents, pollen data^{19,29} suggest a strong and long increase in
terrestrial productivity with probable impacts on terrestrial vs atmospheric C stocks, during
MIS 11. In the Ocean, MIS 11 displays an unusual increase in carbonate storage in low-latitude
neritic³² and high-latitude pelagic environments³³. While it is clearly associated with a major
phase in coral reef expansion³⁴ and a climax in calcareous phytoplankton productivity
respectively³⁵, the impact of this large carbonate production on atmospheric pCO₂ is not
understood. Therefore, the biosphere productivity fluxes during Termination V and MIS 11
need to be investigated.

Here we present the first measurements of the triple isotopic composition of atmospheric
oxygen ($\Delta^{17}\text{O}$ of O₂) over Termination V. Using these measurements and new correction factors

compared to the previous record of Blunier et al., (2012)²⁴, we reconstruct the oxygen fluxes
associated with biosphere productivity over this particularly strong Termination and compare
it with biosphere productivity reconstruction of the 4 youngest Terminations. Ice core $\delta^{18}\text{O}_{\text{atm}}$
(or $\delta^{18}\text{O}$ of O_2) record, and terrestrial and oceanic records related to biosphere productivity are
used to discuss the relative contribution of changes in oceanic vs terrestrial biosphere fluxes to
the atmospheric CO_2 rise during Termination V.

The biosphere productivity over Termination V and the beginning of MIS 11 is found
to be 10-30% higher than productivity over the pre-industrial period, an exceptional value never
encountered over the last 4 interglacial periods. This productivity peak is most probably due to
an increase of the terrestrial productivity during this period favored by a particular context of
low eccentricity leading to long growing summer season, warm temperature at high latitudes of
the northern hemisphere enabling biosphere productivity from terrestrial soils usually frozen
and possibly a relatively slow sea level rise during Termination V enabling productivity from

[revised manuscript text omitted]
 water cycle²⁴, photosynthesis⁴² and respiration⁴³. The estimate for triple isotopic
composition of oxygen produced by terrestrial and oceanic biospheres, $\Delta^{17}\text{O}_{\text{bio}}$, was revised

compared from those of Landais et al. (2007)⁴⁰ taking into account recent studies. The final
estimates vary between 180±50 ppm and 143±50 ppm for the $\Delta^{17}\text{O}_{\text{bio}}$ during the Last Glacial
Maximum and both the pre-industrial period and MIS 11 respectively. The maximum ranges of
variations are not always independent from one period to another and results from uncertainties
in the fractionation coefficients for the different biospheric processes⁴⁰, on the ratio between
terrestrial and oceanic biosphere productivity⁴⁰ as well as on variations of the water cycle
(Supplementary Table 2). Combining these estimates with the covariation of CO₂ concentration
and flux of negative $\Delta^{17}\text{O}$ of O₂ anomaly from the stratosphere, we converted the full $\Delta^{17}\text{O}$ of
O₂ record over the last 445 ka into an estimate for the evolution of the global oxygen
productivity (Figure 3). This reconstruction was also completed with an alternative
reconstruction of global oxygen productivity using an alternative model, the one of Blunier et
al. (2012)²⁴, forced by our $\Delta^{17}\text{O}$ of O₂ data (Supplementary Figure 2).

Combining our different sensitivity studies, we find that the global productivity is reduced by
10 to 40% during glacial periods compared to interglacials. At the beginning of most
interglacial periods, the global productivity remains close to the preindustrial level. The only
exception to this general behavior is the strong oxygen biosphere productivity at the end of
Termination V reaching values 10 to 30% higher than during the pre-industrial period.

**DISCUSSION**

Complementary information on the origin and specificity of the $\Delta^{17}\text{O}$ of O₂ signal over
Termination V can be obtained from the ice core $\delta^{18}\text{O}_{\text{atm}}$ (or $\delta^{18}\text{O}$ of O₂) record over the last
800 ka²¹ (Figure 3). $\delta^{18}\text{O}_{\text{atm}}$ is a complex parameter resulting from both biosphere productivity
and low latitude water cycle⁴⁴. In particular, it has been shown that Weak Monsoon Intervals
observed during Heinrich events lead to increases in $\delta^{18}\text{O}_{\text{atm}}$ via changes in the low latitude

water cycle⁴⁵. Another way to increase the $\delta^{18}\text{O}_{\text{atm}}$ is to increase the ratio between terrestrial
 and oceanic biosphere productivity^{8,44}.

[revised manuscript text omitted]
 δ¹⁸O_{atm}, δ¹⁷O_{atm}, δO₂/Ar measurements and Δ¹⁷O of O₂ calculation, respectively.

**Corrections on Δ¹⁷O of O₂**

**Atmospheric air calibration.** Every day, δ¹⁸O, δ¹⁷O and δO₂/Ar of atmospheric air are
measured and are then used to calibrate our measurements following:

$$\delta^{18}O_{ext\ air\ corr} = \left[\frac{(\delta^{18}O_{sample}/1000) + 1}{(\delta^{18}O_{ext\ air}/1000) + 1} - 1 \right] * 1000$$

$$\delta^{17}O_{ext\ air\ corr} = \left[\frac{(\delta^{17}O_{sample}/1000) + 1}{(\delta^{17}O_{ext\ air}/1000) + 1} - 1 \right] * 1000$$

The $\delta^{18}O_{ext\ air}$ and $\delta^{17}O_{ext\ air}$ were constant during the two measurement periods so that we used
 the average values over the two corresponding periods to correct the raw data. The daily
 correction was the same every day during each period.

**Correction due to fractionation in the firn column**

- **Gravitational fractionation**

Gravitational fractionation operates in firn due to Earth gravity field. $\delta^{18}O$ and $\delta^{17}O$ were
 obtained by corrections for this diffusive process using $\delta^{15}N$ in neighbouring samples⁶⁷. The
 correction applied depends on the difference of mass between the two isotopes considered so
 that:

$$\delta^{18}O_{gravitational\ corr} = \delta^{18}O_{measured} - 2 * \delta^{15}N$$

$$\delta^{17}O_{gravitational\ corr} = \delta^{17}O_{measured} - 1 * \delta^{15}N$$

- **Thermal fractionation**

In the firn column, diffusive processes operate due to changes in temperature or because of the
 Earth gravity. These processes lead to isotopic fractionation of O_2 that was taken into account
 for $\delta^{18}O_{atm}$ reconstruction in the NGRIP ice core over abrupt temperature changes of the last
 glacial period⁶⁸.

To check the effect of thermal fractionation on $\Delta^{17}\text{O}$ of O_2 , we performed measurements of
$\Delta^{17}\text{O}$ of O_2 over the top 20 m of the EastGRIP firn where a strong seasonal gradient is present.
However, the resulting $\Delta^{17}\text{O}$ of O_2 was not showing any significant deviation from the
atmospheric value hence suggesting that thermal fractionation does not modify the $\Delta^{17}\text{O}$ of O_2
of the atmosphere in the firn column. Moreover, in Antarctica, surface temperature variations
are much lower than in Greenland during deglaciations or climatic variability of the last glacial
period so that thermal fractionation is not expected to have a significant effect on the isotopic
composition of trapped oxygen.

- **Pore close-off effect**

Pore close-off at the bottom of the firn has been shown to affect $\delta\text{O}_2/\text{N}_2$, $\delta\text{Ar}/\text{N}_2$ with potential
effects on $\delta^{15}\text{N}$ and $\delta^{40}\text{Ar}$ in certain cases⁶⁹. We checked this possible effect on $\Delta^{17}\text{O}$ of O_2 by
comparing $\Delta^{17}\text{O}$ of O_2 in bubbly ice at the top of the NEEM ice core. After correction of
gravitational effect, we found a systematic enrichment of 13 per meg which could potentially
bias the reconstruction of atmospheric $\Delta^{17}\text{O}$ of O_2 from $\Delta^{17}\text{O}$ of O_2 in trapped air.

**Gas loss correction.** During storage, O_2 in ice samples is subject to gas loss fractionation due
to diffusion processes⁷⁰ and O_2/N_2 ratio is always lower by several % in ice samples stored
several years at -20°C than ice samples stored at -50°C ⁷¹. Such gas loss effect is also associated
with isotopic fractionation of oxygen, $\delta^{18}\text{O}_{\text{atm}}$ trapped in the ice being higher when $\delta\text{O}_2/\text{N}_2$
decreases with a slope for the relationship of -0.01 ($\delta^{18}\text{O}_{\text{atm}}$ vs $\delta\text{O}_2/\text{N}_2$)^{20,21,72,73}. We thus expect
that $\delta^{17}\text{O}$ can also be affected by this gas loss process and that it may create an anomaly of $\Delta^{17}\text{O}$
of O_2 . To check this effect, measurements of $\delta^{17}\text{O}$, $\delta^{18}\text{O}_{\text{atm}}$, $\Delta^{17}\text{O}$ of O_2 and $\delta\text{O}_2/\text{Ar}$ have been
performed on 3 samples of GRIP ice core (clathrate ice stored during more than 20 years at -
20°C). Each of the 3 ice samples have been cut in order to analyse the interior and the exterior
of the sample separately. $\delta\text{O}_2/\text{N}_2$ measurements could not be performed on exactly the same

[revised manuscript text omitted]

From the calculation of $\Delta^{17}\text{O}_{\text{bio, pre-industrial}}$ and $\Delta^{17}\text{O}_{\text{bio, LGM}}$ (Supplementary Table 2), $\Delta^{17}\text{O}_{\text{bio, t}}$ is
calculated through a scaling on the variations of CO_2 concentration between pre-industrial
period (280 ppmv) and the LGM (190 ppmv)⁷⁵ such as:

$$534 \quad \Delta^{17}\text{O}_{\text{bio, t}} = \Delta^{17}\text{O}_{\text{bio, pre-industrial}} + (\Delta^{17}\text{O}_{\text{bio, LGM}} - \Delta^{17}\text{O}_{\text{bio, pre-industrial}}) * \left(\frac{280 - (\text{CO}_2)_t}{90} \right) \text{ (eq S2)}$$

The reconstructed biosphere productivity using the Landais et al. (2007)⁴⁰ formulation and
uncertainties is detailed in Supplementary Table 2 for different time periods of interest within
this study, i.e. the pre-industrial, the LGM and the MIS 11. On average, we obtain a biosphere
productivity at the beginning of MIS 11 which is 17 % higher than during MIS 1 and a biosphere
productivity during the LGM which is 31% lower than during MIS 1, a result in agreement with
output of the IPSL coupled model equipped with vegetation and ocean productivity modules
(Supplementary Figure 4).

As in Landais et al. (2007)⁴⁰, it is shown in Supplementary Table 2 that the uncertainty in
fractionation coefficients as well as on the ratio of oceanic to terrestrial productivity is leading
to uncertainties in the reconstructed past global productivity. The fractionation coefficients are

based on physical properties and are hence not expected to vary with time so that the possible
bias on these coefficients should apply to the different periods. On opposite, the ratio of oceanic
to terrestrial productivity is expected to vary with time and is a large source of uncertainty. We
thus present in Supplementary Table 2 calculations performed with the largest range of possible
ratios of oceanic to terrestrial productivity estimated in Landais et al. (2007)⁴⁰.

Since the publication of Landais et al. (2007)⁴⁰, new estimates of fractionation coefficients
within the oxygen cycle are available^{22,43,76} and influence of the water cycle organisation has
also been suggested²⁴. As a consequence, we performed 3 **additional** types of sensitivity tests
to address the influences of such new determination and associated uncertainty ranges on our
results.

- **Uncertainty in photosynthesis fractionation**. Some marine species show fractionation
during marine photosynthesis⁴² while it has long been assumed that photosynthesis does
not fractionate⁷⁷. We performed sensitivity tests to compare biosphere productivity
reconstructions without any fractionation and reconstruction with the different
fractionation effects during marine photosynthesis as observed in Eisenstadt et al.
(2010)⁴². The largest change on the reconstructed biosphere productivity is obtained
using the observed photosynthesis fractionation effect associated with *Emiliana huxleyi*
($\delta^{18}\text{O}=5.81\text{‰}$ and slope between $\ln(\delta^{17}\text{O}+1)$ and $\ln(\delta^{18}\text{
[revised manuscript text omitted]

- 10. Crichton, K. A., Bouttes, N., Roche, D. M., Chappellaz, J. & Krinner, G. Permafrost carbon as a
missing link to explain CO₂ changes during the last deglaciation. *Nature Geoscience* **9**, 683–686
(2016).
- 11. Peterson, C. D., Lisiecki, L. E. & Stern, J. V. Deglacial whole-ocean $\delta^{13}\text{C}$ change estimated from
480 benthic foraminiferal records. *Paleoceanography* **29**, 549–563 (2014).

- 12. Eggleston, S., Schmitt, J., Bereiter, B., Schneider, R. & Fischer, H. Evolution of the stable carbon
isotope composition of atmospheric CO₂ over the last glacial cycle. *Paleoceanography* **31**, 434–
452 (2016).
- 13. Schmitt, J. *et al.* Carbon Isotope Constraints on the Deglacial CO₂ Rise from Ice Cores. *Science*
**336**, 711–714 (2012).
- 14. Diekmann, B. & Kuhn, G. Sedimentary record of the mid-Pleistocene climate transition in the
southeastern South Atlantic (ODP Site 1090). *Palaeogeography, Palaeoclimatology,*
*Palaeoecology* **182**, 241–258 (2002).
- 15. Salter, I. *et al.* Carbonate counter pump stimulated by natural iron fertilization in the Polar
Frontal Zone. *Nature Geoscience* **7**, 885–889 (2014).
- 16. Duchamp-Alphonse, S. *et al.* Enhanced ocean-atmosphere carbon partitioning via the carbonate
counter pump during the last deglacial. *Nature Communications* **9**, (2018).
- 17. Hain, M. P., Sigmal, D. & Haug, G. H. 8.18–The biological Pump in the Past. in *Treatise on*
*Geochemistry* vol. 8 485–517 (2014).
- 18. Tzedakis, P. C., Hooghiemstra, H. & Pälike, H. The last 1.35 million years at Tenaghi Philippon:
revised chronostratigraphy and long-term vegetation trends. *Quaternary Science Reviews* **25**,
3416–3430 (2006).
- 19. Melles, M. *et al.* 2.8 Million Years of Arctic Climate Change from Lake El’gygytgyn, NE Russia.
*Science* **337**, 315–320 (2012).
- 20. Severinghaus, J. P., Beaudette, R., Headly, M. A., Taylor, K. & Brook, E. J. Oxygen-18 of O₂ Records
the Impact of Abrupt Climate Change on the Terrestrial Biosphere. *Science* **324**, 1431–1434
(2009).
- 21. Extier, T. *et al.* On the use of δ¹⁸O_{atm} for ice core dating. *Quaternary Science Reviews* **185**,
244–257 (2018).
- 22. Luz, B. & Barkan, E. The isotopic composition of atmospheric oxygen. *Global Biogeochemical*
*Cycles* **25**, GB3001 (2011).

- 23. Luz, B., Barkan, E., Bender, M. L., Thieme, M. H. & Boering, K. A. Triple-isotope composition of
atmospheric oxygen as a tracer of biosphere productivity. *Nature* **400**, 547–550 (1999).
- 24. Blunier, T., Bender, M. L., Barnett, B. & von Fischer, J. C. Planetary fertility during the past 400 ka
based on the triple isotope composition of O₂ in trapped gases from the Vostok ice core. *Climate*
*of the Past* **8**, 1509–1526 (2012).
- 25. Hoffmann, G. *et al.* A model of the Earth's Dole effect. *Global Biogeochemical Cycles* **18**, GB1008
(2004).
- 26. Bazin, L. *et al.* An optimized multi-proxy, multi-site Antarctic ice and gas orbital chronology
(AICC2012): 120–800 ka. *Clim. Past* **9**, 1715–1731 (2013).
- 27. Jouzel, J. *et al.* Orbital and Millennial Antarctic Climate Variability over the Past 800,000 Years.
*Science* **317**, 793–796 (2007).
- 28. Siegenthaler, U. Stable Carbon Cycle-Climate Relationship During the Late Pleistocene. *Science*
**310**, 1313–1317 (2005).
- 29. de Vernal, A. & Hillaire-Marcel, C. Natural Variability of Greenland Climate, Vegetation, and Ice
Volume During the Past Million Years. *Science* **320**, 1622–1625 (2008).
- 30. Raymo, M. E. & Mitrovica, J. X. Collapse of polar ice sheets during the stage 11 interglacial.
*Nature* **483**, 453–456 (2012).

[revised manuscript text omitted]

- 80. Barkan, E. & Luz, B. Diffusivity fractionations of $\text{H}_2^{16}\text{O}/\text{H}_2^{17}\text{O}$ and $\text{H}_2^{16}\text{O}/\text{H}_2^{18}\text{O}$ in air and their
implications for isotope hydrology. *Rapid Commun. Mass Spectrom.* **21**, 2999–3005 (2007).
- 81. Prokopenko, M. G., Pauluis, O. M., Granger, J. & Yeung, L. Y. Exact evaluation of gross
photosynthetic production from the oxygen triple-isotope composition of O_2 : Implications for
the net-to-gross primary production ratios. *Geophysical Research Letters* **38**, L14603 (2011).

**Acknowledgments**

This work benefited from funding from the ANR program HUM17 as well as the INSU-LEFE-
IMAGO program BIOCOD. M.B. receives an IDEX-IDI PhD grant from Paris-Saclay. It is a
contribution to the European Project for Ice Coring in Antarctica (EPICA). It is a pleasure to
thank also Jean Jouzel, Jochen Schmitt, Ji-Woong Yang and Michael Bender for useful
discussions as well as Laurent Bopp and Masa Kageyama for their contribution on model
reconstructions. We also thank Gregory Teste involved in cutting EDC samples.

**Author contributions**

839 A.L., S.D.A and M.B. designed this study. M.B, F.P., V.F, H.A and L.S performed the $\Delta^{17}\text{O}$ of
840 O_2 measurements. M.B and A.L. corrected the data and calculated the flux of oxygen biosphere
productivity with the help of T.B. M.B., A.L. and S.D.A. wrote the manuscript with the
contribution of T.B and T.E.

**Competing interests:** The authors declare no competing interests.

Exceptionally high biosphere productivity at the beginning of Marine Isotopic Stage 11

SUPPLEMENTARY INFORMATION

**Supplementary Figure 1. Relationship between CO₂¹ and Δ¹⁷O of O₂² and this study for Termination I to**

**V.** The slope of the Δ¹⁷O of O₂ vs CO₂ anti-correlation is much lower over Termination V than for

Terminations I-IV. The correlation coefficient is also lower for Termination V than for younger

Terminations.

**Supplementary Figure 2. Biosphere productivity reconstructions over the last 450 ka.** All the lines

represent the ratio of global biospheric productivity between the considered time and pre-industrial.

The dark grey area represents the ratio between biosphere productivity at the considered time and

pre-industrial biosphere productivity as calculated from the Landais et al. (2007) model with the

uncertainty bars deduced from uncertainties in the value of $\Delta^{17}\text{O}$ of O_2 produced by the earth

biosphere calculated from uncertainties in the values of the fractionation coefficients, uncertainties in

the ratio of oceanic to terrestrial biosphere as well as uncertainties on isotopic composition of meteoric

water linked to temporal changes in the hydrological cycle (see Method and Supplementary Table 2).

The solid and dotted black lines display the ratio between biosphere productivity at the considered

time and pre-industrial biosphere productivity as calculated with the model of Blunier et al., 2012² with

two different assumptions for the isotopic composition of meteoric water: solid line is associated with

a constant H_2O anomaly with time while the dotted black line was obtained with a 20 ppm lower

anomaly during the glacial periods. The light grey area represents the maximum uncertainty range of

the biosphere productivity reconstruction, taking into account all uncertainties listed on

Supplementary Table 2 as well as uncertainty in the model used for reconstruction and the biosphere

productivity reconstruction of Blunier et al., 2012².

**Supplementary Figure 3. Variation of the oxygen biosphere productivity compared with pollen and**
 **oceanic records.** a Eccentricity⁶; b atmospheric CO₂¹; c global oxygen biosphere productivity (this
 study) after interpolation to 200 years and 101 binomial smoothing with Igor software; d Trees, shrubs
 and Picea pollen record (%) from El'Gygytgn Lake⁷; e Si/Ti ratio from El'Gygytgn Lake⁷, a proxy of
 biogenic silica normalized to detrital, reflecting the changes in diatom productivity in the lake; f Pollen
 record from oceanic core ODP 646⁸; g Arboreal and Quercus robur pollen records (%) from Lake Ohrid,
 Balkan Peninsula^{9,10}; h Arboreal pollen record (%) from Tenaghi Philippon Lake¹¹; i alkenone mass
 accumulation rate (MAR) ($\mu\text{g}\cdot\text{m}^{-2}\cdot\text{y}^{-1}$) and TOC MAR ($\text{mg}\cdot\text{m}^{-2}\cdot\text{y}^{-1}$) records at Site PS2489-2/ODP1090¹²; j
 CaCO₃ (%) record from Site PS2489-2/ODP1090, Atlantic sector of the Southern Ocean¹³; k TOC/CaCO₃

ratio at Site PS2489-2/ODP1090, Atlantic Southern Ocean¹³. The grey shadow bars represent the period
of rapid increase in CO₂ during deglaciations.

Supplementary Figure 3 combines well-dated palynological and geochemical data covering the last five
deglaciations and related to terrestrial and oceanic biological productivities, respectively. El'Gygytgyn
core age model is based, in first order, on magnetostratigraphy and in second and third orders on the
correlation between sedimentary proxy data to the LR04 stack¹⁴ and insolation patterns⁶. The
chronology of marine core ODP 646 is based on the $\delta^{18}\text{O}$ in *N. pachyderma* and on the correlation with
the stack LR04 of Lisiecki and Raymo¹⁴. The age-model of Lake Ohrid record^{9,10} is based on
tephrochronology on 11 tephra layers¹⁵ and on a second order on the tuning of biogeochemical proxy
data to orbital parameters¹⁶. The Tenaghi Philippon core age-model is based on the correlation
between vegetation changes and March and June perihelion configuration. The age-models of PS2489-
2 and ODP Site 1090 were calculated using the correlation between the alkenone-based SST with the
EDC ice core temperature record using EDC3 chronology^{12,17,18}. Decreases in TOC and alkenone MARS
from PS2489-2/ODP 1090 are correlated, indicating that TOC decrease is not a consequence of the
increase in CaCO₃ in the sediment.

**Supplementary Figure 4. Agreement between reconstructions of biospheric productivity fluxes of**
 **oxygen over the last deglaciation from $\Delta^{17}\text{O}$ of O_2 and output of coupled model equipped with**
 **vegetation and marine productivity model. Top panel: ratio between global biosphere productivity at**
 **time “t” and global biosphere productivity for pre-industrial period (expressed in O_2 flux) obtained by**
 **the coupled IPSL model equipped with the PISCES¹⁹ and ORCHIDEE²⁰ models (crosses, Bopp and**
 **Kageyama, personal communication) and by interpretation of $\Delta^{17}\text{O}$ of O_2 data using the model of**
 **Landais et al. (2007)³ and uncertainty of Supplementary Table 2 (grey envelope). Bottom panel:**
 **evolutions of CO_2 (black)¹ and $\Delta^{17}\text{O}$ of O_2 (green)² as in the main manuscript.**

Sample	Interior sample		Exterior sample	
	$\delta O_2/Ar$	$\Delta^{17}O$ of O_2	$\delta O_2/Ar$	$\Delta^{17}O$ of O_2
GRIP Sample 1	-74	29	-146	13
GRIP Sample 2	-82	50	-134	13
GRIP Sample 3	-94	50	-156	26

**Supplementary Table 1. Comparison of $\delta O_2/Ar$ and $\Delta^{17}O$ of O_2 values between the interior and the**

**exterior part of the ice core**

Sensitivity test	$F_{\text{oce,PST}}/F_{\text{terr,PST}}$	$F_{\text{oce,LGM}}/F_{\text{terr,LGM}}$	$F_{\text{oce,MIS 11}}/F_{\text{terr,MIS 11}}$	$\Delta^{17}\text{O}_{\text{bio,PST}}$	$\Delta^{17}\text{O}_{\text{bio,LGM}}$	$\Delta^{17}\text{O}_{\text{bio,MIS 11}}$	$F_{\text{bio,LGM}}/F_{\text{bio,PST}}$	$F_{\text{bio,MIS 11}}/F_{\text{bio,PST}}$
Average situation	0.52	0.82	0.52	158	195	158	0.69	1.17
High $F_{\text{oce,PST}}/F_{\text{terr,PST}}$	0.59	0.82	0.52	162	195	162	0.71	1.17
Low $F_{\text{oce,PST}}/F_{\text{terr,PST}}$	0.45	0.82	0.52	153	195	153	0.67	1.18
High $F_{\text{oce,LGM}}/F_{\text{terr,LGM}}$	0.52	1.08	0.52	158	203	158	0.66	1.17
Low $F_{\text{oce,LGM}}/F_{\text{terr,LGM}}$	0.52	0.56	0.52	158	183	158	0.75	1.17
High $F_{\text{oce,MIS 11}}/F_{\text{terr,MIS 11}}$	0.52	0.82	0.59	158	195	162	0.69	1.14
Low $F_{\text{oce,MIS 11}}/F_{\text{terr,MIS 11}}$	0.52	0.82	0.45	158	195	153	0.69	1.22
Lowest estimate of the slopes associated with fractionation factor in the water cycle and biosphere (cumulative errors)	0.52	0.82	0.52	129	166	129	0.69	1.22
Highest estimate of the slopes associated with fractionation factor in the water cycle and biosphere (cumulative errors)	0.52	0.82	0.52	187	224	187	0.69	1.14
Maximum uncertainty on photosynthesis fractionation (Eisenstadt et al., 2010) ²¹	0.52	0.82	0.52	193	230	193	0.69	1.14
Maximum uncertainty on the slope of respiration (Stolper et al., 2018) ²²	0.52	0.82	0.52	93	130	93	0.7	1.33
Change of hydrological cycle ($\Delta^{17}\text{O}$ of O_2 from terrestrial biosphere lower by 20 ppm during LGM and 10 ppm higher during MIS 11)	0.52	0.82	0.52	158	183	164	0.75	1.12

**Supplementary Table 2. Estimates of $\Delta^{17}\text{O}_{\text{bio}}$ and associated reconstruction of the productivity for**
**pre-industrial, Last Glacial Maximum (LGM) and MIS 11, using equations S1 and S2 with various**
**sensitivity tests. The first 9 lines (in grey) were directly taken from sensitivity tests of Landais et al.**
**(2007) taking into account uncertainties in the ratio between oceanic and terrestrial biosphere**
**productivity or in the estimates of the fractionation factors used for calculations of $\Delta^{17}\text{O}_{\text{terr}}$ and $\Delta^{17}\text{O}_{\text{oce}}$.**
**Note that these fractionation factors are based on physical processes which do not vary with time so**
**that the associated uncertainties are not independent for pre-industrial, LGM and MIS 11.**
**The last 3 lines correspond to 3 new sensitivity tests for the influence of fractionation during**
**photosynthesis, possible low slope of $\delta^{17}\text{O}$ vs $\delta^{18}\text{O}$ during respiration, and possible changes in the water**

cycle leading to modification of the triple isotopic composition of oxygen in water (^{17}O -excess, see text)
directly transmitted to the $\Delta^{17}\text{O}$ of O_2 produced by terrestrial productivity.
For all sensitivity tests, we considered extreme values so that the reconstructed global productivities
are also showing extreme values.

**Supplementary references**

1. Siegenthaler, U. Stable Carbon Cycle-Climate Relationship During the Late Pleistocene. *Science*
310, 1313–1317 (2005).

2. Blunier, T., Bender, M. L., Barnett, B. & von Fischer, J. C. Planetary fertility during the past 400
938 ka based on the triple isotope composition of O_2 in trapped gases from the Vostok ice core. *Climate of*
939 *the Past* 8, 1509–1526 (2012).

3. Landais, A., Lathiere, J., Barkan, E. & Luz, B. Reconsidering the change in global biosphere
productivity between the Last Glacial Maximum and present day from the triple oxygen isotopic
composition of air trapped in ice cores. *Global Biogeochemical Cycles* 21, GB1025 (2007).

4. Eisenstadt, D., Barkan, E., Luz, B. & Kaplan, A. Enrichment of oxygen heavy isotopes during
photosynthesis in phytoplankton. *Photosynthesis Research* 103, 97–103 (2010).

5. Stolper, D. A., Fischer, W. W. & Bender, M. L. Effects of temperature and carbon source on
the isotopic fractionations associated with O_2 respiration for $^{17}\text{O}/^{16}\text{O}$ and $^{18}\text{O}/^{16}\text{O}$ ratios in *E. coli*.
*Geochimica et Cosmochimica Acta* 240, 152–172 (2018).

6. Laskar, J. et al. A long-term numerical solution for the insolation quantities of the Earth.
*Astronomy & Astrophysics* 428, 261–285 (2004).

7. Melles, M. et al. 2.8 Million Years of Arctic Climate Change from Lake El'gygytyn, NE Russia.
*Science* 337, 315–320 (2012).

8. de Vernal, A. & Hillaire-Marcel, C. Natural Variability of Greenland Climate, Vegetation, and
Ice Volume During the Past Million Years. *Science* 320, 1622–1625 (2008).

9. Sadori, L. et al. Pollen-based paleoenvironmental and paleoclimatic change at Lake Ohrid
(south-eastern Europe) during the past 500 ka. *Biogeosciences* 13, 1423–1437 (2016).

10. Kousis, I. et al. Centennial-scale vegetation dynamics and climate variability in SE Europe during
Marine Isotope Stage 11 based on a pollen record from Lake Ohrid. *Quaternary Science Reviews* 190,
20–38 (2018).

11. Tzedakis, P. C., Hooghiemstra, H. & Pälike, H. The last 1.35 million years at Tenaghi Philippon:
revised chronostratigraphy and long-term vegetation trends. *Quaternary Science Reviews* 25, 3416–
3430 (2006).

- 12. Martínez-García, A. et al. Links between iron supply, marine productivity, sea surface
temperature, and CO₂ over the last 1.1 Ma. *Paleoceanography* 24, PA1207 (2009).
- 13. Diekmann, B. & Kuhn, G. Sedimentary record of the mid-Pleistocene climate transition in the
southeastern South Atlantic (ODP Site 1090). *Palaeogeography, Palaeoclimatology, Palaeoecology* 182,
241–258 (2002).
- 14. Lisiecki, L. E. & Raymo, M. E. A Pliocene-Pleistocene stack of 57 globally distributed benthic δ
18 O records. *Paleoceanography* 20, PA1003 (2005).
- 15. Leicher, N. et al. First tephrostratigraphic results of the DEEP site record from Lake Ohrid
(Macedonia and Albania). *Biogeosciences* 13, 2151–2178 (2016).
- 16. Francke, A. et al. Sedimentological processes and environmental variability at Lake Ohrid
(Macedonia, Albania) between 637 ka and the present. *Biogeosciences* 13, 1179–1196 (2016).
- 17. Jouzel, J. et al. Orbital and Millennial Antarctic Climate Variability over the Past 800,000
Years. *Science* **317**, 793–796 (2007).
- 18. Parrenin, F. et al. The EDC3 chronology for the EPICA Dome C ice core. *Clim. Past* 13 (2007).
- 19. Aumont, O., Ethé, C., Tagliabue, A., Bopp, L. & Gehlen, M. PISCES-v2: an ocean
biogeochemical model for carbon and ecosystem studies. *Geoscientific Model Development* **8**, 2465–
2513 (2015).
- 20. Krinner, G. et al. A dynamic global vegetation model for studies of the coupled atmosphere-
biosphere system. *Global Biogeochemical Cycles* **19**, (2005).
- 21. Eisenstadt, D., Barkan, E., Luz, B. & Kaplan, A. Enrichment of oxygen heavy isotopes during
photosynthesis in phytoplankton. *Photosynthesis Research* **103**, 97–103 (2010).
- 22. Stolper, D. A., Fischer, W. W. & Bender, M. L. Effects of temperature and carbon source on
the isotopic fractionations associated with O₂ respiration for ¹⁷O/¹⁶O and ¹⁸O/¹⁶O ratios in *E. coli*.
*Geochimica et Cosmochimica Acta* **240**, 152–172 (2018).

REVIEWERS' COMMENTS:

Reviewer #3 (Remarks to the Author):

I appreciate the work the authors have done to respond to my comments. The authors argue strongly for their interpretation of the data. I believe that the manuscript plus the SOM collectively rise to the level that the paper should be accepted. My concern is that readers have to wade through the very detailed SOM in order to understand the uncertainties, which is to say, in order to understand how much weight they should give to the conclusions. I feel that this division of the substance between the main text and the SOM is an editorial question that needs to be resolved between the editor and the author, and I do not need to see this paper again before it is published.

We thank Reviewer #3 for his/her comments that helped providing clearer information on the uncertainty of our study to the reader. As for his/her concern about the SOM, we now detail the uncertainty tests in the Results part of the main text (lines 192 to 237): “The reconstructed biosphere productivity using the Landais et al. (2007)³³ formulation and uncertainties is detailed in Supplementary Table 2 for different time periods of interest within this study, i.e. the pre-industrial, the LGM and the MIS 11. On average, we obtain a biosphere productivity at the beginning of MIS 11 which is 17 % higher than during MIS 1 and a biosphere productivity during the LGM which is 31% lower than during MIS 1, a result in agreement with output of the IPSL coupled model equipped with vegetation and ocean productivity modules (Supplementary Figure 4).

As in Landais et al. (2007)³³, it is shown in Supplementary Table 2 that the uncertainty in fractionation coefficients as well as on the ratio of oceanic to terrestrial productivity is leading to uncertainties in the reconstructed past global productivity. The fractionation coefficients are based on physical properties and are hence not expected to vary with time so that the possible bias on these coefficients should apply to the different periods. On opposite, the ratio of oceanic to terrestrial productivity is expected to vary with time and is a large source of uncertainty. We thus present in Supplementary Table 2 calculations performed with the largest range of possible ratios of oceanic to terrestrial productivity estimated in Landais et al. (2007)³³. Since the publication of Landais et al. (2007)³³, new estimates of fractionation coefficients within the oxygen cycle are available^{19,35,36} and influence of the water cycle organisation has also been suggested²². As a consequence, we performed 3 additional types of sensitivity tests to better estimate the uncertainties related to fractionation processes within the photosynthesis³⁷, respiration³⁶ and water cycle²² (see SOM for details).

First, we address the uncertainty in photosynthesis fractionation. Some marine species show fractionation during marine photosynthesis³⁷ while it was assumed in Landais et al. (2007)³³ that photosynthesis does not fractionate³⁸. We performed sensitivity tests with the largest fractionation effects during marine photosynthesis observed in Eisenstadt et al. (2010)³⁷ and obtained a MIS 11 productivity level is decreased by 3% with such fractionation effect.

The second uncertainty we tested is the uncertainty in respiration fractionation. Recent studies have highlighted large variations in the relationship between $\delta^{17}\text{O}$ and $\delta^{18}\text{O}$ during respiration linked to temperature variations³⁶. When taking into account the maximum effect, i.e. a decrease of 0.005 for the slope of the relationship between $\ln(\delta^{17}\text{O}+1)$ and $\ln(\delta^{18}\text{O}+1)$ during respiration, we end up with a resulting biosphere productivity reconstruction 16% higher during MIS 11 than the average situation. Then, we estimated the uncertainty in the fractionation within water cycle. The general assumption in Landais et al. (2007)³³ was that the relationship between $\delta^{17}\text{O}$ and $\delta^{18}\text{O}$ of water (meteoric water line) remains the same over glacial-interglacial cycles. However, measurements in the Vostok ice core have shown that the 17O-excess, defined as $17\text{O-excess} = \ln(1+\delta^{17}\text{O}) - 0.528 \cdot \ln(1+\delta^{18}\text{O})$, is lower by up to 20 ppm during the last glacial maximum³⁹. While it has been shown that this is a local effect⁴⁰, we still performed a sensitivity test with decreasing 17O-excess of all continental meteoric waters by 20 ppm during glacial periods and increasing 17O-excess by 10 ppm during MIS 11 with respect to our current interglacial period.

Taking into account such variations, the reconstructed biosphere productivity does not differ by more than 6% from the case without change in meteoric water 17O-excess (Supplementary Table 2). Finally, we also took into account the uncertainty in the model used by Landais et al. (2007) by combining the above reconstruction with an alternative reconstruction of global oxygen productivity using an alternative model, the one of Blunier et al. (2012)²², forced by our $\Delta^{17}\text{O}$ of O_2 data (Supplementary Figure 2).”